# Grassmann Manifold Flows for Stable Shape Generation

**Ryoma Yataka**[1,2]**, Kazuki Hirashima**[1]**, Masashi Shiraishi**[1]
[1]Information Technology R&D Center (ITC), Mitsubishi Electric Corporation,
Kamakura, Kanagawa, 247-8501, Japan
[2]Mitsubishi Electric Research Laboratories (MERL), Cambridge, MA 02139, USA
`{Yataka.Ryoma@dw.MitsubishiElectric.co.jp, yataka@merl.com}`

## Abstract

Recently, studies on machine learning have focused on methods that use symmetry implicit in a specific manifold as an inductive bias. Grassmann manifolds provide the ability to handle fundamental shapes represented as shape spaces, enabling stable shape analysis. In this paper, we present a novel approach in which we establish the theoretical foundations for learning distributions on the Grassmann manifold via continuous normalization flows, with the explicit goal of generating stable shapes. Our approach facilitates more robust generation by effectively eliminating the influence of extraneous transformations, such as rotations and inversions, through learning and generating within a Grassmann manifold designed to accommodate the essential shape information of the object. The experimental results indicated that the proposed method could generate high-quality samples by capturing the data structure. Furthermore, the proposed method significantly outperformed state-of-the-art methods in terms of the log-likelihood or evidence lower bound. The results obtained are expected to stimulate further research in this field, leading to advances for stable shape generation and analysis.

## 1 Introduction

Many machine learning algorithms are designed to automatically learn and extract latent factors that explain a specific dataset. Symmetry is known to be an inductive bias (that is, prior knowledge other than training data that can contribute significantly to learning results) for learning latent factors (Cohen & Welling (2016, 2017); Weiler & Cesa (2019); Satorras et al. (2021); Bronstein et al. (2021); Puny et al. (2022)). Moreover, they exist in many phenomena in the natural sciences. If a target $M$ is invariant[1] when the operation $g \in S$ designated by $S$ applied to $M$, $M$ has symmetry $S$. For example, a sphere remains a sphere even if a rotation $R$ is applied; a symmetric shape remains symmetric even if a left–right reversal is applied.

Recent studies have focused on methods incorporating symmetry into models based on equivariance and invariance, when the data space forms a non-Euclidean space (a sphere $S^n$ or a special unitary group $SU(n)$) (Cohen et al. (2018, 2019b); Graham et al. (2020); Haan et al. (2021); Boyda et al. (2021a)). Among them, discriminative and generative models for subspace data (e.g., shape matrices of point cloud data such as three-dimensional molecules and general object shapes) have been proposed as viable approaches using Grassmann manifolds, which are quotient manifolds obtained by introducing a function that induces invariance with respect to orthogonal transformations into

---

[1]Invariance implies the transformation $\pi$ satisfies $\pi(x) = \pi(x') = \pi(g(x))$ for $x' = g(x)$ obtained by applying the operation $g$ to the input $x$. Transformation $\pi$ is considered invariant with respect to operation $g$. A transformation $\pi$ is considered equivariant with respect to an operation $p$ if $\pi$ satisfies $g(\pi(x)) = \pi(g(x))$. Invariance is a special case of equivariance where $\pi(g)$ is an identity transformation.

37th Conference on Neural Information Processing Systems (NeurIPS 2023).

a set of orthonormal basis matrices namely Stiefel Manifold. Numerous studies have confirmed their effectiveness (Liu et al. (2003); Hamm & Lee (2008); Turaga et al. (2008); Harandi et al. (2011); Fan et al. (2011); Lui (2012); Huang et al. (2015, 2018); Souza et al. (2020); Haitman et al. (2021); Doronina et al. (2022); Souza et al. (2022)). Shape theory studies the equivalent class of all configurations that can be obtained by a specific class of transformation (e.g. linear, affine, projective) on a single basis shape (Patrangenaru & Mardia (2003); Sepiashvili et al. (2003); Begelfor & Werman (2006)), and it can be shown that affine and linear shape spaces for specific configurations can be identified by points on the Grassmann manifold. By applying this model to inverse molecular design for stable shape generation to discover new molecular structures, they can contribute to the development of drug discovery, computational anatomy, and materials science (Sanchez-Lengeling & Aspuru-Guzik (2018); Bilodeau et al. (2022)).

Continuous normalizing flows (CNFs; Chen et al. (2018); Grathwohl et al. (2019); Lou et al. (2020); Kim et al. (2020); Mathieu & Nickel (2020)) are generative models that have attracted attention in recent years along with the variational auto-encoders (VAE; Kingma & Welling (2014)), generative adversarial networks (GAN; Goodfellow et al. (2014)), autoregressive models (Germain et al. (2015); Oord et al. (2016)), and diffusion models (Sohl-Dickstein et al. (2015); Ho et al. (2020); Song et al. (2021); Huang et al. (2022); De Bortoli et al. (2022)). The CNF is a method with theoretically superior properties that allow rigorous inference and evaluation of the log-likelihood. It can be trained by maximum likelihood using the change of variables formula, which allows to specify a complex normalized distribution implicitly by warping a normalized base distribution through an integral of continuous-time dynamics. In this paper, we present a novel approach with the explicit goal of generating stable shapes. We employ the CNF to achieve stable shape generation on a Grassmann manifold; however, previous studies have lacked the theoretical foundation to handle the CNF on the Grassmann manifold. To construct this, we focused on the quotient structure of a Grassmann manifold and translated the problem of flow learning on a Grassmann manifold into that of preserving equivariance for orthogonal groups on a Stiefel manifold, which is its total space. The contributions of this study are as follows.

- A theory and general framework for learning flows on a Grassmann manifold are proposed. In our setting, we can train flows on a Grassmann manifold of arbitrary dimension. To the best of our knowledge, this is the first study in which a CNF was constructed on a Grassmann manifold via a unified approach, focusing on the quotient structure.

- The validity of the proposed approach was demonstrated by learning densities on a Grassmann manifold using multiple artificial datasets with complex data distributions. In particular, orthogonally transformed data were proven to be correctly learned without data augmentations by showing that untrained transformed (i.e., rotated or mirrored) data can be generated from the trained model. The model was evaluated on multiple patterns of training data and performed well with a small amount of training data.

- The effectiveness of our approach was confirmed by its state-of-the-art performance in a molecular positional generation task.

## 2 Related Works

**Symmetry-based Learning**   The concept of equivariance has been studied in recent years to leverage the symmetries inherent in data (Cohen & Welling (2017); Kondor & Trivedi (2018); Cohen et al. (2018, 2019b,a); Finzi et al. (2020); Haan et al. (2021)). Early work by Cohen & Welling (2017) indicated that when data are equivariant, they can be processed with a lower computational cost and fewer parameters. In the context of CNFs (Rezende & Mohamed (2015); Chen et al. (2018); Grathwohl et al. (2019)), which are generative models, Rezende et al. (2019), Köhler et al. (2020) and Garcia Satorras et al. (2021) proposed equivariant normalizing flows to learn symmetric densities on Euclidean spaces. Furthermore, the symmetries that appear in learning densities on a manifold were introduced by Boyda et al. (2021b) and Katsman et al. (2021) as a conjugate equivariant flow on $SU(n)$, which is a quotient manifold, for use in lattice gauge theory. However, a normalizing flow on a Grassmann manifold capable of handling subspace data has not yet been established.

**Shape Data Analysis and Other Applications with Subspace on the Grassmann Manifold**   $k$-dimensional shape data (Begelfor & Werman (2006); Yoshinuma et al. (2016); Haitman et al. (2021);

Doronina et al. (2022)) such as point clouds are essentially represented as subspace (shape space (Sepiashvili et al. (2003); Srivastava et al. (2005); Begelfor & Werman (2006); Yataka & Fukui (2017)) data on a Grassmann manifold. Furthermore, subspace data can be obtained on many types of data and can provide advantages such as practicability and noise robustness. For example, multiview image sets or video data (Fan et al. (2011); Turaga et al. (2011); Lui (2012); Alashkar et al. (2016)), signal data (Srivastava & Klassen (2004); Gatto et al. (2017); Souza et al. (2019); Yataka et al. (2019)), and text data (Shimomoto et al. (2021)) are often provided as a set of feature vectors. Such raw data in matrix form is not very useful owing to its considerable size and noise. The analysis of its eigenspace, or column subspace, is important because alternatively, the raw data matrix can be well approximated by a low-dimensional subspace with basis vectors corresponding to the maximum eigenvalues of the matrix. Therefore, they inevitably give rise to the necessity of analyzing them on a Grassmann manifold.

# 3 Mathematical Preliminaries

In this section, the basic mathematical concepts covered in this paper are described. Further details on the fundamentals of a Grassmann manifold are summarized in the Appendix D.

## 3.1 Grassmann Manifold Defined as Quotient Manifold

**Definition of a Grassmann Manifold** A Grassmann manifold $\mathrm{Gr}(k, D)$ is a set of $k$-dimensional subspaces $\mathrm{span}(\boldsymbol{Y})$ ($\boldsymbol{Y}$ is a matrix of $k$ basis vectors and the function $\mathrm{span}(\cdot)$ is onto $\mathrm{Gr}(k, D)$) in the $D$-dimensional Euclidean space $\mathbb{R}^D$, and is defined as $\mathrm{Gr}(k, D) = \{\mathrm{span}(\boldsymbol{Y}) \subset \mathbb{R}^D \mid \dim(\mathrm{span}(\boldsymbol{Y})) = k\}$ (Absil et al. (2008)). The $\mathrm{span}(\boldsymbol{Y})$ is the same subspace regardless of a $k$-dimensional rotation or $k$-dimensional reflection applied to $\boldsymbol{Y}$ on which it is spanned. With respect to the compact Stiefel manifold $\mathrm{St}(k, D) := \{\boldsymbol{Y} \in \mathbb{R}^{D \times k} \mid \boldsymbol{Y}^\top \boldsymbol{Y} = \boldsymbol{I}_k\}$ defined as the set of $D \times k$-orthonormal basis matrices $\boldsymbol{Y}$, the equivalence class of $\boldsymbol{Y} \in \mathrm{St}(k, D)$ determined from the equivalence relation $\sim$[2] is defined by $[\boldsymbol{Y}] := \pi(\boldsymbol{Y}) = \{\boldsymbol{Y}\boldsymbol{Q} \in \mathrm{St}(k, D) \mid \boldsymbol{Q} \in \mathcal{O}(k)\}$, where $\pi(\boldsymbol{Y})$ is a continuous surjection referred to as a quotient map. The equivalence class corresponds one-to-one with the $k$-dimensional subspace:

$$[\boldsymbol{Y}] = [\boldsymbol{X}] \iff \mathrm{span}(\boldsymbol{Y}) = \mathrm{span}(\boldsymbol{X}), \tag{1}$$

where $\boldsymbol{Y} \in \mathrm{St}(k, D)$ is the representative of $[\boldsymbol{Y}]$. $\mathrm{span}(\boldsymbol{Y})$ is termed invariant under $\sim$. The quotient set composed of such $[\boldsymbol{Y}]$ as a whole can introduce the structure of a manifold (Sato & Iwai (2014)).

**Definition 1.** *A Grassmann manifold as a quotient manifold is defined as follows:*

$$\mathrm{Gr}(k, D) := \mathrm{St}(k, D) / \mathcal{O}(k) = \{[\boldsymbol{Y}] = \pi(\boldsymbol{Y}) \mid \boldsymbol{Y} \in \mathrm{St}(k, D)\}, \tag{2}$$

*where $\mathrm{St}(k, D) / \mathcal{O}(k)$ is the quotient manifold by the $k$-dimensional orthogonal group $\mathcal{O}(k)$ with the total space $\mathrm{St}(k, D)$, and $\pi(\boldsymbol{Y})$ is the quotient map $\pi : \mathrm{St}(k, D) \to \mathrm{St}(k, D) / \mathcal{O}(k)$.*

**Tangent Space and Vector Field** Let $T_{[\boldsymbol{Y}]} \mathrm{Gr}(k, D)$ be the tangent space of $[\boldsymbol{Y}] \in \mathrm{Gr}(k, D)$. As the point $[\boldsymbol{Y}]$ is not a matrix, $T_{[\boldsymbol{Y}]} \mathrm{Gr}(k, D)$ cannot be represented by a matrix. Therefore, treating these directly in numerical calculations is challenging. To solve this problem, we can use the representative $\boldsymbol{Y} \in \mathrm{St}(k, D)$ for $[\boldsymbol{Y}]$ and the tangent vector $\overline{\boldsymbol{\xi}}_{\boldsymbol{Y}}^{\mathrm{h}} \in T_{\boldsymbol{Y}}^{\mathrm{h}} \mathrm{St}(k, D)$, which is referred to as the horizontal lift of $\boldsymbol{\xi}_{[\boldsymbol{Y}]} \in T_{[\boldsymbol{Y}]} \mathrm{Gr}(k, D)$, for $\boldsymbol{\xi}_{[\boldsymbol{Y}]}$. These facilitate computation with matrices (Absil et al. (2008)). $T_{\boldsymbol{Y}}^{\mathrm{h}} \mathrm{St}(k, D)$ is a subspace of the tangent space $T_{\boldsymbol{Y}} \mathrm{St}(k, D)$ at $\boldsymbol{Y}$, which is referred to as a horizontal space, and $\overline{\boldsymbol{\xi}}_{\boldsymbol{Y}}^{\mathrm{h}}$ is referred to as a horizontal vector. The tangent bundle $T \mathrm{Gr}(k, D) = \bigcup_{[\boldsymbol{Y}] \in \mathrm{Gr}(k, D)} T_{[\boldsymbol{Y}]} \mathrm{Gr}(k, D)$ that sums up the tangent spaces $T_{[\boldsymbol{Y}]} \mathrm{Gr}(k, D)$ form vector fields

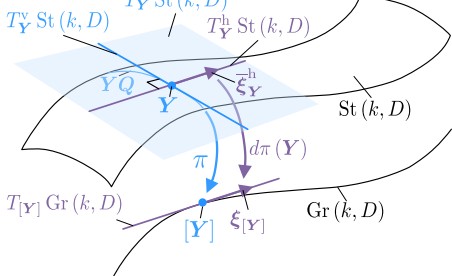

Figure 1: Conceptual diagram of spaces with horizontal lift.

---

[2]When there exists some $\boldsymbol{Q} \in \mathcal{O}(k)$ such that $\boldsymbol{X} = \boldsymbol{Y}\boldsymbol{Q}$, then $\boldsymbol{X}$ and $\boldsymbol{Y}$ are defined to be equivalences $\boldsymbol{X} \sim \boldsymbol{Y}$. Appendix D.2 provides further details.

$\mathsf{X} : \mathrm{Gr}(k, D) \to T\,\mathrm{Gr}(k, D)$. A conceptual diagram of the various spaces is shown in Figure 1. Further details on these concepts can be found in Appendix D.4.

## 3.2 Manifold Normalizing Flow

Let $(\mathcal{M}, h)$ be a Riemannian manifold. We consider the time evolution of a base point $\boldsymbol{x} = \gamma(0)$, $\gamma : [0, \infty) \to \mathcal{M}$, whose velocity is expressed by a vector field $\mathsf{X}(t, \gamma(t))$. Intuitively, $\mathsf{X}(t, \gamma(t))$ represents the direction and speed at which $\boldsymbol{x}$ moves on the curve $\gamma(t)$. Let $T_{\boldsymbol{x}}\mathcal{M}$ be the tangent space at $\boldsymbol{x}$ and $T\mathcal{M} = \bigcup_{\boldsymbol{x} \in \mathcal{M}} T_{\boldsymbol{x}}\mathcal{M}$ be the tangent bundle. The time evolution of a point according to a vector field $\mathsf{X} : \mathcal{M} \times \mathbb{R} \to T\mathcal{M}$ is expressed by the differential equation $\frac{d\gamma(t)}{dt} = \mathsf{X}(t, \gamma(t)), \gamma(0) = \boldsymbol{x}$. Let $F_{\mathsf{X}, T} : \mathcal{M} \to \mathcal{M}$ be defined as the map from $^{\forall}\boldsymbol{x} \in \mathcal{M}$ to the evaluated value at time $T$ on the curve $\gamma(t)$ starting at $\boldsymbol{x}$. This map $F_{\mathsf{X}, T}$ is known as the flow of $\mathsf{X}$ (Lee (2003)). Recently, Mathieu & Nickel (2020) introduced the Riemann CNF (RCNF), wherein the random variable $\boldsymbol{z}(t) \in \mathcal{M}$ is assumed to be time-dependent and the change in its log-likelihood follows the instantaneous change formula for the variable. This is an extension of the CNF (Chen et al. (2018); Grathwohl et al. (2019)) to a Riemannian manifold. Specifically, when $p_\theta$ is the density parameterized by $\theta$, the derivative of the log-likelihood is expressed as $\frac{d \log p_\theta(\boldsymbol{z}(t))}{dt} = -\mathrm{div}(\mathsf{X}_\theta(t, \boldsymbol{z}(t)))$, where $\mathsf{X}_\theta$ is the vector field parameterized by $\theta$ and $\mathrm{div}(\cdot)$ is the divergence. By integrating this over time, the sum of the changes in the log-likelihood with flow $F_{\mathsf{X}, t_1 \theta}$ can be computed:

$$\log p_\theta(\boldsymbol{z}(t_1)) = \log p\left(F_{\mathsf{X}, t_1 \theta}^{-1}(\boldsymbol{z}(t_1))\right) - \int_{t_0}^{t_1} \mathrm{div}(\mathsf{X}_\theta(t, \boldsymbol{z}(t)))dt. \tag{3}$$

## 4 Invariant Densities from Grassmann Manifold Flow

This section provides a tractable and efficient method for learning densities on a Grassmann manifold $\mathrm{Gr}(k, D)$. The method for preserving the flow on $\mathrm{Gr}(k, D)$ is non-trivial. Therefore, we derive the following implications.

1. **Vector field on $\mathrm{Gr}(k, D)$ $\Leftrightarrow$ Flow on $\mathrm{Gr}(k, D)$ (Proposition 1)**.

2. **Flow on $\mathrm{Gr}(k, D)$ $\Leftrightarrow$ Probability density on $\mathrm{Gr}(k, D)$ (Proposition 2)**.

3. **Construction of a prior probability density for an efficient sampling (Proposition 3)**.

The essence of the proofs is to show that Corollary 2 (Homogeneity Property (4) with regard to $^{\forall}\boldsymbol{Q} \in \mathcal{O}(k)$, which is equivariance in the horizontal space) is satisfied.

$$\overline{\boldsymbol{\xi}}_{\boldsymbol{Y}\boldsymbol{Q}}^{\mathrm{h}} = \overline{\boldsymbol{\xi}}_{\boldsymbol{Y}}^{\mathrm{h}}\boldsymbol{Q}, \tag{4}$$

where $\overline{\boldsymbol{\xi}}_{\boldsymbol{Y}}^{\mathrm{h}}$ is a horizontal lift at representative $\boldsymbol{Y}$ relative to $\boldsymbol{\xi}_{[\boldsymbol{Y}]} \in T_{[\boldsymbol{Y}]}\,\mathrm{Gr}(k, D)$. We defer the proofs of all propositions to Appendix B. These series of propositions show that by using a prior distribution on $\mathrm{Gr}(k, D)$ that can be easily sampled, a flow that generates a complex probability density distribution on $\mathrm{Gr}(k, D)$ can be obtained.

### 4.1 Construction of Flow from Vector Field

To construct flows on $\mathrm{Gr}(k, D)$, we use tools in the theory of manifold differential equations. In particular, there is a natural correspondence between the vector fields on $\mathrm{Gr}(k, D)$ and the flows on $\mathrm{Gr}(k, D)$. This is formalized in the following proposition.

**Proposition 1.** *Let $\mathrm{Gr}(k, D)$ be a Grassmann manifold, $\mathsf{X}$ be any time-dependent vector field on $\mathrm{Gr}(k, D)$, and $F_{\mathsf{X}, T}$ be a flow on a $\mathsf{X}$. Let $\overline{\mathsf{X}}$ be any time-dependent horizontal lift and $\overline{F}_{\overline{\mathsf{X}}, T}$ be a flow of $\overline{\mathsf{X}}$. $\overline{\mathsf{X}}$ is a vector field on $\mathrm{St}(k, D)$ if and only if $\overline{F}_{\overline{\mathsf{X}}, T}$ is a flow on $\mathrm{St}(k, D)$ and satisfies invariance condition $\overline{\mathsf{X}} \sim \overline{\mathsf{X}}'$ for all $\overline{F}_{\overline{\mathsf{X}}, T} \sim \overline{F}_{\overline{\mathsf{X}}', T}$. Therefore, $\mathsf{X}$ is a vector field on $\mathrm{Gr}(k, D)$ if and only if $F_{\mathsf{X}, T} := \left[\overline{F}_{\overline{\mathsf{X}}, T}\right]$ is a flow on $\mathrm{Gr}(k, D)$, and vice versa.*

**Algorithm 1** Random Sampling from $p_{\mathrm{Gr}(k,D)}\left([\boldsymbol{X}];[\boldsymbol{M}],\boldsymbol{U},\boldsymbol{V}\right)$

---

**Require:** A mean matrix $\boldsymbol{M} \in \mathrm{St}(k,D)$, a rows covariance matrix $\boldsymbol{U} \in \mathbb{R}^{D \times D}$ and a columns covariance matrix $\boldsymbol{V} \in \mathbb{R}^{k \times k}$.
1: Sample a $\mathrm{Vec}\left(\boldsymbol{Z}\right) \in \mathbb{R}^{Dk}$ from Gaussian Distribution $\mathcal{N}_{Dk}\left(\boldsymbol{0},\boldsymbol{V} \otimes \boldsymbol{U}\right)$ and reshape to $\boldsymbol{Z} \in \mathbb{R}^{D \times k}$.
2: Compute a projected horizontal vector $\overline{\boldsymbol{\xi}}_{\boldsymbol{M}}^{\mathrm{h}} = \boldsymbol{Z} - \boldsymbol{M}\left(\boldsymbol{M}^{\top}\boldsymbol{Z}\right) \in T_{\boldsymbol{M}}^{\mathrm{h}}\,\mathrm{St}(k,D)$.
3: Compute a representative $\boldsymbol{Y} = \overline{R}_{\boldsymbol{M}}\left(\overline{\boldsymbol{\xi}}_{\boldsymbol{M}}^{\mathrm{h}}\right) = \boldsymbol{M} + \overline{\boldsymbol{\xi}}_{\boldsymbol{M}}^{\mathrm{h}} - \left(\frac{1}{2}\boldsymbol{M} + \frac{1}{4}\overline{\boldsymbol{\xi}}_{\boldsymbol{M}}^{\mathrm{h}}\right)\left(\boldsymbol{I}_k + \frac{1}{4}\overline{\boldsymbol{\xi}}_{\boldsymbol{M}}^{\mathrm{h}\top}\overline{\boldsymbol{\xi}}_{\boldsymbol{M}}^{\mathrm{h}}\right)^{-1}\overline{\boldsymbol{\xi}}_{\boldsymbol{M}}^{\mathrm{h}\top}\overline{\boldsymbol{\xi}}_{\boldsymbol{M}}^{\mathrm{h}}$
   of the equivalence class $[\boldsymbol{Y}] \in \mathrm{Gr}(k,D)$.

---

### 4.2 Construction of Probability Densities with Flow

We show that the flow on $\mathrm{Gr}(k,D)$ induces density on $\mathrm{Gr}(k,D)$.

**Proposition 2.** *Let $\mathrm{Gr}(k,D)$ be a Grassmann manifold. Let $p$ be the probability density on $\mathrm{Gr}(k,D)$ and $F$ be the flow on $\mathrm{Gr}(k,D)$. Suppose $\overline{p}$ is a density on $\mathrm{St}(k,D)$ and $\overline{F}$ is a flow on $\mathrm{St}(k,D)$. Then, the distribution $\overline{p}_{\overline{F}}$ after transformations by $\overline{F}$ is also a density on $\mathrm{St}(k,D)$. Further, the invariance condition $\overline{p}_{\overline{F}} \sim \overline{p}_{\overline{F}'}$ is satisfied for all $\overline{F} \sim \overline{F}'$. Therefore, $p_F := [\overline{p}_{\overline{F}}]$ is a distribution on $\mathrm{Gr}(k,D)$.*

In the context of the RCNF, Proposition 2 implies that the application of a flow on $\mathrm{Gr}(k,D)$ to a prior distribution on $\mathrm{Gr}(k,D)$ results in a probability density on $\mathrm{Gr}(k,D)$. Thus, the problem of constructing a probability density on $\mathrm{Gr}(k,D)$ is reduced to that of constructing a vector field.

### 4.3 Prior Probability Density Function

To construct a flow on $\mathrm{Gr}(k,D)$, a prior distribution that is easy to sample as a basis for the transformation is required, although the method for constructing such a distribution is non-trivial. In this study, a distribution based on the matrix-variate Gaussian distribution is introduced as a prior on $\mathrm{Gr}(k,D)$ that is easy to sample, and a flow on $\mathrm{Gr}(k,D)$ is constructed.

**Proposition 3.** *The distribution $p_{\mathrm{Gr}(k,D)}$ on a Grassmann manifold $\mathrm{Gr}(k,D)$ based on the matrix-variate Gaussian distribution $\mathcal{MN}$ can be expressed as follows.*

$$p_{\mathrm{Gr}(k,D)}\left([\boldsymbol{X}];[\boldsymbol{M}],\boldsymbol{U},\boldsymbol{V}\right) = V_{\mathrm{Gr}(k,D)}\mathcal{MN}\left(\overline{\boldsymbol{\xi}}_{\boldsymbol{M}}^{\mathrm{h}};\boldsymbol{0},\boldsymbol{U},\boldsymbol{V}\right)\left|\det\left(\nabla_{\overline{\boldsymbol{\xi}}_{\boldsymbol{M}}^{\mathrm{h}}}\overline{R}_{\boldsymbol{M}}\right)\right|, \qquad (5)$$

*where $\boldsymbol{M}$ is an orthonormal basis matrix denoting the mean of the distribution, $\boldsymbol{U}$ is a positive definite matrix denoting the row directional variance, $\boldsymbol{V}$ is a positive definite matrix denoting the column directional variance, and $\overline{\boldsymbol{\xi}}_{\boldsymbol{M}}^{\mathrm{h}}$ is a random sample from $\mathcal{MN}$ in an $(D-k) \times k$-dimensional horizontal space $T_{\boldsymbol{M}}^{\mathrm{h}}\,\mathrm{St}(k,D)$. $V_{\mathrm{Gr}(k,D)}$ denotes the total volume of $\mathrm{Gr}(k,D)$ defined by (113), $\overline{R}_{\boldsymbol{M}}$ denotes the horizontal retraction at $\boldsymbol{M}$, and $\left|\det\left(\nabla_{\overline{\boldsymbol{\xi}}_{\boldsymbol{M}}^{\mathrm{h}}}\overline{R}_{\boldsymbol{M}}\right)\right|$ denotes the Jacobian.*

A retraction is a map for communicating data between a manifold and its tangent bundles and is a first-order approximation of an exponential map (Zhu & Sato (2021)). Various methods of retraction have been proposed at (Absil et al. (2008); Fiori et al. (2015); Zhu & Sato (2021)); in particular, the one based on the Cayley transform is differentiable and does not require matrix decomposition. We use (40) as a Cayley transform based horizontal retraction. Further details are provided in Appendix D.6.

## 5 Learning Probability Densities with Flow

### 5.1 Training Paradigms

Using the results of Section 4, a flow model on a Grassmann manifold $\mathrm{Gr}(k,D)$ is constructed. Herein, the flow model on $\mathrm{Gr}(k,D)$ is expressed as **GrCNF**. In the proposed GrCNF, the probability density function in Section 4.3 is first constructed as a prior distribution. Subsequently, the vector field $\mathsf{X}_\theta : \mathrm{Gr}(k,D) \times \mathbb{R} \to T\,\mathrm{Gr}(k,D)$ that generates the flow model $F_{\mathsf{X}_\theta,T}$ is constructed using a neural network, wherein stepwise integration on the manifold occurs in accordance with (3). The

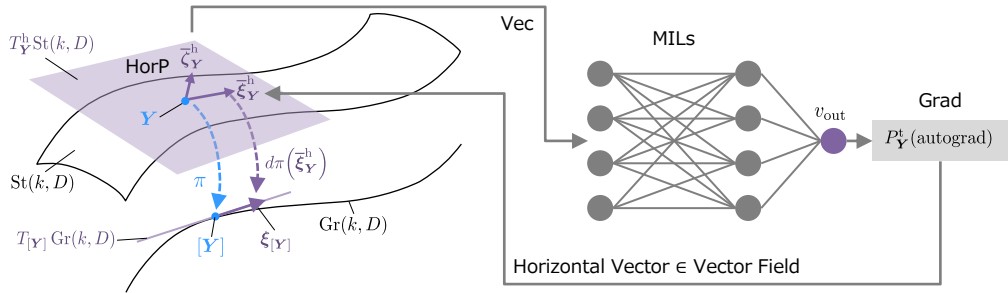

Figure 2: Conceptual diagram of the vector field calculation procedure. HorP denotes the horizontal projection layer, MILs denotes the multiple intermediate layers, and Grad denotes the horizontal lift layer. The sequence of procedure represents that, given an input of $Y \in [Y]$, a horizontal vector $\overline{\xi}_Y^{\mathrm{h}}$, which is equivalent to determining $\xi_{[Y]}$ since $\overline{\xi}_Y^{\mathrm{h}}$ corresponds one-to-one with $\xi_{[Y]}$, is obtained as a result.

RCNF (Mathieu & Nickel (2020)) framework is used for loss function, integration and divergence calculations. In the integration steps, the ordinary differential equation (ODE) is solved using the ODE solver with orthogonal integration. The learnable parameters are updated using backpropagation to maximize the sum of the computed log-likelihood. In order to efficiently compute gradients, we introduce an adjoint method to calculate analytically the derivative of a manifold ODE, based on the results of Lou et al. (2020).

## 5.2   Sampling Algorithm from Prior

We propose a sampling algorithm on $\mathrm{Gr}(k, D)$ derived from the Cayley transform, according to the results of Section 4.3. The algorithm of sampling from a distribution on $p_{\mathrm{Gr}(k,D)}$ is described in Algorithm 1. Vec denotes the map of vertically concatenating matrices and converting them into a vector and $\boldsymbol{I}_D$ is a $D \times D$ identity matrix.

## 5.3   Construction of Vector Field

We describe the method for constructing the vector field $\mathsf{X}_\theta$. The learnable parameters $\theta$ are feedforward neural networks, which accept the representative $Y \in \mathrm{St}(k, D)$ of $[Y] \in \mathrm{Gr}(k, D)$ inputs and output a horizontal lift $\overline{\xi}_Y^{\mathrm{h}} \in T_Y^{\mathrm{h}} \mathrm{St}(k, D)$ of $\xi_{[Y]} \in T_{[Y]} \mathrm{Gr}(k, D)$ that satisfies (4). The structure of $\mathsf{X}_\theta$ is important because it directly determines the ability of the distribution to be represented. To address these geometric properties, the following specific input and intermediate layers are used to obtain a $\mathcal{O}(k)$-invariant function value $v_{\mathrm{out}}$. Figure 2 shows a conceptual diagram.

**Input Layer**   The input layer maps the input representative $Y$ to a point $\overline{\zeta}_Y^{\mathrm{h}} \in T_Y^{\mathrm{h}} \mathrm{St}(k, D)$. A horizontal projection $\mathsf{HorP} : \mathrm{St}(k, D) \to T_Y^{\mathrm{h}} \mathrm{St}(k, D) \cong \mathbb{R}^{(D-k) \times k}$ is constructed as an input layer:

$$\mathsf{HorP}\left(\boldsymbol{Y}\right) = \boldsymbol{W} - \boldsymbol{Y} \boldsymbol{Y}^\top \boldsymbol{W}, \tag{6}$$

where $\boldsymbol{W}$ is the output of NN obtained with $\boldsymbol{Y} \boldsymbol{Y}^\top$ as input. HorP is well-defined because it is uniquely determined regardless of how the representative $\boldsymbol{Y} \in [\boldsymbol{Y}]$ is chosen.

**Intermediate Layer**   To model the dynamics over the horizontal space, intermediate layers based on neural networks are constructed. The layers are particularly important for the ability to represent the distribution, and various configurations are possible such as concatsquash (CS) layer (Grathwohl et al. (2019)). The layer can be stacked multiply, and the overall power of expression increases with the number of layers. The input $x$ of the first intermediate layer is $x = \mathsf{Vec} \circ \mathsf{HorP}$. The last intermediate layer (one layer before the output layer) must be constructed such that it exhibits a potential function value. This is to obtain the gradient for the potential function in the output layer.

**Output Layer**   We construct a gradient $\mathsf{Grad} : (\mathbb{R}, \mathrm{St}(k, D)) \to T_Y^{\mathrm{h}} \mathrm{St}(k, D)$ as an output layer. In Grad, $\overline{\xi}_Y^{\mathrm{h}} \in T_Y^{\mathrm{h}} \mathrm{St}(k, D)$ is obtained by taking the gradient of the potential function $v_{\mathrm{out}}$:

$$\mathsf{Grad}\left(v_{\mathrm{out}}, \boldsymbol{Y}\right) = P_{\boldsymbol{Y}}^{\mathrm{t}}\left(\mathsf{autograd}\left(v_{\mathrm{out}}, \boldsymbol{Y}\right)\right) \text{ s.t. } P_{\boldsymbol{Y}}^{\mathrm{t}}\left(\boldsymbol{Z}\right) = \boldsymbol{Z} - \boldsymbol{Y} \operatorname{sym}\left(\boldsymbol{Y}^\top \boldsymbol{Z}\right), \tag{7}$$

where autograd $(v_{\text{out}}, \boldsymbol{Y})$ denotes automatic differentiation (Paszke et al. (2017)) $\frac{\partial v_{\text{out}}}{\partial \boldsymbol{Y}}$, $P_{\boldsymbol{Y}}^{\text{t}}(\boldsymbol{Z})$ denotes the projection of $\boldsymbol{Z}$ onto $T_{\boldsymbol{Y}} \operatorname{St}(k, D)$ and $\operatorname{sym}(\boldsymbol{B}) = (\boldsymbol{B} + \boldsymbol{B}^{\top})/2$ is a symmetric part of $\boldsymbol{B}$. Grad is well-defined on $\operatorname{Gr}(k, D)$ by the horizontal lift (Absil et al. (2008)).

## 5.4 ODE Solver with Orthogonal Integration

We present an ordinary differential equation (ODE) solver on $\operatorname{Gr}(k, D)$. Celledoni & Owren (2002) proposed an intrinsic ODE solver that was applicable to $\operatorname{St}(k, D)$ and did not assume an outer Euclidean space. This solver works on $\operatorname{St}(k, D)$, thus it must be reformulated into a solver suitable for $\operatorname{Gr}(k, D)$. We introduce a solver on $\operatorname{Gr}(k, D)$ via ODE operating on $T_{\boldsymbol{Y}}^{\text{h}} \operatorname{St}(k, D)$, based on results from Section 4. This is an intrinsic approach regardless of whether the manifold has been embedded in a bigger space with a corresponding extension of the vector field. The ODE defined on $T_{\boldsymbol{Y}}^{\text{h}} \operatorname{St}(k, D)$ is obtained as:

$$\frac{d\operatorname{Vec}(\epsilon(t))}{dt} = \nabla_{\gamma} \overline{R}_{\boldsymbol{Y}}^{-1} \operatorname{Vec}(\mathsf{X}_{\theta}(t, \gamma(t))) \text{ s.t. } \gamma(t) = \overline{R}_{\boldsymbol{Y}}(\epsilon(t)), \tag{8}$$

where $\epsilon : [0, \infty) \to T_{\boldsymbol{Y}}^{\text{h}} \operatorname{St}(k, D)$ is the curve on $T_{\boldsymbol{Y}}^{\text{h}} \operatorname{St}(k, D)$, $\overline{R}_{\boldsymbol{Y}}$ denotes the horizontal retraction, $\nabla_{\gamma} \overline{R}_{\boldsymbol{Y}}^{-1}$ can be calculated using (55) and Vec denotes the map of vertically concatenating matrices and converting them into a single vector. Further details on our ODE can be found in Appendix C.1.

## 5.5 Loss Function

The total change in log-likelihood using GrCNF can be calculated using the following equation.

$$\log p_{\theta}(\boldsymbol{Y}(t_1)) = \log p_{\operatorname{Gr}(k, D)}\left(F_{\mathsf{X}, t_1\theta}^{-1}(\boldsymbol{Y}(t_1))\right) - \int_{t_0}^{t_1} \operatorname{div}(\mathsf{X}_{\theta}(t, \boldsymbol{Y}(t)))dt, \tag{9}$$

where $F_{\mathsf{X}, t_1\theta}$ is a flow. In this study, we defined the loss function Loss for maximizing the log-likelihood: $\operatorname{Loss} = \operatorname{NLL} = -\log p_{\theta}(\boldsymbol{Y}(t_1))$ where NLL denotes negative log-likelihood.

# 6 Experimental Results

In this section, we discuss the results of several experiments for validating GrCNF. Further details regarding each experimental condition, such as the architectures and hyperparameters used in training, can be found in Appendix C.3.

## 6.1 Generation and Density Estimation on Artificial Textures

We first trained GrCNF on five different $\operatorname{Gr}(1, 3)$ (1-dimensional subspace in $\mathbb{R}^3$, that is, a line through the origin) data to visualize the model and the trained dynamics. The five datasets[3] were 2 spirals, a swissroll, 2 circles, 2 sines, and Target. We represented $\operatorname{Gr}(1, 3)$ with a sphere of radius 1 centered at the origin by mapping a 1-dimensional subspace to two points on the sphere (a point on the sphere and its antipodal point) for visualization.

**Result** The five types of probability densities transformed by GrCNF and the results of data generation are shown in Figure 3. The top, middle, and bottom rows in the figure present the correct data, the generated data with GrCNF when the left-most column is the prior distribution, and the probability density on a $\operatorname{Gr}(1, 3)$ obtained by training when the leftmost column was the prior distribution, respectively. Regarding the probability density, a brighter region corresponds to a higher probability. As shown in the figure, GrCNF can generate high-quality samples that are sufficiently accurate to the distribution of the correct data. To investigate whether GrCNF learns the distribution on $\operatorname{Gr}(1, 3)$, we generated antipodal points in Figure 3 with the trained GrCNF used in Figure 3. As shown in Figure 4, GrCNF generated exactly the same high quality samples as the original for the untrained antipodal points. This experimental result implies that the proposed flow accurately learns the distribution on a $\operatorname{Gr}(k, D)$ and that all the orthogonal transformed samples can be obtained with equal quality by only training with arbitrary representatives.

---

[3] The code for the data distributions is presented in Appendix C.3.1

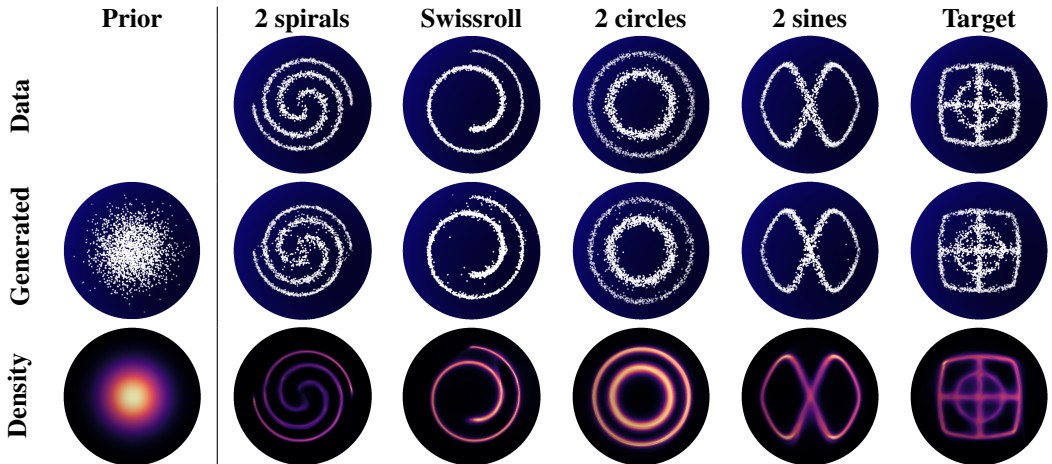

Figure 3: Generated samples and probability densities using the GrCNF trained on each of the five distributions. (top) Ground truth data, (middle) Generated data with the GrCNF when the leftmost column represents the prior distribution, and (bottom) densities obtained via training.

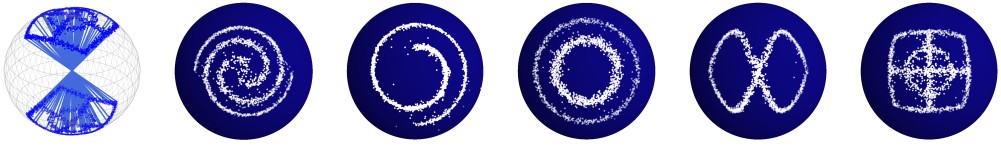

Figure 4: Results of the GrCNF transformation for the antipodal point of the prior in the middle panel of the Figure 3. The antipodal points are those that pass through the origin of the sphere and appear in the southern hemisphere (leftmost). It is evident that the antipodal points are generated with exactly the same quality as in the Figure 3 for the un-trained antipodal points.

## 6.2 Comparison with Conventional Methods for DW4 and LJ13

Comparative experiments were performed using DW4 and LJ13—the two systems presented by Köhler et al. (2020). These datasets were generated synthetically by sampling from their respective energy functions using Markov chain Monte Carlo (MCMC) methods. Both energy functions (DW4 and LJ13) are ideal for analyzing the advantages of various methods when they exist on data that are equivariant for rotation and reflection, respectively. We used the datasets generated by MCMC that were employed in Köhler et al. (2020). The orthonormalized data $Y$ were used for this study by applying Gram-Schmidt's orthogonalization to each column of $P$ such that the $D$-dimensional data was a matrix with $k$ orthonormal basis vectors (DW4: $D = 4, k = 2$; LJ13: $D = 13, k = 3$).

To match the experiment conducted in Garcia Satorras et al. (2021), $1,000$ validation and testing samples each were used for both datasets. For DW4 and LJ13, different numbers of training samples, i.e., $\{10^2, 10^3, 10^4, 10^5\}$ and $\{10, 10^2, 10^3, 10^4\}$, respectively, were selected, and their performance for each amount of data was examined. The proposed approach was compared with the state-of-the-art E($n$) equivariant flow (E-NF) presented by Garcia Satorras et al. (2021) and simple dynamics presented by Köhler et al. (2020). In addition, comparisons with graph normalizing flow (GNF), GNF with attention (GNF-att), and GNF with attention and data augmentation (GNF-att-aug) (data augmentation by rotation), which are non-equivariant variants of E-NF, were performed. All the reported values are averages of cross-validations (three runs). Otherwise, the network structures of all these conventional methods were the same as that used in Garcia Satorras et al. (2021).

**Result**  Table 1 presents the results of the cross-validated experiments (negative log-likelihood; NLL) for the test samples. The proposed GrCNF outperformed both the conventional non-equivariant models (GNF, GNF-att, and GNF-att-aug) and the conventional equivariant models (Simple dynamics, E-NF) in all data domains.

Table 1: Negative log-likelihood comparison on the test partition of different methods for DW4 and LJ13 datasets for different amount of training samples averaged over 3 runs.

| # of Samples | DW4 | | | | LJ13 | | | |
|---|---|---|---|---|---|---|---|---|
| | $10^2$ | $10^3$ | $10^4$ | $10^5$ | 10 | $10^2$ | $10^3$ | $10^4$ |
| GNF | -2.30 | -7.04 | -7.19 | -7.93 | 6.77 | -0.76 | -4.26 | -12.43 |
| GNF-att | -2.02 | -4.13 | -5.25 | -6.74 | 6.91 | 1.40 | -6.81 | -12.05 |
| GNF-att-aug | -3.11 | -4.04 | -6.51 | -9.42 | 2.95 | -6.11 | -13.94 | -15.74 |
| Simple dynamics | -1.22 | -1.28 | -1.36 | -1.39 | -1.10 | -3.87 | -3.72 | -3.59 |
| E-NF | -0.54 | -9.89 | -12.15 | -15.29 | -12.86 | -15.75 | -31.51 | -32.83 |
| GrCNF | **-12.53** | **-13.74** | **-14.09** | **-16.07** | **-23.64** | **-44.24** | **-58.02** | **-58.71** |

Table 2: Results for the QM9 Positional dataset. (Left) Negative log-likelihood, −ELBO and JSD for the QM9 Positional dataset on the test data, and (Right) normalized histogram of relative distances between atoms for QM9 Positional and generated samples with GrCNF.

| | NLL | −ELBO | JSD |
|---|---|---|---|
| Simple dynamics | 73.0 | - | 0.086 |
| Kernel dynamics | 38.6 | - | 0.029 |
| GNF | -00.9 | - | 0.011 |
| GNF-att | -26.6 | - | 0.007 |
| GNF-att-aug | -33.5 | - | 0.006 |
| E-NF | -70.2 | - | 0.006 |
| GrCNF | NLL ≤ **-85.3** | -85.3 | 0.005 |

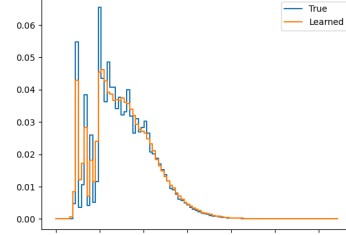

## 6.3  Comparison with Conventional Methods for QM9 Positional

A comparative experiment was performed using the QM9 Positional—a subset of the QM9 molecular dataset that considers only positional information. The purpose of this experiment was to evaluate the feasibility of generating a practical point cloud. The QM9 Positional comprises only molecules with 19 atoms/nodes, and each node has a 3-dimensional position vector associated with it. However, the likelihood of the molecules must be invariant to translation and rotation in 3-dimensional space; thus, the proposed model is suitable for this type of data. The dataset consisted of 13,831, 2501, and 1813 training, validation, and testing samples, respectively. In this experiment, we used evaluation metrics based on the NLL and Jensen-Shannon divergence (JSD; Lin (1991)), in accordance with the experiments conducted in Garcia Satorras et al. (2021). The JSD calculates the distance between the normalized histograms that are generated from the model and obtained from the training set, by creating a histogram of the relative distances among all the node pairs in each molecule.

GrCNF handles subspace data. Therefore, orthonormalized data $Y$ was used only in the proposed GrCNF, as in $PP^\top \simeq Y\Lambda Y^\top$ s.t. $\Lambda$ is diagonal (Huang et al. (2015)), such that the $k$-dimensional point cloud data $P$ of $N$ points $N \times k$ matrix is a matrix with $k$ orthonormal basis vectors ($D = 19$, $k = 3$). However, a complete point cloud generating task, such as QM9 Positional, must also store a scale parameter $\sqrt{\Lambda}$ such that $P = Y\sqrt{\Lambda}$. Therefore, GrCNF incorporates an additional architecture to estimate $\sqrt{\Lambda}$. In addition, the proposed GrCNF encounters difficulties in computing the NLL in distribution $p_\psi(P)$ of $P$; thus, a new loss function is required to address this. In this study, the evidence lower bound (ELBO), i.e., the lower bound of the log-likelihood $\log p_\psi(P)$, was maximized using a variational inference framework. This is equal to the minimization of $-\text{ELBO}(P)$. Further details regarding ELBO can be found in Appendix C.2.

We compared our GrCNF with the GNF, GNF-att, GNF-att-aug, Simple dynamics, Kernel dynamics, and E-NF methods. The network structure and experimental conditions for all these conventional methods were identical to those used in Garcia Satorras et al. (2021). In all the experiments, the training was performed for 160 epochs. The JSD values were the averages of the last 10 epochs for all the models.

**Result**  Table 2 presents the NLL and JSD cross-validated against the test data. Although the proposed GrCNF could not directly compute the NLL in this problem setting, it outperformed all

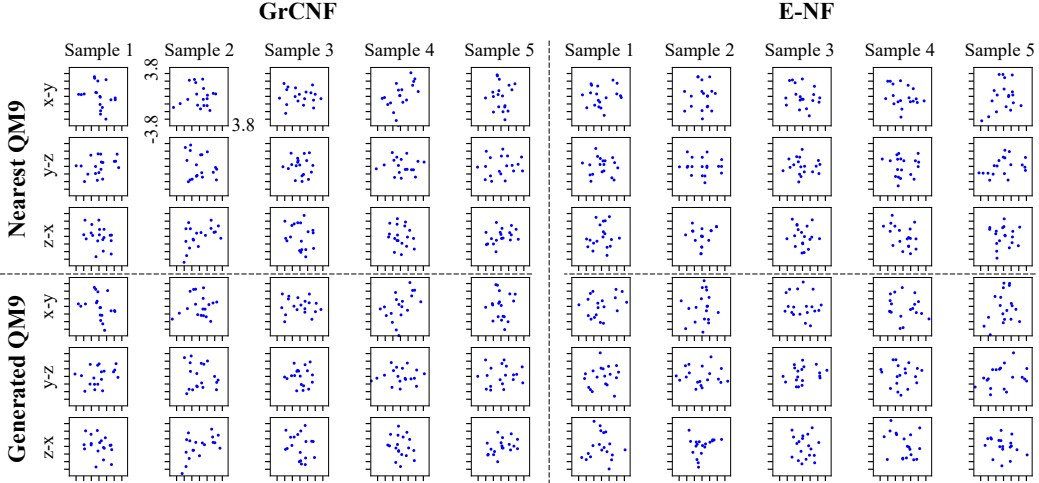

Figure 5: Visualization results of generated samples in Section 6.3. We visualize Generated QM9 by GrCNF and E-NF, as well as the aligned data points from the QM9 dataset that are closest to the Generated QM9 samples. These visualizations illustrate that GrCNF is capable of generating realistic data points that align well with the existing dataset.

the other algorithms because the relation NLL $\leq$ −ELBO is true in general. With regard to the JSD, GrCNF achieved the best performance. Figure 2 presents the normalized histograms of the QM9 Positional molecular data and the data generated by GrCNF, in relation to the JSD of GrCNF. As shown, the histograms of the molecules generated by GrCNF are close to the histograms of the data set; thus, GrCNF can generate realistic molecules stably.

## 7    Conclusion

We proposed the concept of CNF on a Grassmann manifold (GrCNF) for stable shape generation. The proposed model is a generative model capable of handling subspace data; i.e., it is a CNF that considers the equivariance on the Stiefel manifold. Through suitable experiments, the ability of the proposed GrCNF to generate qualitatively in 1-dimensional subspace datasets in $\mathbb{R}^3$ was confirmed. Further, GrCNF significantly outperformed existing normalizing flows methods in terms of the log-likelihood or ELBO for DW4, LJ13, and QM9 Positional. It was shown that GrCNF can be used to generate realistic data more stably.

**Societal Impact and Limitations**    Our work is concerned with accurately modeling the data topology; thus, we do not expect there to be any negative consequence in our application field. The proposed theory and implementation are valid in a remarkably general setting, although there are limitations that can be addressed in future works: **1.** The proposed method requires calculation of the Kronecker product at each ODE step (Section 5.4), which is computationally expensive. **2.** Our current paper does not include experiments on high-dimensional data. Therefore, future experiments should be conducted using higher dimensional data, such as point clouds that represent general object geometry. However, the results presented in this paper still provide numerous valuable insights in the field of stable shape generation.

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

## A  Appendix Overview

First, we present the details of the various proofs of Section 4 in Appendix B. Next, in Appendix C, we describe the network layers for the building GrCNF, and the detailed model architectures, hyper-parameters, and implementation on the experiments. Finally, in Appendix D, we provide a summary of the fundamentals of a Grassmann manifold, which is the core concept of this study.

## B  Proofs

### B.1  Proposition 1

First, we invoked the following two corollaries.

**Corollary 1** (Diffeomorphism Invariance of Flows)**.** *Let* $F : \mathcal{M} \to \mathcal{N}$ *be a diffeomorphism. If* $X$ *is a smooth vector field over* $\mathcal{M}$ *and* $\theta$ *is the flow of* $X$, *then the flow of* $F_* X$[4] *is* $\eta_t = F \circ \theta_t \circ F^{-1}$, *with domain* $N_t = F(M_t)$ *for each* $t \in \mathbb{R}$.

*Proof.* See Lee (2003, Corollary 9.14). $\square$

**Corollary 2** (Homogeneity Property)**.** *The horizontal lift* $\overline{\xi}_{\boldsymbol{Y}}^{\mathrm{h}}$ *at representative* $\boldsymbol{Y} \in \mathrm{St}(k, D)$ *relative to* $\boldsymbol{\xi}_{[\boldsymbol{Y}]} \in T_{[\boldsymbol{Y}]} \mathrm{Gr}(k, D)$ *satisfies the following homogeneity (equivariance) property (4) with regard to* $^\forall \boldsymbol{Q} \in \mathcal{O}(k)$.

$$\overline{\xi}_{\boldsymbol{YQ}}^{\mathrm{h}} = \overline{\xi}_{\boldsymbol{Y}}^{\mathrm{h}} \boldsymbol{Q}.$$

*Proof.* $\pi(\boldsymbol{Y}) = \pi(\boldsymbol{YQ})$ is true for $^\forall \boldsymbol{Y} \in \mathrm{St}(k, D), \boldsymbol{Q} \in \mathcal{O}(k)$. Therefore, $\pi(\boldsymbol{Y}) = (\pi \circ q)(\boldsymbol{Y})$ is true when defined as $q(\boldsymbol{Y}) = \boldsymbol{YQ}$. When the derivative $d\pi(\cdot)[\cdot]$ of both sides is applied to the horizontal lift $\overline{\xi}_{\boldsymbol{Y}}^{\mathrm{h}}$ of $\boldsymbol{\xi}_{[\boldsymbol{Y}]}$, the following is obtained:

$$d\pi(\boldsymbol{Y})\left[\overline{\xi}_{\boldsymbol{Y}}^{\mathrm{h}}\right] = d(\pi \circ q)(\boldsymbol{Y})\left[\overline{\xi}_{\boldsymbol{Y}}^{\mathrm{h}}\right] = d\pi(q(\boldsymbol{Y}))\left[dq(\boldsymbol{Y})\left[\overline{\xi}_{\boldsymbol{Y}}^{\mathrm{h}}\right]\right] = d\pi(\boldsymbol{YQ})\left[\overline{\xi}_{\boldsymbol{Y}}^{\mathrm{h}}\boldsymbol{Q}\right]. \quad (10)$$

Moreover, from (98) which is definition of horizontal lift, the following equation is true.

$$\boldsymbol{\xi}_{[\boldsymbol{Y}]} = d\pi(\boldsymbol{Y})\left[\overline{\xi}_{\boldsymbol{Y}}^{\mathrm{h}}\right] = d\pi(\boldsymbol{YQ})\left[\overline{\xi}_{\boldsymbol{YQ}}^{\mathrm{h}}\right]. \quad (11)$$

Subsequently, we obtain the following equation.

$$\boldsymbol{\xi}_{[\boldsymbol{Y}]} = d\pi(\boldsymbol{YQ})\left[\overline{\xi}_{\boldsymbol{YQ}}^{\mathrm{h}}\right] = d\pi(\boldsymbol{YQ})\left[\overline{\xi}_{\boldsymbol{Y}}^{\mathrm{h}}\boldsymbol{Q}\right]. \quad (12)$$

Finally, the uniqueness of the horizontal lift yields $\overline{\xi}_{\boldsymbol{YQ}}^{\mathrm{h}} = \overline{\xi}_{\boldsymbol{Y}}^{\mathrm{h}}\boldsymbol{Q}$. $\square$

**Proposition 1.** *Let* $\mathrm{Gr}(k, D)$ *be a Grassmann manifold,* $\mathsf{X}$ *be any time-dependent vector field on* $\mathrm{Gr}(k, D)$, *and* $F_{\mathsf{X},T}$ *be a flow on a* $\mathsf{X}$. *Let* $\overline{\mathsf{X}}$ *be any time-dependent horizontal lift and* $\overline{F}_{\overline{\mathsf{X}},T}$ *be a flow of* $\overline{\mathsf{X}}$. $\overline{\mathsf{X}}$ *is a vector field on* $\mathrm{St}(k, D)$ *if and only if* $\overline{F}_{\overline{\mathsf{X}},T}$ *is a flow on* $\mathrm{St}(k, D)$ *and satisfies invariance condition* $\overline{\mathsf{X}} \sim \overline{\mathsf{X}}'$ *for all* $\overline{F}_{\overline{\mathsf{X}},T} \sim \overline{F}_{\overline{\mathsf{X}}',T}$. *Therefore,* $\mathsf{X}$ *is a vector field on* $\mathrm{Gr}(k, D)$ *if and only if* $F_{\mathsf{X},T} := \left[\overline{F}_{\overline{\mathsf{X}},T}\right]$ *is a flow on* $\mathrm{Gr}(k, D)$, *and vice versa.*

*Proof.* **Flow** $F_{\mathsf{X},T}$ **on Gr**$(k, D) \Rightarrow$ **Vector Field** $\mathsf{X}$ **on Gr**$(k, D)$**.** Let $\theta : \mathrm{St}(k, D) \times \mathcal{O}(k) \to \mathrm{St}(k, D), (\boldsymbol{Y}, \boldsymbol{Q}) \mapsto \boldsymbol{YQ}$ be a map representing the right action of the orthogonal group. In addition, let $F_{\mathsf{X},T}$ be a flow on $\mathrm{Gr}(k, D)$ and $\overline{F}_{\overline{\mathsf{X}},T}$ be a flow on $\mathrm{St}(k, D)$. These satisfy $\overline{F}_{\overline{\mathsf{X}}\boldsymbol{Q},T} \sim$

---
[4] $F_*$ denotes the pushforward, that is, another notation for the differential of $F$.

$\overline{F}_{\overline{\mathsf{X}},T}, \overline{F}_{\overline{\mathsf{X}}\boldsymbol{Q},T} \in F_{\mathsf{X},T}, \overline{F}_{\overline{\mathsf{X}},T} \in F_{\mathsf{X},T}$.

$$\overline{\mathsf{X}}\left(t, \overline{F}_{\overline{\mathsf{X}}\boldsymbol{Q},t}\left(\boldsymbol{Y}\boldsymbol{Q}\right)\right) = \overline{\mathsf{X}}\left(t, \overline{F}_{\overline{\mathsf{X}},t}\left(\boldsymbol{Y}\right)\boldsymbol{Q}\right) \tag{13}$$

$$= \frac{d}{dt}\left\{\overline{F}_{\overline{\mathsf{X}},t}\left(\boldsymbol{Y}\right)\boldsymbol{Q}\right\} \tag{14}$$

$$= \frac{d}{dt}\left(\theta \circ \overline{F}_{\overline{\mathsf{X}},t}\right)\left(\boldsymbol{Y}\right) \tag{15}$$

$$= d(\theta)_{\boldsymbol{Y}}\left\{\frac{d}{dt}\overline{F}_{\overline{\mathsf{X}},t}\left(\boldsymbol{Y}\right)\right\} \tag{16}$$

$$= d(\theta)_{\boldsymbol{Y}}\left\{\overline{\mathsf{X}}\left(t, \overline{F}_{\overline{\mathsf{X}},t}\left(\boldsymbol{Y}\right)\right)\right\} \tag{17}$$

$$= \overline{\mathsf{X}}\left(t, \overline{F}_{\overline{\mathsf{X}},t}\left(\boldsymbol{Y}\right)\right)\boldsymbol{Q}. \tag{18}$$

Thus, $\overline{\mathsf{X}} \sim \overline{\mathsf{X}}\boldsymbol{Q}$ is true. Therefore, $\overline{\mathsf{X}}$ is the horizontal lift of the vector field $\mathsf{X}$ on $\mathrm{Gr}(k, D)$ and is unique for $\mathsf{X}$.

**Flow $F_{\mathsf{X},T}$ on $\mathbf{Gr(k, D)}$ $\Leftarrow$ Vector Field $\mathsf{X}$ on $\mathbf{Gr(k, D)}$.** Let $\theta : \mathrm{St}(k, D) \times \mathcal{O}(k) \to \mathrm{St}(k, D), (\boldsymbol{Y}, \boldsymbol{Q}) \mapsto \boldsymbol{Y}\boldsymbol{Q}$ be a map representing the right action of the orthogonal group. In addition, let $\overline{\mathsf{X}}$ be a vector field over a horizontal bundle $T^{\mathrm{h}}\mathrm{St}(k, D)$ on $\mathrm{St}(k, D)$ and $\overline{F}_{\overline{\mathsf{X}},T}$ be its flow. From the Corollary 1,

$$\overline{F}_{\theta_* \circ \overline{\mathsf{X}},T} = \theta \circ \overline{F}_{\overline{\mathsf{X}},T} \circ \theta^{-1} \tag{19}$$

$$\overline{F}_{\theta_* \circ \overline{\mathsf{X}},T} \circ \theta = \theta \circ \overline{F}_{\overline{\mathsf{X}},T} \tag{20}$$

$$\overline{F}_{d(\theta)_{\boldsymbol{Y}}\overline{\mathsf{X}},T} \circ \theta = \theta \circ \overline{F}_{\overline{\mathsf{X}},T} \tag{21}$$

$$\overline{F}_{\overline{\mathsf{X}}\boldsymbol{Q},T}\left(\boldsymbol{Y}\boldsymbol{Q}\right) = \overline{F}_{\overline{\mathsf{X}},T}\left(\boldsymbol{Y}\right)\boldsymbol{Q}. \tag{22}$$

Note that $d(\theta)_{\boldsymbol{Y}}\overline{\mathsf{X}} = \overline{\mathsf{X}}\boldsymbol{Q}$ is derived from the Corollary 2 and (4) in Zhu & Sato (2021). This indicates that $\overline{F}_{\overline{\mathsf{X}},T} \sim \overline{F}_{\overline{\mathsf{X}}',T}$ is true for any $\overline{\mathsf{X}}, \overline{\mathsf{X}}' \in T^{\mathrm{h}}\mathrm{St}(k, D)$ that satisfies $\overline{\mathsf{X}} \sim \overline{\mathsf{X}}'$. Thus, a new flow can be defined as $F_{\mathsf{X},T} := \left[\overline{F}_{\overline{\mathsf{X}},T}\right]$. This is a flow on a $\mathrm{Gr}(k, D)$. Because $\overline{\mathsf{X}}$ is a vector field in a horizontal bundle $T^{\mathrm{h}}\mathrm{St}(k, D)$ on $\mathrm{St}(k, D)$, it is a horizontal lift of the vector field $\mathsf{X}$ on $\mathrm{Gr}(k, D)$ and is therefore unique for $\mathsf{X}$.

Thus, the proof is complete. $\qquad\square$

## B.2 Proposition 2

**Proposition 2.** *Let $\mathrm{Gr}(k, D)$ be a Grassmann manifold. Let $p$ be the probability density on $\mathrm{Gr}(k, D)$ and $F$ be the flow on $\mathrm{Gr}(k, D)$. Suppose $\overline{p}$ is a density on $\mathrm{St}(k, D)$ and $\overline{F}$ is a flow on $\mathrm{St}(k, D)$. Then, the distribution $\overline{p}_{\overline{F}}$ after transformations by $\overline{F}$ is also a density on $\mathrm{St}(k, D)$. Further, the invariance condition $\overline{p}_{\overline{F}} \sim \overline{p}_{\overline{F}'}$ is satisfied for all $\overline{F} \sim \overline{F}'$. Therefore, $p_F := [\overline{p}_{\overline{F}}]$ is a distribution on $\mathrm{Gr}(k, D)$.*

*Proof.* Let $\theta : \mathrm{St}(k, D) \times \mathcal{O}(k) \to \mathrm{St}(k, D) : (\boldsymbol{Y}, \boldsymbol{Q}) \mapsto \boldsymbol{Y}\boldsymbol{Q}$ be a map representing the right action of the orthogonal group.

$$\bar{p}_F \left( \theta \circ \boldsymbol{Y} \right) = \bar{p}_F \left( \theta \circ \boldsymbol{Y} \right) \frac{\left| \det \left\{ J_\theta \left( \boldsymbol{Y} \right) \right\} \right|}{\left| \det \left\{ J_\theta \left( \boldsymbol{Y} \right) \right\} \right|} = \frac{\bar{p}_{\theta^{-1} \circ F} \left( \boldsymbol{Y} \right)}{\left| \det \left\{ J_\theta \left( \boldsymbol{Y} \right) \right\} \right|} \tag{23}$$

$$= \bar{p} \left( \left( F^{-1} \circ \theta \right) \left( \boldsymbol{Y} \right) \right) \frac{\left| \det \left\{ J_{F^{-1} \circ \theta} \left( \boldsymbol{Y} \right) \right\} \right|}{\left| \det \left\{ J_\theta \left( \boldsymbol{Y} \right) \right\} \right|} \tag{24}$$

$$= \left( \bar{p} \circ F^{-1} \right) \circ \theta \left( \boldsymbol{Y} \right) \frac{\left| \det \left\{ J_{\theta \circ F^{-1}} \left( \boldsymbol{Y} \right) \right\} \right|}{\left| \det \left\{ J_\theta \left( \boldsymbol{Y} \right) \right\} \right|} \tag{25}$$

$$= \theta \circ \left( \bar{p} \circ F^{-1} \right) \left( \boldsymbol{Y} \right) \frac{\left| \det \left\{ J_\theta \left( F^{-1} \left( \boldsymbol{Y} \right) \right) J_{F^{-1}} \left( \boldsymbol{Y} \right) \right\} \right|}{\left| \det \left\{ J_\theta \left( \boldsymbol{Y} \right) \right\} \right|} \tag{26}$$

$$= \theta \circ \left( \bar{p} \circ F^{-1} \right) \left( \boldsymbol{Y} \right) \frac{\left| \det \left\{ J_\theta \left( F^{-1} \left( \boldsymbol{Y} \right) \right) \right\} \right| \left| \det \left\{ J_{F^{-1}} \left( \boldsymbol{Y} \right) \right\} \right|}{\left| \det \left\{ J_\theta \left( \boldsymbol{Y} \right) \right\} \right|} \tag{27}$$

$$= \theta \circ \bar{p} \left( F^{-1} \right) \left( \boldsymbol{Y} \right) \left| \det \left\{ J_{F^{-1}} \left( \boldsymbol{Y} \right) \right\} \right| \frac{\left| \det \left\{ J_\theta \left( F^{-1} \left( \boldsymbol{Y} \right) \right) \right\} \right|}{\left| \det \left\{ J_\theta \left( \boldsymbol{Y} \right) \right\} \right|} \tag{28}$$

$$= \theta \circ \bar{p}_F \left( \boldsymbol{Y} \right) \frac{\left| \det \left\{ J_\theta \left( F^{-1} \left( \boldsymbol{Y} \right) \right) \right\} \right|}{\left| \det \left\{ J_\theta \left( \boldsymbol{Y} \right) \right\} \right|} \tag{29}$$

$$= \theta \circ \bar{p}_F \left( \boldsymbol{Y} \right), \tag{30}$$

where $\left| \det \left\{ J_\theta \left( \boldsymbol{X} \right) \right\} \right| = 1$ is true because $\theta$ is the action of the orthogonal group. Therefore, as $\bar{p}_F \left( \theta \circ \boldsymbol{Y} \right) = \theta \circ \bar{p}_F \left( \boldsymbol{Y} \right)$ is true, $\bar{p}_F \sim \bar{p}_F \boldsymbol{Q} \in p_F$ is true. Based on this, it can be concluded that the subject is satisfied. □

### B.3 Proposition 3

**Proposition 3.** *The distribution $p_{\mathrm{Gr}(k, D)}$ on a Grassmann manifold $\mathrm{Gr}(k, D)$ based on the matrix-variate Gaussian distribution $\mathcal{MN}$ can be expressed as follows.*

$$p_{\mathrm{Gr}(k,D)} \left( [\boldsymbol{X}]; [\boldsymbol{M}], \boldsymbol{U}, \boldsymbol{V} \right) = V_{\mathrm{Gr}(k,D)} \mathcal{MN} \left( \bar{\boldsymbol{\xi}}_{\boldsymbol{M}}^{\mathrm{h}}; \boldsymbol{0}, \boldsymbol{U}, \boldsymbol{V} \right) \left| \det \left( \nabla_{\bar{\boldsymbol{\xi}}_{\boldsymbol{M}}^{\mathrm{h}}} \overline{R}_{\boldsymbol{M}} \right) \right|, \tag{31}$$

*where $\boldsymbol{M}$ is an orthonormal basis matrix denoting the mean of the distribution, $\boldsymbol{U}$ is a positive definite matrix denoting the row directional variance, $\boldsymbol{V}$ is a positive definite matrix denoting the column directional variance, and $\bar{\boldsymbol{\xi}}_{\boldsymbol{M}}^{\mathrm{h}}$ is a random sample from $\mathcal{MN}$ in an $(D - k) \times k$-dimensional horizontal space $T_{\boldsymbol{M}}^{\mathrm{h}} \mathrm{St}(k, D)$. $V_{\mathrm{Gr}(k,D)}$ denotes the total volume of $\mathrm{Gr}(k, D)$ defined by (113), $\overline{R}_{\boldsymbol{M}}$ denotes the horizontal retraction at $\boldsymbol{M}$, and $\left| \det \left( \nabla_{\bar{\boldsymbol{\xi}}_{\boldsymbol{M}}^{\mathrm{h}}} \overline{R}_{\boldsymbol{M}} \right) \right|$ denotes the Jacobian.*

*Proof.* Let $p_{\mathrm{Gr}} \left( [\boldsymbol{X}] \right)$ be a probability density function on a $\mathrm{Gr}(k, D)$. From (101), let $(d\boldsymbol{X})$ be the invariant measure on $\mathrm{Gr}(k, D)$ and $d\bar{\boldsymbol{\xi}}_{\boldsymbol{M}}^{\mathrm{h}}$ be the Lebesgue measure on $T_{\boldsymbol{M}}^{\mathrm{h}} \mathrm{St}(k, D)$. Subsequently, a change of variables was performed according to the following:

$$p_{\mathrm{Gr}(k,D)} \left( [\boldsymbol{X}] \right) (d\boldsymbol{X}) = p_{\mathrm{Gr}(k,D)} \left( \left[ \overline{R}_{\boldsymbol{M}} \left( \bar{\boldsymbol{\xi}}_{\boldsymbol{M}}^{\mathrm{h}} \right) \right] \right) d\bar{\boldsymbol{\xi}}_{\boldsymbol{M}}^{\mathrm{h}} \tag{32}$$

$$p_{\mathrm{Gr}(k,D)} \left( [\boldsymbol{X}] \right) = p_{\mathrm{Gr}(k,D)} \left( \left[ \overline{R}_{\boldsymbol{M}} \left( \bar{\boldsymbol{\xi}}_{\boldsymbol{M}}^{\mathrm{h}} \right) \right] \right) \left| \det \left( \frac{d\bar{\boldsymbol{\xi}}_{\boldsymbol{M}}^{\mathrm{h}}}{d\overline{R}_{\boldsymbol{M}}} \right) \right| \tag{33}$$

$$p_{\mathrm{Gr}(k,D)} \left( [\boldsymbol{X}] \right) = p_{\mathrm{Gr}(k,D)} \left( \left[ \overline{R}_{\boldsymbol{M}} \left( \bar{\boldsymbol{\xi}}_{\boldsymbol{M}}^{\mathrm{h}} \right) \right] \right) \left| \det \left( \nabla_{\bar{\boldsymbol{\xi}}_{\boldsymbol{M}}^{\mathrm{h}}} \overline{R}_{\boldsymbol{M}} \right) \right|^{-1}. \tag{34}$$

Suppose $p_{\mathrm{Gr}}\left([\boldsymbol{X}]\right)$ is integrable with the probability measure $[d\boldsymbol{X}]$ on $\mathrm{Gr}(k, D)$ defined by (114). Then, we obtain the following relation.

$$\int_{\mathrm{Gr}(k,D)} p_{\mathrm{Gr}(k,D)}\left([\boldsymbol{X}]\right)[d\boldsymbol{X}] \tag{35}$$

$$= \frac{1}{V_{\mathrm{Gr}(k,D)}} \int_{\mathrm{Gr}(k,D)} p_{\mathrm{Gr}(k,D)}\left([\boldsymbol{X}]\right)(d\boldsymbol{X}) \tag{36}$$

$$= \frac{1}{V_{\mathrm{Gr}(k,D)}} \int_{T_M^{\mathrm{h}} \mathrm{St}(k,D)} p_{\mathrm{Gr}(k,D)}\left(\left[\overline{R}_M\left(\overline{\boldsymbol{\xi}}_M^{\mathrm{h}}\right)\right]\right)\left|\det\left(\nabla_{\overline{\boldsymbol{\xi}}_M^{\mathrm{h}}} \overline{R}_M\right)\right|^{-1} d\overline{\boldsymbol{\xi}}_M^{\mathrm{h}}. \tag{37}$$

In addition, we obtain the following equation based on $[d\boldsymbol{X}]$.

$$\int_{\mathrm{Gr}(k,D)} p_{\mathrm{Gr}(k,D)}\left([\boldsymbol{X}]\right)[d\boldsymbol{X}] = \int_{T_M^{\mathrm{h}} \mathrm{St}(k,D)} \mathcal{MN}\left(\overline{\boldsymbol{\xi}}_M^{\mathrm{h}}\right) d\overline{\boldsymbol{\xi}}_M^{\mathrm{h}} = 1, \tag{38}$$

where $\mathcal{MN}\left(\overline{\boldsymbol{\xi}}_M^{\mathrm{h}}\right)$ denotes the matrix-variate Gaussian distribution (Mathai et al. (2022)). Thus, the probability density function on $\mathrm{Gr}(k, D)$ can be expressed as follows:

$$p_{\mathrm{Gr}(k,D)}\left(\left[\overline{R}_M\left(\overline{\boldsymbol{\xi}}_M^{\mathrm{h}}\right)\right]\right) = V_{\mathrm{Gr}(k,D)}\mathcal{MN}\left(\overline{\boldsymbol{\xi}}_M^{\mathrm{h}}\right)\left|\det\left(\nabla_{\overline{\boldsymbol{\xi}}_M^{\mathrm{h}}} \overline{R}_M\right)\right|. \tag{39}$$

The Jacobian can be represented as $\left|\det\left(\nabla_{\overline{\boldsymbol{\xi}}_M^{\mathrm{h}}} \overline{R}_M\right)\right| = \left|\det\left\{\left(\partial\overline{R}_M\right)^{\top}\left(\partial\overline{R}_M\right)\right\}\right|^{\frac{1}{2}}$ from Evans & Ronald (2015), where $\partial\overline{R}_M = \frac{\partial\overline{R}_M}{\partial\overline{\boldsymbol{\xi}}_M^{\mathrm{h}}}$. Further, $\partial\overline{R}_M$ can be computed as follows. First, we define the horizontal retraction $\overline{R}_Y : T_Y^{\mathrm{h}} \mathrm{St}(k, D) \to \mathrm{St}(k, D)$ based on the Cayley transform from Zhu & Sato (2021).

$$\boldsymbol{X} = \overline{R}_M\left(\overline{\boldsymbol{\xi}}_M^{\mathrm{h}}\right) = \boldsymbol{M} + \overline{\boldsymbol{\xi}}_M^{\mathrm{h}} - \left(\frac{1}{2}\boldsymbol{M} + \frac{1}{4}\overline{\boldsymbol{\xi}}_M^{\mathrm{h}}\right)\left(\boldsymbol{I}_k + \frac{1}{4}\overline{\boldsymbol{\xi}}_M^{\mathrm{h}\top}\overline{\boldsymbol{\xi}}_M^{\mathrm{h}}\right)^{-1}\overline{\boldsymbol{\xi}}_M^{\mathrm{h}\top}\overline{\boldsymbol{\xi}}_M^{\mathrm{h}}. \tag{40}$$

This is a fixed time ($t = 1$) version of (122). Next, for improved visibility in subsequent calculations, let $\boldsymbol{E} = \overline{\boldsymbol{\xi}}_M^{\mathrm{h}}$, $\boldsymbol{F} = \frac{1}{2}\boldsymbol{M} + \frac{1}{4}\boldsymbol{E}$, $\boldsymbol{G} = \left(\boldsymbol{I}_k + \frac{1}{4}\boldsymbol{H}\right)^{-1}$, $\boldsymbol{H} = \boldsymbol{E}^{\top}\boldsymbol{E}$. In addition, let D be defined as the operator for the derivative of a matrix by a matrix. Then, the derivative $\nabla_{\overline{\boldsymbol{\xi}}_M^{\mathrm{h}}} \overline{R}_M = \mathsf{D}\boldsymbol{X}$ by $\boldsymbol{E}$ is as follows:

$$\nabla_{\overline{\boldsymbol{\xi}}_M^{\mathrm{h}}} \overline{R}_M = \mathsf{D}\boldsymbol{X} = \mathsf{D}\boldsymbol{M} + \mathsf{D}\boldsymbol{E} - \mathsf{D}\left(\boldsymbol{F}\boldsymbol{G}\boldsymbol{H}\right). \tag{41}$$

Finally, each derivative can be calculated as follows:

$$\mathsf{D}\boldsymbol{M} = \boldsymbol{0}, \tag{42}$$

$$\mathsf{D}\boldsymbol{E} = \boldsymbol{I}_{Dk}, \tag{43}$$

$$\mathsf{D}\left(\boldsymbol{F}\boldsymbol{G}\boldsymbol{H}\right) = \mathsf{D}\left(\boldsymbol{F}\left(\boldsymbol{G}\boldsymbol{H}\right)\right) \tag{44}$$

$$= \left\{\left(\boldsymbol{G}\boldsymbol{H}\right)^{\top} \otimes \boldsymbol{I}_D\right\}\mathsf{D}\boldsymbol{F} + \left(\boldsymbol{I}_k \otimes \boldsymbol{F}\right)\mathsf{D}\left(\boldsymbol{G}\boldsymbol{H}\right) \tag{45}$$

$$= \left\{\left(\boldsymbol{G}\boldsymbol{H}\right)^{\top} \otimes \boldsymbol{I}_D\right\}\mathsf{D}\boldsymbol{F} + \left(\boldsymbol{I}_k \otimes \boldsymbol{F}\right)\left\{\left(\boldsymbol{H}^{\top} \otimes \boldsymbol{I}_k\right)\mathsf{D}\boldsymbol{G} + \left(\boldsymbol{I}_k \otimes \boldsymbol{G}\right)\mathsf{D}\boldsymbol{H}\right\} \tag{46}$$

$$= \left\{\left(\boldsymbol{G}\boldsymbol{H}\right)^{\top} \otimes \boldsymbol{I}_D\right\}\mathsf{D}\boldsymbol{F}$$
$$+ \left(\boldsymbol{I}_k \otimes \boldsymbol{F}\right)\left(\boldsymbol{H}^{\top} \otimes \boldsymbol{I}_k\right)\mathsf{D}\boldsymbol{G} + \left(\boldsymbol{I}_k \otimes \boldsymbol{F}\right)\left(\boldsymbol{I}_k \otimes \boldsymbol{G}\right)\mathsf{D}\boldsymbol{H} \tag{47}$$

$$= \left(\boldsymbol{G}^{\top}\boldsymbol{H}^{\top} \otimes \boldsymbol{I}_D\right)\mathsf{D}\boldsymbol{F} + \left(\boldsymbol{H}^{\top} \otimes \boldsymbol{F}\right)\mathsf{D}\boldsymbol{G} + \left(\boldsymbol{I}_k \otimes \boldsymbol{F}\boldsymbol{G}\right)\mathsf{D}\boldsymbol{H}, \tag{48}$$

$$\mathsf{D}\boldsymbol{F} = \mathsf{D}\left(\frac{1}{2}\boldsymbol{M}\right) + \mathsf{D}\left(\frac{1}{4}\boldsymbol{E}\right) = \frac{1}{4}\mathsf{D}\boldsymbol{E}, \tag{49}$$

$$\mathsf{D}\boldsymbol{G} = -\left(\boldsymbol{G}^{\top} \otimes \boldsymbol{G}\right)\mathsf{D}\boldsymbol{H}, \tag{50}$$

$$\mathsf{D}\boldsymbol{H} = \left(\boldsymbol{I}_{k^2} + \boldsymbol{K}_{k,k}\right)\left(\boldsymbol{I}_k \otimes \boldsymbol{E}^{\top}\right)\mathsf{D}\boldsymbol{E}, \tag{51}$$

where $\otimes$ denotes the Kronecker product. $\boldsymbol{K}_{D,k}$ is a $Dk \times Dk$ matrix $\boldsymbol{K}_{D,k} = \sum_{i=1}^{m}\sum_{j=1}^{n}\left(\boldsymbol{L}_{i,j} \otimes \boldsymbol{L}_{i,j}^{\top}\right)$ referred to as the commutation matrix, which denotes the transposition operation of $D \times k$. Further, $\boldsymbol{L}_{i,j}$ is a $D \times k$ matrix whose $(i, j)$ component is 1 whereas all other components are 0. $\qquad\square$

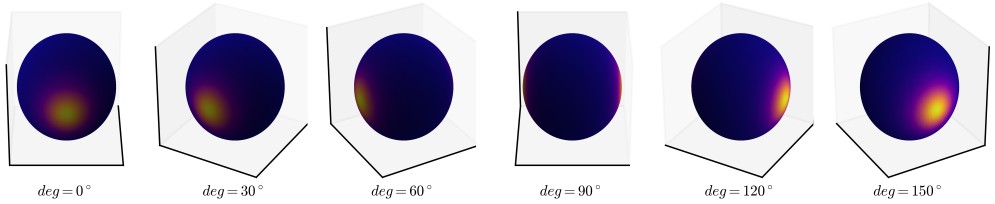

Figure 6: $p_{\mathrm{Gr}(1,3)}\left([\boldsymbol{X}]\right)$ with $\boldsymbol{M} = (1.0, 0.0, 0.0)^{\top}, \boldsymbol{U} = \sigma^2 \boldsymbol{I}_3, \boldsymbol{V} = \boldsymbol{I}_1, \sigma^2 = 0.5$. Each sphere in the figure indicates $\mathrm{Gr}(1,3)$, with brighter spheres representing higher densities.

For details on the formulae for matrix derivatives used in this proof, please refer to Magnus & Neudecker (2019).

$p_{\mathrm{Gr}(k,D)}\left([\boldsymbol{X}]\right) = p_{\mathrm{Gr}(k,D)}\left([\boldsymbol{X}]; [\boldsymbol{M}], \boldsymbol{U}, \boldsymbol{V}\right)$ is a probability distribution following mean $[\boldsymbol{M}]$ and matrix variance $\boldsymbol{U}, \boldsymbol{V}$. Using $\mathrm{Gr}(1,3)$ as an example, we qualitatively confirmed through visualization that $p_{\mathrm{Gr}(1,3)}\left([\boldsymbol{X}]\right)$ is a density on $\mathrm{Gr}(1,3)$. $\mathrm{Gr}(1,3)$ is a 1-dimensional subspace in a 3-dimensional space; that is, a space whose elements are lines passing through the origin in 3-dimensional space. For the visualization, we expressed $\mathrm{Gr}(1,3)$ by mapping a 1-dimensional subspace to two points on a sphere (one point on the sphere and its antipodal point) of radius 1 centered at the origin.

Figure 6 shows the density of $p_{\mathrm{Gr}(1,3)}\left([\boldsymbol{X}]\right)$ with $\boldsymbol{M} = (1.0, 0.0, 0.0)^{\top}, \boldsymbol{U} = \sigma^2 \boldsymbol{I}_3, \boldsymbol{V} = \boldsymbol{I}_1, \sigma^2 = 0.5$. Each sphere in the figure indicates $\mathrm{Gr}(1,3)$, with brighter spheres representing higher densities. The leftmost figure shows $\boldsymbol{M}$ as viewed from the front diagonally above, and the other figures present the views when the viewpoint is rotated clockwise around the $z$-axis by $30°$ to $150°$ with movement to the right. In the leftmost figure, the density is highly spread around $\boldsymbol{M}$. In the other figures (particularly the rightmost one), the antipodal point $(-\boldsymbol{M} = (-1.0, 0.0, 0.0)^{\top})$ is densely spread out. This implies that when only one $\boldsymbol{M}$ is specified as the representative of the equivalence class $[\boldsymbol{M}]$, the density around the other elements in the equivalence class $[\boldsymbol{M}]$ is as high as that around the representative. Thus, we can confirm that $p_{\mathrm{Gr}(k,D)}\left([\boldsymbol{X}]\right)$ has a density of $\mathrm{Gr}(1,3)$.

## C  Experimental Details for Learning GrCNF

### C.1  Details on ODE Solver with Orthogonal Integration

We explain in detail an ordinary differential equation (ODE) solver on $\mathrm{Gr}(k, D)$. Several studies on ODE solvers employed on a manifold $\mathcal{M}$ have been reported (Munthe-Kaas (1999); Iserles et al. (2000); Hairer (2006)). Hairer (2006) proposed a simple projection method that projected onto the manifold at each step and the symmetric projection method suitable for long-time integration. However, this method requires $\mathcal{M}$ to be a submanifold in Euclidean space, and thus cannot be applied on $\mathrm{Gr}(k, D)$. In contrast, Celledoni & Owren (2002) proposed an intrinsic ODE solver that was applicable to a Stiefel manifold and did not assume an outer Euclidean space. This solver works on a Stiefel manifold, thus it must be reformulated into a solver suitable for $\mathrm{Gr}(k, D)$, which is our problem setting. We introduce below a solver on $\mathrm{Gr}(k, D)$ via ODE operating on the horizontal space $T_{\boldsymbol{Y}}^{\mathrm{h}} \mathrm{St}(k, D)$, based on results from Celledoni & Owren (2002) and Section 4. This is an intrinsic approach regardless of whether $\mathcal{M}$ has been embedded in a larger space with a corresponding extension of the vector field.

First, consider an ODE expressed using a vector field $\mathsf{X}_\theta$ on a curve $\gamma(t) : [0, \infty) \to \mathrm{St}(k, D)$ such that for each time $t$.

$$\frac{d\gamma(t)}{dt} = \mathsf{X}_\theta(t, \gamma(t)), \quad \gamma(0) = \boldsymbol{Y}, \tag{52}$$

where $\mathsf{X}_\theta$ is constructed by a neural network with parameter $\theta$, as described in Section 5.3. Horizontal retraction $\overline{R}_{\boldsymbol{Y}}$ which is defined as (122) and is described in Appendix D.6, defines local coordinates of $\mathrm{Gr}(k, D)$ in a neighborhood of the point $[\boldsymbol{Y}]$. Thus, the solution of ODE can be expressed as:

$$\gamma(t) = \overline{R}_{\boldsymbol{Y}}(\epsilon(t)), \tag{53}$$

where $\epsilon : [0, \infty) \to T_{\boldsymbol{Y}}^{\mathrm{h}} \mathrm{St}(k, D)$ is the curve on $T_{\boldsymbol{Y}}^{\mathrm{h}} \mathrm{St}(k, D)$. By differentiating (53) with $t$, the following equation is obtained:

$$\frac{d\gamma(t)}{dt} = \frac{d}{dt}\overline{R}_{\boldsymbol{Y}}(\epsilon(t)) = \mathsf{X}_\theta(t, \gamma(t)). \tag{54}$$

Therefore, the ODE defined on $T_{\boldsymbol{Y}}^{\mathrm{h}} \mathrm{St}(k, D)$ is obtained as (8):

$$\frac{d\mathsf{Vec}(\epsilon(t))}{dt} = \left(\nabla_\epsilon \overline{R}_{\boldsymbol{Y}}\right)^{-1} \mathsf{Vec}\left(\mathsf{X}_\theta\left(t, \overline{R}_{\boldsymbol{Y}}(\epsilon(t))\right)\right)$$
$$= \nabla_\gamma \overline{R}_{\boldsymbol{Y}}^{-1} \mathsf{Vec}\left(\mathsf{X}_\theta\left(t, \overline{R}_{\boldsymbol{Y}}(\epsilon(t))\right)\right),$$

where $\mathsf{Vec}$ denotes the map of vertically concatenating matrices and converting them into a single vector. $\nabla_\gamma \overline{R}_{\boldsymbol{Y}}^{-1}$ can be calculated using the derivative of (123):

$$\nabla_\gamma \overline{R}_{\boldsymbol{Y}}^{-1} = 2\left(\boldsymbol{N}^{-\top} \otimes \boldsymbol{I}_D\right)\nabla \boldsymbol{M} + 2\left(\boldsymbol{I}_k \otimes \boldsymbol{M}\right)\nabla \boldsymbol{N}^{-1}, \tag{55}$$

where $\boldsymbol{X} = \gamma(t)$, $\boldsymbol{M} = \boldsymbol{Y} - \boldsymbol{X}\boldsymbol{X}^\top \boldsymbol{Y}$, $\boldsymbol{N} = \boldsymbol{I}_k + \boldsymbol{X}^\top \boldsymbol{Y}$, $\nabla \boldsymbol{M} = \boldsymbol{I}_{Dk} - \left(\boldsymbol{I}_k \otimes \boldsymbol{X}\boldsymbol{X}^\top\right)$ and $\nabla \boldsymbol{N}^{-1} = -\left(\boldsymbol{N}^{-\top} \otimes \boldsymbol{N}^{-1}\right)\left(\boldsymbol{I}_k \otimes \boldsymbol{X}^\top\right)$. Because (8) is an ODE on $T_{\boldsymbol{Y}}^{\mathrm{h}} \mathrm{St}(k, D) \cong \mathbb{R}^{(D-k)\times k}$ from Absil et al. (2008), it can be solved using an ODE solver such as Runge-Kutta methods that operate on Euclidean space. This study used Algorithm 5.1 presented in Celledoni & Owren (2002). In each step, first, (8) was solved using the Runge-Kutta method of order 5 as in Dormand & Prince (1980). Subsequently, the solution of ODE (52) was obtained by applying the solution $\epsilon$ to (53).

### C.2  Loss Function based on Variational Inference

In the setting in Section 6.3, we used orthonormalized data $\boldsymbol{Y}$ as in $\boldsymbol{P}\boldsymbol{P}^\top \simeq \boldsymbol{Y}\boldsymbol{\Lambda}\boldsymbol{Y}^\top$, where $\boldsymbol{\Lambda}$ is diagonal (Huang et al. (2015)), such that the $k$-dimensional point cloud data $\boldsymbol{P}$ of $N$ points $N \times k$ matrix is a matrix with $k$ orthonormal basis vectors. Thus, generating a complete point cloud requires the estimation of the scale parameters $\sqrt{\boldsymbol{\Lambda}}$ to be $\boldsymbol{P} = \boldsymbol{Y}\sqrt{\boldsymbol{\Lambda}}$ and a loss function that incorporates this. In this study, we approximated by maximizing the evidence lower bound (ELBO), which is the lower bound of the overall log-likelihood $\log p_\psi(\boldsymbol{P})$ of $p_\psi(\boldsymbol{P})$, using a variational inference framework. The loss function is the variational energy $-\mathrm{ELBO}(\boldsymbol{P})$ with negative ELBO.

$$\mathrm{NLL} = -\log p_\psi(\boldsymbol{P}) \leq -\mathrm{ELBO}(\boldsymbol{P}) = \mathrm{Loss}. \tag{56}$$

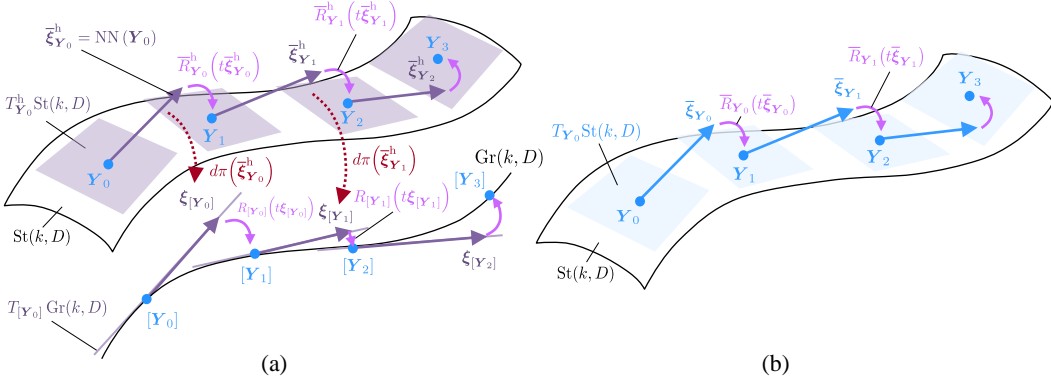

(a)                                                     (b)

Figure 7: The proposed ODE solver on the Grassmann manifold and the ODE solver on the Stiefel manifold (Celledoni & Owren (2002)). (a) The proposed ODE solver working on a Grassmann manifold. The $\bar{\boldsymbol{\xi}}_{\boldsymbol{Y}}^{\mathrm{h}}$ obtained by the proposed neural architecture NN is the horizontal lift (98), that is, $d\pi\left(\boldsymbol{Y}\right)\left[\bar{\boldsymbol{\xi}}_{\boldsymbol{Y}}^{\mathrm{h}}\right] = \boldsymbol{\xi}_{[\boldsymbol{Y}]}$, of the tangent vector $\boldsymbol{\xi}_{[\boldsymbol{Y}]}$ at the point $[\boldsymbol{Y}]$ on the Grassmann manifold. The proposed ODE solver maps and updates the tangent vector $\boldsymbol{\xi}_{[\boldsymbol{Y}]}$ at the point $[\boldsymbol{Y}]$ onto the Grassmann manifold at each step using horizontal retraction (121). On the other hand, (b) the solver of Celledoni & Owren (2002), working on Stiefel manifolds, updates in each step by mapping the tangent vector $\bar{\boldsymbol{\xi}}_{\boldsymbol{Y}}$ at the point $\boldsymbol{Y}$ onto the Stiefel manifold. In other words, the difference between the proposed ODE solver and the ODE solver on Stiefel manifolds is that the ODE solver on Stiefel manifolds works only on Stiefel manifolds, while the proposed ODE solver always updates in each step with the Stiefel manifold and Grassmann manifolds linked together.

ELBO $(\boldsymbol{P})$ can be decomposed as follows.

$$\text{ELBO}\left(\boldsymbol{P}\right) = \log p_\psi(\boldsymbol{P}) - D_{KL}\left(q_\phi(\boldsymbol{Y}\,|\,\boldsymbol{P})||p_\psi(\boldsymbol{Y}\,|\,\boldsymbol{P})\right) \tag{57}$$

$$= \mathbb{E}_{q_\phi(\boldsymbol{Y}\,|\,\boldsymbol{P})}\left[\log p_\psi(\boldsymbol{P}\,|\,\boldsymbol{Y})\right] - D_{KL}\left(q_\phi(\boldsymbol{Y}\,|\,\boldsymbol{P})||p_\theta(\boldsymbol{Y})\right), \tag{58}$$

where $q_\phi(\boldsymbol{Y}\,|\,\boldsymbol{P})$ is the inference model with parameter $\phi$, $p_\psi(\boldsymbol{P}\,|\,\boldsymbol{Y})$ is the decoder model with parameter $\psi$, $p_\psi(\boldsymbol{Y}\,|\,\boldsymbol{P})$ is the posterior distribution with parameter $\psi$, and $p_\theta(\boldsymbol{Y})$ is the prior distribution with parameter $\theta$. Further, $D_{KL}\left(q_\phi(\boldsymbol{Y}\,|\,\boldsymbol{P})||p_\theta(\boldsymbol{Y})\right)$ can be formulated using differential entropy as follows.

$$D_{KL}\left(q_\phi(\boldsymbol{Y}\,|\,\boldsymbol{P})||p_\theta(\boldsymbol{Y})\right) = -\mathbb{E}_{q_\phi(\boldsymbol{Y}\,|\,\boldsymbol{P})}\left[p_\theta(\boldsymbol{Y})\right] - H\left[q_\phi(\boldsymbol{Y}\,|\,\boldsymbol{P})\right]. \tag{59}$$

Thus, the final loss function is as follows.

$$\text{Loss} = -\text{ELBO}\left(\boldsymbol{P}\right) \tag{60}$$

$$= -\mathbb{E}_{q_\phi(\boldsymbol{Y}\,|\,\boldsymbol{P})}\left[\log p_\psi(\boldsymbol{P}\,|\,\boldsymbol{Y})\right] - \mathbb{E}_{q_\phi(\boldsymbol{Y}\,|\,\boldsymbol{P})}\left[p_\theta(\boldsymbol{Y})\right] - H\left[q_\phi(\boldsymbol{Y}\,|\,\boldsymbol{P})\right]. \tag{61}$$

Each term of the loss function can be calculated as follows.

**Expectation of log-likelihood** $\mathbb{E}_{q_\phi(\boldsymbol{Y}\,|\,\boldsymbol{P})}\left[\log p_\psi(\boldsymbol{P}\,|\,\boldsymbol{Y})\right]$ is the reconstruction log-likelihood of $\boldsymbol{P}$. The expectation is estimated by Monte Carlo sampling.

**Differential entropy** In the decomposition of a point cloud $\boldsymbol{P}$, there exists arbitrariness in the choice of $\boldsymbol{Y}$ and $\boldsymbol{\Lambda}$, as in $\boldsymbol{P}\boldsymbol{P}^\top \simeq \boldsymbol{Y}\boldsymbol{\Lambda}\boldsymbol{Y}^\top$. In this study, we assumed that the diagonal components of $\boldsymbol{\Lambda}$ are in descending order, and we restricted the decomposition arbitrariness to be an action $\boldsymbol{Q} \in \mathcal{O}(k)$. If we suppose that the action $\boldsymbol{Q}$ follows a uniform distribution when $\boldsymbol{Y} = \boldsymbol{X}\boldsymbol{Q}$ holds, then $\boldsymbol{Y}$ also follows a uniform distribution in the $k$-dimensional subspace $\text{span}(\boldsymbol{Y})$. Although this is a uniform distribution on $\text{St}(k,k)$, we can consider a uniform distribution on $\mathcal{O}(k)$ because $\text{St}(k,k) = \mathcal{O}(k)$. The probability density function of $\boldsymbol{Q}$ is represented by $U_{\mathcal{O}(k)}\left(\boldsymbol{Q}\right)$ in (111).

Therefore, the differential entropy of the decoder model can be calculated as follows.

$$H\left[q_\phi(\boldsymbol{Y}\,|\,\boldsymbol{P})\right] = \mathbb{E}\left[-\log q_\phi(\boldsymbol{Y}\,|\,\boldsymbol{P})\right] \tag{62}$$

$$= -\int_{\mathcal{O}(k)} U_{\mathcal{O}(k)}\left(\boldsymbol{Q}\right)\log U_{\mathcal{O}(k)}\left(\boldsymbol{Q}\right)[d\boldsymbol{Q}] \tag{63}$$

$$= -\int_{\mathcal{O}(k)} \frac{1}{V_{\mathcal{O}(k)}}\log\frac{1}{V_{\mathcal{O}(k)}}[d\boldsymbol{Q}] \tag{64}$$

$$= \frac{\log V_{\mathcal{O}(k)}}{V_{\mathcal{O}(k)}}\int_{\mathcal{O}(k)}[d\boldsymbol{Q}] \tag{65}$$

$$= \frac{\log V_{\mathcal{O}(k)}}{V_{\mathcal{O}(k)}}. \tag{66}$$

**Expectation of prior distribution**   We used (9) for the prior distribution $p_\theta(\boldsymbol{Y})$. Further, reparameterization was used to enable differentiable Monte Carlo estimation of expectations.

$$\mathbb{E}_{q_\phi(\boldsymbol{Y}\,|\,\boldsymbol{P})}\left[p_\theta(\boldsymbol{Y})\right] = \frac{1}{L}\sum_{l=1}^{L} p_\theta(\boldsymbol{X}\boldsymbol{Q}_l) \quad \text{s.t.} \quad \boldsymbol{X}\sim\boldsymbol{Y},\ \boldsymbol{Q}_l\in\mathcal{O}(k)\,, \tag{67}$$

where it was assumed that $\boldsymbol{Q}$ is sampled from a uniform distribution, which follows Haar measure on $\mathcal{O}(k)$. $L$ is set $L = 1$.

### C.3   Implementation Details and Experimental setting

The following sections present more details about the network architectures, training hyperparameters, and experimental conditions for each of the experiments in Section 6.

#### C.3.1   Artificial Textures

**Network Architecture**   The vector field was constructed with the specific input, intermediate, and output layers described in Section 5.3. The GrCNF architecture is shown on top in Table 3. Layers are denoted as Layer in the table, and were processed from top to bottom. Norm. and Act. denote the normalization and activation functions to be applied immediately after the Layer, and the Norm. and Act. were applied in that order. Further, Out Size denotes the output size after Act. Vec denotes the map of vertically concatenating matrices and converting them into a single vector. Moreover, only row Input denotes the size of the input data, not the input layer (HorP). (9) was used for the loss function.

**Hyper-parameters**   The mean $\boldsymbol{M}$ and covariances $\boldsymbol{U}$ and $\boldsymbol{V}$ in the prior distribution on the Grassmann manifold were set to $\boldsymbol{M} = (1.0, 0.0, 0.0)^\top$, $\boldsymbol{U} = \sigma^2 \boldsymbol{I}_3$, and $\boldsymbol{V} = \boldsymbol{I}_1$, $\sigma = 0.3$, respectively. Other hyperparameters used during the training of GrCNF are shown in Table 4.

**Implementation**   We used PyTorch (Paszke et al. (2019)) to implement the model and run the experiments. The CNF is based on the implementation[5] in Chen et al. (2018) and the framework of the RCNF (Mathieu & Nickel (2020)). Thus, the ODE was solved using the explicit and adaptive Runge–Kutta method (Dormand & Prince (1980)) of order 5, and worked by projecting each step onto a manifold (Hairer (2006)). The autograd in (7) was calculated with torch.autograd.grad (Paszke et al. (2017)) in PyTorch. The experimental hardware was built with an Intel Core i7-9700 CPU and a single NVIDIA GTX 1060 GPU with 6 GB of RAM.

The code used in the experiment to generate the data distributions on $\mathrm{Gr}(1,3)$ is shown in Listing 1. This implementation of the data distributions is based on the codes in Kim et al. (2020) and Grathwohl et al. (2019)[6].

---

[5]We used the authors' implementation: `https://github.com/rtqichen/torchdiffeq.git`.

[6]We used the authors' implementations: `https://github.com/ANLGBOY/SoftFlow.git` and `https://github.com/rtqichen/ffjord.git`.

Listing 1: Code for the data distributions.

```python
import numpy as np

def get_data_batch(batch_size, dist):
    rng = np.random.RandomState()

    if dist == "2spirals":
        n = np.sqrt(np.random.rand(batch_size // 2, 1)) * 540 * (2 * np.pi) / 360
        d1x = -np.cos(n) * n + np.random.rand(batch_size // 2, 1) * 0.1
        d1y = np.sin(n) * n + np.random.rand(batch_size // 2, 1) * 0.1
        x = np.vstack((np.hstack((d1x, d1y)), np.hstack((-d1x, -d1y)))) / 3
        sample_2d = x + np.random.randn(*x.shape) * 0.1

    elif dist == "swissroll":
        data = sklearn.datasets.make_swiss_roll(n_samples=batch_size, noise=.3)[0]
        data = data.astype("float32")[:, [0, 2]]
        sample_2d = data / 5

    elif dist == "2circles":
        data = sklearn.datasets.make_circles(n_samples=batch_size, \
                factor=.5, noise=0.05)[0]
        data = data.astype("float32")
        sample_2d = data * 3

    elif dist == "2sines":
        x = (rng.rand(batch_size) - 0.5) * 2 * np.pi
        u = (rng.binomial(1, 0.5, batch_size) - 0.5) * 2
        y = u * np.sin(x) * 2.5
        x += np.random.randn(*x.shape) * 0.1
        y += np.random.randn(*y.shape) * 0.1
        sample_2d = np.stack((x, y), 1)

    elif dist == "target":
        shapes = np.random.randint(7, size=batch_size)
        mask = []
        for i in range(7):
            mask.append((shapes == i) * 1.)

        theta = np.linspace(0, 2 * np.pi, batch_size, endpoint=False)
        x = (mask[0] + mask[1] + mask[2]) * (rng.rand(batch_size) - 0.5) * 4 + \
            (-mask[3] + mask[4] * 0.0 + mask[5]) * 2 * np.ones(batch_size) + \
            mask[6] * np.cos(theta)
        y = (mask[3] + mask[4] + mask[5]) * (rng.rand(batch_size) - 0.5) * 4 + \
            (-mask[0] + mask[1] * 0.0 + mask[2]) * 2 * np.ones(batch_size) + \
            mask[6] * np.sin(theta)
        x += np.random.randn(*x.shape) * 0.1
        y += np.random.randn(*y.shape) * 0.1
        sample_2d = np.stack((x, y), 1)

    norm = sample_2d / np.max(np.linalg.norm(sample_2d, axis=1))
    sample_3d = np.concatenate((np.ones((batch_size, 1)), norm), axis=1)
    return sample_3d / np.linalg.norm(sample_3d, axis=1)[:, np.newaxis]
```

### C.3.2 DW4 and LJ13

**Network Architecture**    As in Appendix C.3.1, the vector field was constructed with the specific input, intermediate, and output layers described in Section 5.3. The GrCNF architecture is shown on the bottom left and right in Table 3. The bottom left and right were used for experiments on the DW4 and LJ13 datasets, respectively. The views presented in the table is the same as in Appendix C.3.1. (9) was used for the loss function. In addition, for architectures in methods other than GrCNF, please refer to Garcia Satorras et al. (2021).

**Hyperparameters**    The mean $M$ and covariances $U$ and $V$ in the prior distribution on the Grassmann manifold were set to $M = I_{4\times2}$, $U = \sigma^2 I_4$, and $V = \sigma^2 I_2$, $\sigma = 0.3$ for DW4 and $M = I_{13\times3}$, $U = \sigma^2 I_{13}$, and $V = \sigma^2 I_3$, $\sigma = 0.3$ for LJ13, respectively. Other hyperparameters used during the training of GrCNF are shown in Table 4. In addition, for the hyperparameters in methods other than GrCNF, please refer to Garcia Satorras et al. (2021).

**Implementation**    The experimental hardware was built using a single NVIDIA Quadro RTX 8000 GPU with 48 GB of GDDR6 RAM. The other environments were the same as in Appendix C.3.1.

Table 3: Network architectures for each experiment; (left) for the Simple Texture dataset, (middle) for the DW4 dataset, (right) for the LJ13 dataset.

**GrCNF for Textures**

| Layer | Out Size | Norm./Act. |
| --- | --- | --- |
| Input | 3×1 | - |
| HorP | 3×1 | -/Tanh |
| Vec | 3 | -/- |
| CS | 64 | -/Tanh |
| CS | 64 | -/Tanh |
| CS | 1 | -/Tanh |
| Grad | 3×1 | - |

**GrCNF for DW4**

| Layer | Out Size | Norm./Act. |
| --- | --- | --- |
| Input | 4×2 | - |
| HorP | 4×2 | -/Tanh |
| Vec | 8 | -/- |
| CS | 64 | -/Tanh |
| CS | 64 | -/Tanh |
| CS | 64 | -/Tanh |
| CS | 1 | -/Tanh |
| Grad | 4×2 | - |

**GrCNF for LJ13**

| Layer | Out Size | Norm./Act. |
| --- | --- | --- |
| Input | 13×3 | - |
| HorP | 13×3 | -/Tanh |
| Vec | 39 | -/- |
| CS | 32 | -/Tanh |
| CS | 32 | -/Tanh |
| CS | 32 | -/Tanh |
| CS | 1 | -/Tanh |
| Grad | 13×3 | - |

**Full results** In Tables 5 and 6, the same DW4 and LJ13 averaged results from Section 6.2 were reported; however, they included the standard deviations over the three runs.

### C.3.3 QM9 Positional

**Network Architecture** On the QM9 Positional, we addressed the task of generating the molecular $P$ by estimating the scale parameter $\sqrt{\Lambda} = \mathrm{diag}\left(\left\{\sqrt{\lambda_i}\right\}_{i=1}^3\right)$, in addition to the generation of the orthonormal basis matrix $Y$ with GrCNF. Because the molecular generation task requires a specialized loss function based on variational inference, we used (56), as explained in Appendix C.2. We designed two networks to achieve this. The first is the same GrCNF architecture as in previous experiments, and the second is a scale estimator. Table 7 shows the architectures. The left side of the table shows the GrCNF architecture and the right side shows the scale estimator. The scale estimator estimated one scale parameter from each of the three orthonormal basis vectors $Y = \left\{y_i \in \mathbb{R}^{19}\right\}_{i=1}^3$, for a total of three parameters $\left\{\sqrt{\lambda_i} \in \mathbb{R}\right\}_{i=1}^3$. With the orthonormal orthogonal basis matrix $Y$ and the estimated scale parameter $\sqrt{\Lambda}$, we generated a point cloud $P = Y\sqrt{\Lambda}$. In this study, the overall architecture that generates $P$ is also named GrCNF.

**Hyperparameters** The mean $M$ and covariances $U$ and $V$ in the prior distribution on the Grassmann manifold were set to $M = I_{19\times3}$, $U = \sigma^2 I_{19}$, and $V = \sigma^2 I_3$, $\sigma = 0.3$, respectively. In addition, for the hyperparameters in methods other than GrCNF, please refer to Garcia Satorras et al. (2021).

**Implementation** The experimental hardware was built using a single NVIDIA A100 GPU with 80GB PCIe of GDDR6 RAM.

Table 4: List of hyperparameters used in various experiments. A "-" indicates that the hyperparameter is unused.

| | | Textures | DW4 | LJ13 | QM9 |
|---|---|---|---|---|---|
| **# of Data** | Train | $\infty$ | $10^2/10^3/10^4/10^5$ | $10/10^2/10^3/10^4$ | $13,831$ |
| | Validation | 500 | 1000 | 1000 | 2,501 |
| | Test | 500 | 1000 | 1000 | 1,813 |
| **Optimizer** | Name | Adam | Adam | Adam | Adam |
| | beta1 | 0.9 | 0.9 | 0.9 | 0.9 |
| | beta2 | 0.999 | 0.999 | 0.999 | 0.999 |
| | Weight Decay | - | 1.0e-12 | 1.0e-12 | 1.0e-12 |
| | Learning Rate | 1.0e-3 | 1.0e-4 | 1.0e-4 | 5.0e-4 |
| **Schedule** | Epoch | 72000 | 1000/300/50/6 | 500/1000/300/50 | 160 |
| | LR Step with 0.1 | 20000 | - | - | - |
| | Batch Size | 500 | 100 | 10/100/100/100 | 128 |
| **NeuralODE** | Integration Time | Training | Training | Training | 0.1 |
| | atol | 1.0e-5 | 1.0e-5 | 1.0e-5 | 1.0e-5 |
| | rtol | 1.0e-5 | 1.0e-5 | 1.0e-5 | 1.0e-5 |
| | Adjoint | ✗ | ✓ | ✓ | ✓ |

Table 5: Negative log-likelihood comparison on the test partition of DW4 dataset for different amounts of training samples; averaged over 3 runs and including standard deviations.

| | DW4 | | | |
|---|---|---|---|---|
| # of Samples | $10^2$ | $10^3$ | $10^4$ | $10^5$ |
| **GNF** | $-2.30 \pm 1.59$ | $-7.04 \pm 0.64$ | $-7.19 \pm 0.99$ | $-7.93 \pm 1.10$ |
| **GNF-att** | $-2.02 \pm 1.34$ | $-4.13 \pm 1.20$ | $-5.25 \pm 0.89$ | $-6.74 \pm 0.89$ |
| **GNF-att-aug** | $-3.11 \pm 2.15$ | $-4.04 \pm 3.40$ | $-6.51 \pm 0.49$ | $-9.42 \pm 1.15$ |
| **Simple dynamics** | $-1.22 \pm 0.05$ | $-1.28 \pm 0.01$ | $-1.36 \pm 0.02$ | $-1.39 \pm 0.04$ |
| **E-NF** | $-0.54 \pm 0.45$ | $-9.89 \pm 2.30$ | $-12.15 \pm 1.16$ | $-15.29 \pm 0.53$ |
| **GrCNF** | $\mathbf{-12.53 \pm 0.92}$ | $\mathbf{-13.74 \pm 0.30}$ | $\mathbf{-14.09 \pm 0.44}$ | $\mathbf{-16.07 \pm 0.46}$ |

Table 6: Negative log-likelihood comparison on the test partition of LJ13 dataset for different amounts of training samples; averaged over 3 runs and including standard deviations.

| | LJ13 | | | |
|---|---|---|---|---|
| # Samples | 10 | $10^2$ | $10^3$ | $10^4$ |
| **GNF** | $6.77 \pm 0.39$ | $-0.76 \pm 1.12$ | $-4.26 \pm 2.76$ | $-12.43 \pm 1.21$ |
| **GNF-att** | $6.91 \pm 0.17$ | $1.40 \pm 0.79$ | $-6.81 \pm 2.09$ | $-12.05 \pm 2.28$ |
| **GNF-att-aug** | $2.95 \pm 0.55$ | $-6.11 \pm 1.12$ | $-13.94 \pm 0.95$ | $-15.74 \pm 0.58$ |
| **Simple dynamics** | $-1.10 \pm 2.55$ | $-3.87 \pm 0.25$ | $-3.72 \pm 0.08$ | $-3.59 \pm 0.52$ |
| **E-NF** | $-12.86 \pm 3.67$ | $-15.75 \pm 5.02$ | $-31.51 \pm 1.19$ | $-32.83 \pm 1.98$ |
| **GrCNF** | $\mathbf{-23.64 \pm 2.23}$ | $\mathbf{-44.24 \pm 4.26}$ | $\mathbf{-58.02 \pm 5.43}$ | $\mathbf{-58.71 \pm 4.71}$ |

Table 7: Network architectures for QM9 Positional. (Left) GrCNF architecture, (Right) scale estimation architecture. The scale estimator estimates one scale parameter from each of the three orthonormal basis vectors $\boldsymbol{Y} = \left\{ \boldsymbol{y}_i \in \mathbb{R}^{19} \right\}_{i=1}^{3}$, for $\sqrt{\boldsymbol{\Lambda}} = \mathrm{diag}\left( \left\{ \sqrt{\lambda_i} \right\}_{i=1}^{3} \right)$ with three parameters $\left\{ \sqrt{\lambda_i} \in \mathbb{R} \right\}_{i=1}^{3}$. Using the orthonormal orthogonal basis matrix $\boldsymbol{Y}$ and the estimated scale parameter $\sqrt{\boldsymbol{\Lambda}}$, we generate a point cloud $\boldsymbol{P} = \boldsymbol{Y}\sqrt{\boldsymbol{\Lambda}}$. SiLU is an activation function proposed in (Ramachandran et al. (2017)) and BatchNorm. is a batch normalization layer in (Ioffe & Szegedy (2015)).

**GrCNF for QM9 Positional**

| Layer | Out Size | Norm./Act. |
| --- | --- | --- |
| Input | 19×3 | - |
| HorP | 19×3 | -/Tanh |
| Vec | 57 | -/- |
| CS | 32 | -/Tanh |
| CS | 32 | -/Tanh |
| CS | 32 | -/Tanh |
| CS | 1 | -/Tanh |
| Grad | 19×3 | - |

**Scale Estimator for QM9 Positional**

| Layer | Out Size | Norm./Act. |
| --- | --- | --- |
| Input | 19×3 | - |
| FC | 128 | BatchNorm./SiLU |
| FC | 256 | BatchNorm./SiLU |
| FC | 256 | BatchNorm./SiLU |
| FC | 128 | BatchNorm./SiLU |
| FC | 3 | -/ReLU |

# D Fundamentals of Concepts Associated with Grassmann Manifold

## D.1 Definition of Stiefel Manifold

**Definition 2.** *An (orthogonal or compact) Stiefel manifold* $\mathrm{St}(k, D)$ *is defined as the set of orthonormal bases of $k$-dimensional subspaces in the Euclidean space $\mathbb{R}^D$ as in (68).*

$$\mathrm{St}(k, D) := \left\{ \boldsymbol{Y} \in \mathbb{R}^{D \times k} \mid \boldsymbol{Y}^\top \boldsymbol{Y} = \boldsymbol{I}_k \right\}. \tag{68}$$

For $\boldsymbol{Y} \in \mathrm{St}(k, D)$, the space $\mathrm{span}(\boldsymbol{Y})$ spanned by its column vectors is the element of $\mathrm{Gr}(k, D)$.

$$f : \mathrm{St}(k, D) \to \mathrm{Gr}(k, D) : \boldsymbol{Y} \mapsto \mathrm{span}(\boldsymbol{Y}). \tag{69}$$

$\mathrm{St}(k, D)$ is a $Dk - \frac{k(k+1)}{2}$-dimensional compact manifold (Absil et al. (2008)).

## D.2 Equivalence Relation

To define the equivalence relation $\sim$ [7] on a Stiefel manifold $\mathrm{St}(k, D)$, we introduce the following two lemmas.

**Lemma 1.** *The necessary and sufficient conditions for $\mathrm{span}(\boldsymbol{Y}_1) = \mathrm{span}(\boldsymbol{Y}_2)$ to hold for $\boldsymbol{Y}_1, \boldsymbol{Y}_2 \in \mathrm{St}(k, D)$ are as follows.*

$$^{\exists}\boldsymbol{Q} \in \mathcal{O}(k) \quad s.t. \quad \boldsymbol{Y}_2 = \boldsymbol{Y}_1 \boldsymbol{Q}. \tag{70}$$

*Proof.* $\mathrm{span}(\boldsymbol{Y}_1) = \mathrm{span}(\boldsymbol{Y}_2) \Leftarrow \boldsymbol{Y}_2 = \boldsymbol{Y}_1 \boldsymbol{Q}$. From the definition, $\boldsymbol{Y}_1^\top \boldsymbol{Y}_1 = \boldsymbol{Y}_2^\top \boldsymbol{Y}_2 = \boldsymbol{I}_k$, the following is obtained:

$$\boldsymbol{I}_k = \boldsymbol{Y}_1^\top \boldsymbol{Y}_1 = \boldsymbol{Q}^\top \boldsymbol{Y}_2^\top \boldsymbol{Y}_2 \boldsymbol{Q} = \boldsymbol{Q}^\top \boldsymbol{Q}. \tag{71}$$

Thus, there exists a $k$-dimensional orthogonal matrix $\boldsymbol{Q} \in \mathcal{O}(k)$. As the subspace $\mathrm{span}(\boldsymbol{Y})$ is invariant to coordinate transformations by orthogonal matrices, $\mathrm{span}(\boldsymbol{Y}_1) = \mathrm{span}(\boldsymbol{Y}_2)$ is true.

$\mathrm{span}(\boldsymbol{Y}_1) = \mathrm{span}(\boldsymbol{Y}_2) \Rightarrow \boldsymbol{Y}_2 = \boldsymbol{Y}_1 \boldsymbol{Q}$. From the assumption, we immediately concluded that $\boldsymbol{Y}_2 = \boldsymbol{Y}_1 \boldsymbol{Q}$ for $\boldsymbol{Q} \in \mathcal{O}(k)$. $\qquad\square$

**Lemma 2.** *We define the equivalence relation $\sim$ on $\mathrm{St}(k, D)$ to be $\boldsymbol{Y}_1 \sim \boldsymbol{Y}_2$ whenever (70) is satisfied with respect to $\boldsymbol{Y}_1, \boldsymbol{Y}_2 \in \mathrm{St}(k, D)$. The fact that a binary relation is an equivalence relation $\sim$ implies that the following three statements hold for $^\forall \boldsymbol{Y}_1, \boldsymbol{Y}_2, \boldsymbol{Y}_3 \in \mathrm{St}(k, D)$.*

**Reflexivity** $\boldsymbol{Y}_1 \sim \boldsymbol{Y}_1$.

**Symmetry** $\boldsymbol{Y}_1 \sim \boldsymbol{Y}_2 \Rightarrow \boldsymbol{Y}_2 \sim \boldsymbol{Y}_1$.

**Transitivity** $\boldsymbol{Y}_1 \sim \boldsymbol{Y}_2 \wedge \boldsymbol{Y}_2 \sim \boldsymbol{Y}_3 \Rightarrow \boldsymbol{Y}_1 \sim \boldsymbol{Y}_3$.

*Proof.* With Lemma 1, we can confirm that it is valid as follows:

**Reflexivity** From $\boldsymbol{Y}_1 = \boldsymbol{Y}_1 \boldsymbol{I}$, $\boldsymbol{I} \in \mathcal{O}$, we obtain $\boldsymbol{Y}_1 \sim \boldsymbol{Y}_1$.

**Symmetry** As $\boldsymbol{Y}_2 = \boldsymbol{Y}_1 \boldsymbol{Q}$ is obtained from $\boldsymbol{Y}_1 \sim \boldsymbol{Y}_2$, and $\boldsymbol{Y}_2 \boldsymbol{Q}^\top = \boldsymbol{Y}_1$, $\boldsymbol{Q}^\top \in \mathcal{O}$ is true, then $\boldsymbol{Y}_2 \sim \boldsymbol{Y}_1$ is obtained.

**Transitivity** $\boldsymbol{Y}_3 = \boldsymbol{Y}_1 \boldsymbol{Q}_1 \boldsymbol{Q}_2$ with $\boldsymbol{Y}_2 = \boldsymbol{Y}_1 \boldsymbol{Q}_1$ and $\boldsymbol{Y}_3 = \boldsymbol{Y}_2 \boldsymbol{Q}_2$. $\boldsymbol{Q}_1 \boldsymbol{Q}_2$ is $(\boldsymbol{Q}_1 \boldsymbol{Q}_2)^\top (\boldsymbol{Q}_1 \boldsymbol{Q}_2) = \boldsymbol{Q}_2^\top \boldsymbol{Q}_1^\top \boldsymbol{Q}_1 \boldsymbol{Q}_2 = \boldsymbol{Q}_2^\top \boldsymbol{Q}_2 = \boldsymbol{I}$. Moreover, as $(\boldsymbol{Q}_1 \boldsymbol{Q}_2)(\boldsymbol{Q}_1 \boldsymbol{Q}_2)^\top = \boldsymbol{Q}_1 \boldsymbol{Q}_2 \boldsymbol{Q}_2^\top \boldsymbol{Q}_1^\top = \boldsymbol{I}$ holds, $\boldsymbol{Q}_1 \boldsymbol{Q}_2 \in \mathcal{O}$ is true. Therefore, we concluded $\boldsymbol{Y}_1 \sim \boldsymbol{Y}_3$. $\qquad\square$

The equivalence class of $\boldsymbol{Y} \in \mathrm{St}(k, D)$ is denoted by $[\boldsymbol{Y}]$. In other words, $[\boldsymbol{Y}]$ is the set of all elements of $\mathrm{St}(k, D)$ that are equivalent to $\boldsymbol{Y}$, and the equivalence relation on $\mathrm{St}(k, D)$ divides $\mathrm{St}(k, D)$ into equivalence classes with no intersection. Thus, $\boldsymbol{Y}$ is then referred to as the representative of the equivalence class $[\boldsymbol{Y}]$. The set of equivalence classes is denoted $\mathrm{St}(k, D)/\sim$ and is referred to as the

---

[7]Reflexive, symmetric and transitive binary relations. As a consequence of these properties, in a given set, one equivalence relation divides (classifies) the set into equivalence classes. Note that $R$ is a binary relation in the set $X$ if for any $x, y \in X$, only either $x$ is related to $y$ by the relation $R$, or $x$ is not related to $y$ based on the relation that $R$ occurs. We write $x$ is related to $y$ by relation $R$" as $xRy$.

quotient of $\mathrm{St}(k, D)$ by the equivalence relation $\sim$. In addition, $\pi :\to \mathrm{St}(k, D) \to \mathrm{St}(k, D)/\sim$ is the natural projection that maps $\boldsymbol{Y} \in \mathrm{St}(k, D)$ onto its equivalence class $[\boldsymbol{Y}]$. This projection $\pi$ is surjection.

## D.3 Quotient Manifold

**Definition 3.** *Let $\overline{\mathcal{M}}$ be a manifold with equivalence relation $\sim$. The quotient space $\overline{\mathcal{M}}/\sim$ with $\sim$ of $\overline{\mathcal{M}}$ is the set of all equivalence classes. Thus, $\overline{\mathcal{M}}/\sim := \{\pi(\overline{\boldsymbol{x}}) \mid \overline{\boldsymbol{x}} \in \overline{\mathcal{M}}\}$, where $\pi : \overline{\mathcal{M}} \to \overline{\mathcal{M}}/\sim$ is the natural projection and $\pi(\overline{\boldsymbol{x}}) := \{\overline{\boldsymbol{y}} \in \overline{\mathcal{M}} \mid \overline{\boldsymbol{y}} \sim \overline{\boldsymbol{x}}\}$. Then, $\overline{\mathcal{M}}$ is referred to as the total space or the total manifold. Moreover, $\overline{\mathcal{M}}/\sim$ is referred to as a quotient manifold of $\overline{\mathcal{M}}$ if $\overline{\mathcal{M}}/\sim$ admits a differentiable structure.*

Let $\mathcal{M} = \overline{\mathcal{M}}/\sim$ be a quotient manifold. Further, suppose that $\overline{\mathcal{M}}$ is endowed with a Riemann metric $\overline{g}$, and let $\boldsymbol{x} = \pi(\boldsymbol{x})$. The horizontal space $T_{\overline{\boldsymbol{x}}}^{\mathrm{h}}\overline{\mathcal{M}}$ is the orthogonal complement of the vertical space $T_{\overline{\boldsymbol{x}}}^{\mathrm{v}}\overline{\mathcal{M}} := T_{\overline{\boldsymbol{x}}}\pi^{-1}(\boldsymbol{x})$ in the tangent space $T_{\overline{\boldsymbol{x}}}\overline{\mathcal{M}}$ and is defined as the follows:

$$T_{\overline{\boldsymbol{x}}}^{\mathrm{h}}\overline{\mathcal{M}} := \left(T_{\overline{\boldsymbol{x}}}^{\mathrm{v}}\overline{\mathcal{M}}\right)^{\perp} = \left\{\overline{\boldsymbol{\eta}}_{\overline{\boldsymbol{x}}} \in T_{\overline{\boldsymbol{x}}}\overline{\mathcal{M}} \mid \overline{g}_{\overline{\boldsymbol{x}}}\left(\overline{\boldsymbol{\xi}}_{\overline{\boldsymbol{x}}}, \overline{\boldsymbol{\eta}}_{\overline{\boldsymbol{x}}}\right) = 0, ^{\vee}\overline{\boldsymbol{\xi}}_{\overline{\boldsymbol{x}}} \in T_{\overline{\boldsymbol{x}}}^{\mathrm{v}}\overline{\mathcal{M}}\right\}. \tag{72}$$

The horizontal lift $\overline{\boldsymbol{\xi}}_{\overline{\boldsymbol{x}}}^{\mathrm{h}} \in T_{\overline{\boldsymbol{x}}}^{\mathrm{h}}\overline{\mathcal{M}}$ of the tangent vector $\boldsymbol{\xi}_{\boldsymbol{x}} \in T_{\boldsymbol{x}}\mathcal{M}$ at point $\overline{\boldsymbol{x}} \in \pi^{-1}(\boldsymbol{x})$ is a tangent vector that is uniquely determined as $d\pi_{\overline{\boldsymbol{x}}}\left(\overline{\boldsymbol{\xi}}_{\overline{\boldsymbol{x}}}^{\mathrm{h}}\right) = \boldsymbol{\xi}_{\boldsymbol{x}}$ (Absil et al. (2008)).

## D.4 Grassmann Manifold Exploiting the Quotient Structure

### D.4.1 Tangent Space on a Stiefel Manifold

We describe the relationship between tangent space $T_{[\boldsymbol{Y}]}\mathrm{Gr}(k, D)$ on $\mathrm{Gr}(k, D)$ and tangent space $T_{\boldsymbol{Y}}\mathrm{St}(k, D)$ on $\mathrm{St}(k, D)$ to relate the tangent vectors of a Grassmann manifold $\mathrm{Gr}(k, D)$ to the tangent vectors of a Stiefel manifold $\mathrm{St}(k, D)$ in a matrix representation. We take the derivative on both sides of $\boldsymbol{Y}(t)^{\top}\boldsymbol{Y}(t) = \boldsymbol{I}_p$ in (68) by $t$ and solve for $t = 0$.

$$\frac{d}{dt}\left\{\boldsymbol{Y}(t)^{\top}\boldsymbol{Y}(t)\right\} = \frac{d}{dt}\boldsymbol{I}_p \tag{73}$$

$$\frac{d}{dt}\boldsymbol{Y}(t)^{\top}\boldsymbol{Y}(t) + \boldsymbol{Y}(t)^{\top}\frac{d}{dt}\boldsymbol{Y}(t) = 0 \tag{74}$$

$$\frac{d}{dt}\boldsymbol{Y}(0)^{\top}\boldsymbol{Y}(0) + \boldsymbol{Y}(0)^{\top}\frac{d}{dt}\boldsymbol{Y}(0) = 0 \tag{75}$$

$$\overline{\boldsymbol{\xi}}_{\boldsymbol{Y}}^{\top}\boldsymbol{Y} + \boldsymbol{Y}^{\top}\overline{\boldsymbol{\xi}}_{\boldsymbol{Y}} = 0, \tag{76}$$

where $\overline{\boldsymbol{\xi}}_{\boldsymbol{Y}} = \frac{d}{dt}\boldsymbol{Y}(0)$ is the tangent vector at $\boldsymbol{Y}$ [8].

**Definition 4.** *Define the tangent space $T_{\boldsymbol{Y}}\mathrm{St}(k, D)$ at $\boldsymbol{Y}$ on the Stiefel manifold as follows:*

$$T_{\boldsymbol{Y}}\mathrm{St}(k, D) = \left\{\overline{\boldsymbol{\xi}}_{\boldsymbol{Y}} \in \mathbb{R}^{D \times k} \mid \overline{\boldsymbol{\xi}}_{\boldsymbol{Y}}^{\top}\boldsymbol{Y} + \boldsymbol{Y}^{\top}\overline{\boldsymbol{\xi}}_{\boldsymbol{Y}} = \boldsymbol{0}_k\right\}. \tag{77}$$

Let matrix $\boldsymbol{Y}_{\perp} \in \mathbb{R}^{D \times (D-k)}$ be a matrix satisfying the following:

$$\boldsymbol{Y}_{\perp}^{\top}\boldsymbol{Y}_{\perp} = \boldsymbol{I}_{D-k}, \quad \boldsymbol{Y}^{\top}\boldsymbol{Y}_{\perp} = 0, \quad \boldsymbol{Y}\boldsymbol{Y}^{\top} + \boldsymbol{Y}_{\perp}\boldsymbol{Y}_{\perp}^{\top} = \boldsymbol{I}_D. \tag{78}$$

As $[\ \boldsymbol{Y} \quad \boldsymbol{Y}_{\perp}\ ]$ is an orthogonal matrix [9], the column vectors of $\boldsymbol{Y}$ and $\boldsymbol{Y}_{\perp}$ form an orthonormal basis in $\mathbb{R}^D$. Thus, any $D \times k$ matrix can be written in terms of the $\boldsymbol{C} \in \mathbb{R}^{k \times k}$ and $\boldsymbol{B} \in \mathbb{R}^{(D-k) \times k}$ coefficient matrices as follows:

$$\boldsymbol{Y}\boldsymbol{C} + \boldsymbol{Y}_{\perp}\boldsymbol{B}, \tag{79}$$

---

[8]The tangent space is defined independently for each point of the manifold; hence, the subscript $\boldsymbol{Y}$, as in $\overline{\boldsymbol{\xi}}_{\boldsymbol{Y}}$, is clearly stated to emphasize that it is a tangent vector at $\boldsymbol{Y}$.

[9]$[\ \boldsymbol{Y} \quad \boldsymbol{Y}_{\perp}\ ]^{-1}[\ \boldsymbol{Y} \quad \boldsymbol{Y}_{\perp}\ ] = [\ \boldsymbol{Y} \quad \boldsymbol{Y}_{\perp}\ ]^{\top}[\ \boldsymbol{Y} \quad \boldsymbol{Y}_{\perp}\ ] = [\ \boldsymbol{Y} \quad \boldsymbol{Y}_{\perp}\ ][\ \boldsymbol{Y} \quad \boldsymbol{Y}_{\perp}\ ]^{\top} = \boldsymbol{I}_D.$

where $\overline{\boldsymbol{\xi}}_{\boldsymbol{Y}} = \boldsymbol{Y}\boldsymbol{C} + \boldsymbol{Y}_{\perp}\boldsymbol{B}$ is inserted. The following equation is obtained.

$$\overline{\boldsymbol{\xi}}_{\boldsymbol{Y}}^{\top}\boldsymbol{Y} + \boldsymbol{Y}^{\top}\overline{\boldsymbol{\xi}}_{\boldsymbol{Y}} = (\boldsymbol{Y}\boldsymbol{C} + \boldsymbol{Y}_{\perp}\boldsymbol{B})^{\top}\boldsymbol{Y} + \boldsymbol{Y}^{\top}(\boldsymbol{Y}\boldsymbol{C} + \boldsymbol{Y}_{\perp}\boldsymbol{B}) \tag{80}$$

$$= \boldsymbol{B}^{\top}\boldsymbol{Y}_{\perp}^{\top}\boldsymbol{Y} + \boldsymbol{C}^{\top}\boldsymbol{Y}^{\top}\boldsymbol{Y} + \boldsymbol{Y}^{\top}\boldsymbol{Y}\boldsymbol{C} + \boldsymbol{Y}^{\top}\boldsymbol{Y}_{\perp}\boldsymbol{B} \tag{81}$$

$$= \boldsymbol{B}^{\top}\boldsymbol{Y}^{\top}\boldsymbol{Y}_{\perp} + \boldsymbol{C}^{\top} + \boldsymbol{C} \tag{82}$$

$$= \boldsymbol{C}^{\top} + \boldsymbol{C} \tag{83}$$

$$= \boldsymbol{0}_k. \tag{84}$$

Thus, the following equation is derived.

$$\boldsymbol{C}^{\top} + \boldsymbol{C} = \boldsymbol{0}_k. \tag{85}$$

Thus, $\boldsymbol{C}$ is a $k \times k$ skew-symmetric matrix $\mathrm{Skew}\,(k)$. Therefore, we obtain the following as another representation of the tangent space on $\mathrm{St}(k, D)$.

$$T_{\boldsymbol{Y}}\mathrm{St}(k, D) = \left\{\boldsymbol{Y}\boldsymbol{C} + \boldsymbol{Y}_{\perp}\boldsymbol{B} \,\Big|\, \boldsymbol{C} \in \mathrm{Skew}\,(k), \boldsymbol{B} \in \mathbb{R}^{(D-k)\times k}\right\}. \tag{86}$$

### D.4.2 Riemannian Metric on a Stiefel Manifold

In the tangent space $T_{\boldsymbol{x}}\mathcal{M}$ defined at each point $\boldsymbol{x} \in \mathcal{M}$ on the manifold $\mathcal{M}$, the inner product $h$ is endowed as a bilinear map. This $h$ is referred to as a Riemannian metric on a manifold, and the manifold $\mathcal{M}$ on which the Riemannian metric $h$ is endowed is referred to as a Riemannian manifold $(\mathcal{M}, h)$. We define the Riemannian metric $\overline{g}$ on $\mathrm{St}(k, D)$ as follows.

$$\overline{g}_{\boldsymbol{Y}}\left(\overline{\boldsymbol{\xi}}_{\boldsymbol{Y}}, \overline{\boldsymbol{\eta}}_{\boldsymbol{Y}}\right) := \mathrm{tr}\left(\overline{\boldsymbol{\xi}}_{\boldsymbol{Y}}^{\top}\overline{\boldsymbol{\eta}}_{\boldsymbol{Y}}\right) \quad \text{s.t.} \quad \overline{\boldsymbol{\xi}}_{\boldsymbol{Y}}, \overline{\boldsymbol{\eta}}_{\boldsymbol{Y}} \in T_{\boldsymbol{Y}}\mathrm{St}(k, D), \boldsymbol{Y} \in \mathrm{St}(k, D). \tag{87}$$

This is the standard inner product of $\mathbb{R}^{D\times k}$ induced by $T_{\boldsymbol{Y}}\mathrm{St}(k, D)$, with $T_{\boldsymbol{Y}}\mathrm{St}(k, D) \subset \mathbb{R}^{D\times k}$ [10][11].

### D.4.3 Tangent Space on a Grassmann Manifold

We describe the relation between tangent spaces $T_{[\boldsymbol{Y}]}\mathrm{Gr}(k, D)$ and $T_{\boldsymbol{Y}}\mathrm{St}(k, D)$ to relate tangent vectors in tangent spaces $T_{[\boldsymbol{Y}]}\mathrm{Gr}(k, D)$ on $\mathrm{Gr}(k, D)$ to tangent vectors $\overline{\boldsymbol{\xi}}_{\boldsymbol{Y}} \in T_{\boldsymbol{Y}}\mathrm{St}(k, D)$.

First, we define the vertical space $T_{\boldsymbol{Y}}^{\mathrm{v}}\mathrm{St}(k, D)$ as a subspace of $T_{\boldsymbol{Y}}\mathrm{St}(k, D)$ as follows.

$$T_{\boldsymbol{Y}}^{\mathrm{v}}\mathrm{St}(k, D) := T_{\boldsymbol{Y}}\pi^{-1}\left([\boldsymbol{Y}]\right), \tag{88}$$

where $\pi : \mathrm{St}(k, D) \to \mathrm{Gr}(k, D)$ is the natural projection defined by $\pi\left(\boldsymbol{Y}\right) = [\boldsymbol{Y}]$ [12]. Thus, $\pi$ converges all $\boldsymbol{Y}' \in \mathrm{St}(k, D)$ such that $\boldsymbol{Y} \sim \boldsymbol{Y}'$ to a point $[\boldsymbol{Y}]$ on $\mathrm{Gr}(k, D)$. Therefore, using (1), (88) can be transformed as follows.

$$T_{\boldsymbol{Y}}^{\mathrm{v}}\mathrm{St}(k, D) = T_{\boldsymbol{Y}}\left\{\boldsymbol{Y}\boldsymbol{Q} \mid \boldsymbol{Q} \in \mathcal{O}(k)\right\}. \tag{89}$$

However, $\overline{\boldsymbol{\xi}}_{\boldsymbol{Y}}^{\mathrm{v}} \in T_{\boldsymbol{Y}}^{\mathrm{v}}\mathrm{St}(k, D)$ can be written as $\overline{\boldsymbol{\xi}}_{\boldsymbol{Y}}^{\mathrm{v}} = \boldsymbol{Y}\boldsymbol{S}$ with $\boldsymbol{S} \in T_{\boldsymbol{I}_k}\mathcal{O}(k)$.

$$T_{\boldsymbol{I}_k}\mathcal{O}(k) = T_{\boldsymbol{I}_k}\mathrm{St}(k, k) \tag{90}$$

$$= \left\{\boldsymbol{I}_k\boldsymbol{C} + (\boldsymbol{I}_{k\perp}\boldsymbol{B} = \boldsymbol{0}_k) \,\Big|\, \boldsymbol{C} \in \mathrm{Skew}\,(k), \boldsymbol{B} \in \mathbb{R}^{(k-k=0)\times k}\right\} \tag{91}$$

$$= \mathrm{Skew}\,(k). \tag{92}$$

---

[10]The inner product $\boldsymbol{A} \cdot \boldsymbol{C} = \boldsymbol{A}^{\top}\boldsymbol{C}$ of a vector is typically referred to as the standard inner product. Further, matrices are similarly defined with a standard inner product, defined as $\boldsymbol{A} \cdot \boldsymbol{C} = \mathrm{tr}\left(\boldsymbol{A}^{\top}\boldsymbol{C}\right)$. A space $\mathbb{R}^{D\times k}$ such that the $D \times k$ matrix $\boldsymbol{A}$ is an element is referred to as a matrix space. The standard basis of the matrix space can be constructed by a matrix wherein only one element in the matrix is 1 and the remaining are 0. The matrix space is a linear space because it satisfies the linearity that is similar to that in case of a linear vector space.

[11]When $\mathcal{N}$ is a submanifold of a Riemannian manifold $(\mathcal{M}, g)$, we define the Riemannian metric $\overline{g}$ of $\mathcal{N}$ to be:

$$\overline{g}_{\boldsymbol{x}}\left(\boldsymbol{\xi}, \boldsymbol{\eta}\right) := g_{\boldsymbol{x}}\left(\boldsymbol{\xi}, \boldsymbol{\eta}\right), \quad \boldsymbol{x} \in \mathcal{N} \subset \mathcal{M}, \boldsymbol{\xi}, \boldsymbol{\eta} \in T_{\boldsymbol{x}}\mathcal{N} \subset T_{\boldsymbol{x}}\mathcal{M}.$$

$\overline{g}$ is an induced metric and $(\mathcal{N}, \overline{g})$ is a Riemannian submanifold of $(\mathcal{M}, g)$. As $\mathrm{St}(k, D)$ is a submanifold of $\mathbb{R}^{D\times k}$, we can define the standard inner product $\boldsymbol{A} \cdot \boldsymbol{C} = \mathrm{tr}\left(\boldsymbol{A}^{\top}\boldsymbol{C}\right)$of $\mathbb{R}^{D\times k}$ as the induced metric $\overline{g}$. Thus, $\mathrm{St}(k, D)$ is a Riemannian submanifold of $\mathbb{R}^{D\times k}$.

[12]Suppose a set is given a suitable equivalence relation. A natural projection is a map that sends each element of a set to the equivalence class to which it belongs.

Thus, we obtain the following formula.

$$T_{\boldsymbol{Y}}^{\mathrm{v}} \operatorname{St}(k, D) = \{\boldsymbol{Y} \boldsymbol{C} \mid \boldsymbol{C} \in \operatorname{Skew}(k)\}. \tag{93}$$

Next, we define the horizontal space $T_{\boldsymbol{Y}}^{\mathrm{h}} \operatorname{St}(k, D)$ as the orthogonal complement of $T_{\boldsymbol{Y}}^{\mathrm{v}} \operatorname{St}(k, D)$ in $T_{\boldsymbol{Y}} \operatorname{St}(k, D)$ endowed with the inner product (87).

$$T_{\boldsymbol{Y}}^{\mathrm{h}} \operatorname{St}(k, D) := (T_{\boldsymbol{Y}}^{\mathrm{v}} \operatorname{St}(k, D))^{\perp} \tag{94}$$

$$= \left\{ \overline{\boldsymbol{\xi}}_{\boldsymbol{Y}}^{\mathrm{h}} \in T_{\boldsymbol{Y}} \operatorname{St}(k, D) \;\middle|\; \operatorname{tr}\left( \overline{\boldsymbol{\xi}}_{\boldsymbol{Y}}^{\mathrm{h}\top} \overline{\boldsymbol{\eta}}_{\boldsymbol{Y}}^{\mathrm{v}} \right) = 0, \overline{\boldsymbol{\eta}}_{\boldsymbol{Y}}^{\mathrm{v}} \in T_{\boldsymbol{Y}}^{\mathrm{v}} \operatorname{St}(k, D) \right\}. \tag{95}$$

Based on the fact that $T_{\boldsymbol{Y}}^{\mathrm{v}} \operatorname{St}(k, D)$ is a subspace of $T_{\boldsymbol{Y}} \operatorname{St}(k, D)$ and $T_{\boldsymbol{Y}}^{\mathrm{h}} \operatorname{St}(k, D)$ is defined as its orthogonal complement, the direct sum decomposition is as follows.

$$T_{\boldsymbol{Y}} \operatorname{St}(k, D) = T_{\boldsymbol{Y}}^{\mathrm{v}} \operatorname{St}(k, D) \oplus T_{\boldsymbol{Y}}^{\mathrm{h}} \operatorname{St}(k, D), \tag{96}$$

where $\oplus$ denotes direct sum. Moreover, the tangent space is a linear space (Absil et al. (2008)). From (86), element $\boldsymbol{Y} \boldsymbol{C}$ of $T_{\boldsymbol{Y}}^{\mathrm{v}} \operatorname{St}(k, D)$ corresponds to the first term of (93); thus, (96) is formulated as follows:

$$T_{\boldsymbol{Y}}^{\mathrm{h}} \operatorname{St}(k, D) = \left\{ \overline{\boldsymbol{\xi}}_{\boldsymbol{Y}}^{\mathrm{h}} = \boldsymbol{Y}_{\perp} \boldsymbol{B} \;\middle|\; \boldsymbol{B} \in \mathbb{R}^{(D-k) \times k} \right\}. \tag{97}$$

Note that the horizontal vector $\overline{\boldsymbol{\xi}}_{\boldsymbol{Y}}^{\mathrm{h}}$ is not necessarily an orthogonal matrix.

Finally, define the element $\overline{\boldsymbol{\xi}}_{\boldsymbol{Y}}^{\mathrm{h}} \in T_{\boldsymbol{Y}}^{\mathrm{h}} \operatorname{St}(k, D)$ of the horizontal space at $\boldsymbol{Y} \in \operatorname{St}(k, D)$ for the tangent vector $\boldsymbol{\xi}_{[\boldsymbol{Y}]} \in T_{[\boldsymbol{Y}]} \operatorname{Gr}(k, D)$ at $[\boldsymbol{Y}] \in \operatorname{Gr}(k, D)$ as satisfying the following formula.

$$d\pi\left(\boldsymbol{Y}\right)\left[\overline{\boldsymbol{\xi}}_{\boldsymbol{Y}}^{\mathrm{h}}\right] = \boldsymbol{\xi}_{[\boldsymbol{Y}]}, \tag{98}$$

where $d\pi\left(\boldsymbol{Y}\right) : T_{\boldsymbol{Y}} \operatorname{St}(k, D) \to T_{[\boldsymbol{Y}]} \operatorname{Gr}(k, D)$ is the derivative $\frac{d\pi(\boldsymbol{Y})}{d\boldsymbol{Y}}$ of $\pi : \operatorname{St}(k, D) \to \operatorname{Gr}(k, D)$ at $\boldsymbol{Y} \in \operatorname{St}(k, D)$. The $\overline{\boldsymbol{\xi}}_{\boldsymbol{Y}}^{\mathrm{h}} \in T_{\boldsymbol{Y}}^{\mathrm{h}} \operatorname{St}(k, D)$ is referred to as the horizontal lift at $\boldsymbol{Y} \in \operatorname{St}(k, D)$ of $[\boldsymbol{Y}] \in \operatorname{Gr}(k, D)$.

We describe the tangent space of $\operatorname{Gr}(k, D)$ with the concept of horizontal lift.

**Definition 5.** *Let $T_{\boldsymbol{Y}}^{\mathrm{h}} \operatorname{St}(k, D)$ be a horizontal space on $\operatorname{St}(k, D)$. Then, we define the tangent space $T_{[\boldsymbol{Y}]} \operatorname{Gr}(k, D)$ of the $\operatorname{Gr}(k, D)$ as follows.*

$$T_{[\boldsymbol{Y}]} \operatorname{Gr}(k, D) = \left\{ \boldsymbol{\xi}_{[\boldsymbol{Y}]} \;\middle|\; d\pi\left(\boldsymbol{Y}\right)\left[\overline{\boldsymbol{\xi}}_{\boldsymbol{Y}}^{\mathrm{h}}\right] = \boldsymbol{\xi}_{[\boldsymbol{Y}]}, \overline{\boldsymbol{\xi}}_{\boldsymbol{Y}}^{\mathrm{h}} \in T_{\boldsymbol{Y}}^{\mathrm{h}} \operatorname{St}(k, D) \right\}. \tag{99}$$

From the above, $\boldsymbol{\xi}_{[\boldsymbol{Y}]} \in T_{[\boldsymbol{Y}]} \operatorname{Gr}(k, D)$ is obtained from the map $d\pi\left(\boldsymbol{Y}\right)\left[\overline{\boldsymbol{\xi}}_{\boldsymbol{Y}}^{\mathrm{h}}\right]$ when $\overline{\boldsymbol{\xi}}_{\boldsymbol{Y}}^{\mathrm{h}}$ is obtained. The $\boldsymbol{\xi}_{[\boldsymbol{Y}]}$ is defined by an equivalence class and cannot be treated numerically in matrix form; however, it is sufficient to obtain the $\overline{\boldsymbol{\xi}}_{\boldsymbol{Y}}^{\mathrm{h}}$ for actual numerical calculations. For $\boldsymbol{\xi}_{[\boldsymbol{Y}]} \in T_{[\boldsymbol{Y}]} \operatorname{Gr}(k, D)$, there exists a $\overline{\boldsymbol{\xi}}_{\boldsymbol{Y}}^{\mathrm{h}} \in T_{\boldsymbol{Y}}^{\mathrm{h}} \operatorname{St}(k, D)$ that uniquely satisfies (98). In other words, we can handle it in matrix form by using elements of the horizontal space of Stiefel manifolds through the concept of horizontal lifting. Figure 1 is a conceptual diagram of the tangent space representation of a Grassmann manifold by horizontal lift.

### D.4.4 Riemannian Metric on a Grassmann Manifold

We define the Riemannian metric $g$ of $\operatorname{Gr}(k, D)$ through the concept of horizontal lift.

**Definition 6.** *Let $\overline{\boldsymbol{\xi}}_{\boldsymbol{Y}}^{\mathrm{h}}$ and $\overline{\boldsymbol{\eta}}_{\boldsymbol{Y}}^{\mathrm{h}}$ be the horizontal lifts that become $d\pi\left(\boldsymbol{Y}\right)\left[\overline{\boldsymbol{\xi}}_{\boldsymbol{Y}}^{\mathrm{h}}\right] = \boldsymbol{\xi}_{[\boldsymbol{Y}]}$ and $d\pi\left(\boldsymbol{Y}\right)\left[\overline{\boldsymbol{\eta}}_{\boldsymbol{Y}}^{\mathrm{h}}\right] = \boldsymbol{\eta}_{[\boldsymbol{Y}]}$, respectively. Then, we define the Riemannian metric on $\operatorname{Gr}(k, D)$ as follows:*

$$g_{[\boldsymbol{Y}]}\left(\boldsymbol{\xi}_{[\boldsymbol{Y}]}, \boldsymbol{\eta}_{[\boldsymbol{Y}]}\right) := \overline{g}_{\boldsymbol{Y}}\left(\overline{\boldsymbol{\xi}}_{\boldsymbol{Y}}^{\mathrm{h}}, \overline{\boldsymbol{\eta}}_{\boldsymbol{Y}}^{\mathrm{h}}\right) = \operatorname{tr}\left(\boldsymbol{B}^{\top} \boldsymbol{D}\right), \tag{100}$$

*where $\boldsymbol{B}$ and $\boldsymbol{D}$ are matrices that are $\overline{\boldsymbol{\xi}}_{\boldsymbol{Y}}^{\mathrm{h}} = \boldsymbol{Y}_{\perp} \boldsymbol{B}$ and $\overline{\boldsymbol{\eta}}_{\boldsymbol{Y}}^{\mathrm{h}} = \boldsymbol{Y}_{\perp} \boldsymbol{D}$, respectively.*

## D.5 Invariant Measures

Let the column vectors of matrix $\boldsymbol{Y} = \{\boldsymbol{y}_1, \cdots, \boldsymbol{y}_k\} \in \mathbb{R}^{D \times k}$ be the orthonormal basis that span the subspace $\mathrm{span}(\boldsymbol{Y}) \in \mathrm{Gr}(k, D)$ in $\mathbb{R}^D$, and the column vectors of $\boldsymbol{Y}_\perp = \{\boldsymbol{y}_{k+1}, \cdots, \boldsymbol{y}_D\} \in \mathbb{R}^{D \times D-k}$ be the orthogonal complementary space $\mathrm{span}(\boldsymbol{Y}_\perp)$ of $\mathrm{span}(\boldsymbol{Y})$, respectively. Then, the following differential form can be defined.

$$(d\boldsymbol{Y}) = \bigwedge_{j=1}^{D-k} \bigwedge_{i=1}^{k} \boldsymbol{y}_{k+j}^\top d\boldsymbol{y}_i \tag{101}$$

$$= \left( \boldsymbol{y}_{k+1}^\top d\boldsymbol{y}_1 \wedge \cdots \wedge \boldsymbol{y}_{k+1}^\top d\boldsymbol{y}_k \right) \wedge \cdots \wedge \left( \boldsymbol{y}_D^\top d\boldsymbol{y}_1 \wedge \cdots \wedge \boldsymbol{y}_D^\top d\boldsymbol{y}_k \right), \tag{102}$$

where $\wedge$ is the wedge product and the relation satisfies $\omega_i \wedge \omega_i = \omega_j \wedge \omega_j = 0$ and $\omega_i \wedge \omega_j = -\omega_j \wedge \omega_i$. The above equation is in $k(D-k)$-order differential form, which is an invariant measure on $\mathrm{Gr}(k, D)$ (Chikuse (2003)).

If we define the matrix $\boldsymbol{X}_\perp$ to be $[\ \boldsymbol{X} \quad \boldsymbol{X}_\perp\ ]$ for any point $\boldsymbol{X} = \{\boldsymbol{x}_1, \cdots, \boldsymbol{x}_k\} \in \mathrm{St}(k, D)$, the differential form for an invariant measure on $\mathrm{St}(k, D)$ is defined as follows.

$$(d\boldsymbol{X}) = \bigwedge_{j=1}^{D-k} \bigwedge_{i=1}^{k} \boldsymbol{x}_{k+j}^\top d\boldsymbol{x}_i \bigwedge_{i<j}^{k} \boldsymbol{x}_j^\top d\boldsymbol{x}_i = (d\boldsymbol{Y})(d\boldsymbol{Q}), \tag{103}$$

where $(d\boldsymbol{Q})$ is the invariant measure of $\mathcal{O}(k)$. The integral of (103), that is, the volume of $\mathrm{St}(k, D)$, can be evaluated as follows:

$$V_{\mathrm{St}(k,D)} = \int_{\mathrm{St}(k,D)} (d\boldsymbol{X}). \tag{104}$$

(104) can be computed as follows. First, the surface $S_D$ of the $D$-dimensional unit sphere can be defined as follows:

$$S_D = \frac{d}{dr}\Big|_{r=1} V_D = D V_D = \frac{D \pi^{\frac{D}{2}}}{\Gamma\left(\frac{D}{2}+1\right)} = \frac{2 \pi^{\frac{D}{2}}}{\Gamma\left(\frac{D}{2}\right)}, \tag{105}$$

where $V_D$ is the volume of a $D$-dimensional sphere $\frac{\pi^{\frac{D}{2}}}{\Gamma\left(\frac{D}{2}+1\right)} r^D$ and $\Gamma\left(\frac{D}{2}\right)$ is the gamma function. Then, the following equation is obtained.

$$\int_{\mathrm{St}(k,D)} (d\boldsymbol{X}) = S_D \int_{\mathrm{St}(k-1,D-1)} (d\boldsymbol{X}_1), \tag{106}$$

where $(d\boldsymbol{X}_1)$ is the differential form of $\mathrm{St}(k-1, D-1)$. Thus, (104) can be transformed as follows.

$$V_{\mathrm{St}(k,D)} = \int_{\mathrm{St}(k,D)} (d\boldsymbol{X}) = \prod_{i=1}^{k} S_D = \frac{2^k \pi^{\frac{Dk}{2}}}{\Gamma_k\left(\frac{D}{2}\right)}, \tag{107}$$

where $\Gamma_k\left(\frac{D}{2}\right)$ is the multidimensional gamma function. The invariant measure $(d\boldsymbol{X})$ is an unnormalized measure. A measure normalized to be a probability measure can be formulated as follows:

$$[d\boldsymbol{X}] = \frac{1}{V_{\mathrm{St}(k,D)}} (d\boldsymbol{X}). \tag{108}$$

This is a uniform distribution on $\mathrm{St}(k, D)$. As $\mathrm{St}(k, k) = \mathcal{O}(k)$, the volume $V_{\mathcal{O}(k)}$ of $\mathcal{O}(k)$ can be represented using $(d\boldsymbol{Q})$ as follows.

$$V_{\mathcal{O}(k)} = V_{\mathrm{St}(k,k)} = \frac{2^k \pi^{\frac{k^2}{2}}}{\Gamma_k\left(\frac{k}{2}\right)}. \tag{109}$$

Furthermore, a measure normalized to be a probability measure can be represented by the following:

$$[d\boldsymbol{Q}] = \frac{1}{V_{\mathcal{O}(k)}} (d\boldsymbol{Q}). \tag{110}$$

From the above, the probability density function $U_{\mathcal{O}(k)}(\boldsymbol{Q})$ of the uniform distribution on $\mathcal{O}(k)$ is as follows:

$$U_{\mathcal{O}(k)}(\boldsymbol{Q}) = \frac{1}{V_{\mathcal{O}(k)}} \quad \text{s.t.} \quad \boldsymbol{Q} \in \mathcal{O}(k), \tag{111}$$

$$\int_{\mathcal{O}(k)} U_{\mathcal{O}(k)}(\boldsymbol{Q})(d\boldsymbol{Q}) = \int_{\mathcal{O}(k)} \frac{1}{V_{\mathcal{O}(k)}}(d\boldsymbol{Q}) = \int_{\mathcal{O}(k)} [d\boldsymbol{Q}] = 1. \tag{112}$$

As $\mathrm{Gr}(k, D)$ is defined as a quotient manifold $\mathrm{St}(k, D)/\mathcal{O}(k)$ as in (2), the volume $V_{\mathrm{Gr}(k,D)}$ of $\mathrm{Gr}(k, D)$ can be defined as follows.

$$V_{\mathrm{Gr}(k,D)} = \int_{\mathrm{Gr}(k,D)} (d\boldsymbol{Y}) = \frac{V_{\mathrm{St}(k,D)}}{V_{\mathcal{O}(k)}} = \frac{V_{\mathrm{St}(k,D)}}{V_{\mathrm{St}(k,k)}} = \frac{\pi^{\frac{k(D-k)}{2}} \Gamma_k\left(\frac{k}{2}\right)}{\Gamma_k\left(\frac{D}{2}\right)}. \tag{113}$$

The measure normalized to be a probability measure is expressed as:

$$[d\boldsymbol{Y}] = \frac{1}{V_{\mathrm{Gr}(k,D)}}(d\boldsymbol{Y}). \tag{114}$$

### D.6 Retraction

In general, the points except the origin ($\boldsymbol{p}(0) = \boldsymbol{x}$) of the tangent space $T_{\boldsymbol{x}}\mathcal{M}$ at $\boldsymbol{x}$ on the manifold $\mathcal{M}$ are not elements on $\mathcal{M}$ ($\boldsymbol{p}(t) \in T_{\boldsymbol{x}}\mathcal{M}, t \neq 0$). Therefore, if the result of the operation on the tangent space is to be used at another point on the manifold $\mathcal{M}$, it is necessary to map $\boldsymbol{p}(t)$ to the manifold $\mathcal{M}$. The map from a tangent space to a manifold is referred to as an exponential map. However, because the exponential map is computationally expensive, retraction based on numerical linear algebra is often used as an alternative (Zhu & Sato (2021)). Retraction is a method for approximating an exponential map to first order while maintaining global convergence in optimization algorithms on Riemannian manifolds. The most commonly used retractions on $\mathrm{Gr}(k, D)$ are methods based on QR decomposition or singular-value decomposition (SVD) (Absil et al. (2008); Zhu & Sato (2021)). In addition, a retraction based on the Cayley transform is introduced in Zhu & Sato (2021). This retraction is closely related to the Cayley transform on $\mathrm{St}(k, D)$ (Wen & Yin (2013); Xiaojing (2017); Zhu & Duan (2019)) and the Projected polynomial retraction (Gawlik & Leok (2018a)).

#### D.6.1 Exponential Map and Retraction

Geodesics on $\mathrm{Gr}(k, D)$ can be expressed as the equivalence class $\left[\exp_{\boldsymbol{Y}}^{\mathrm{Gr}}(t\overline{\boldsymbol{\xi}}_{\boldsymbol{Y}})\right]$, where

$$\exp_{\boldsymbol{Y}}^{\mathrm{Gr}}(t\overline{\boldsymbol{\xi}}_{\boldsymbol{Y}}) = [\ \boldsymbol{Y} \quad \boldsymbol{Y}_\perp\ ] \exp(t\mathfrak{B}) \boldsymbol{I}_{D \times k}. \tag{115}$$

Here, exp on the right-hand side is the matrix exponential, and $\mathfrak{B} = \begin{bmatrix} \boldsymbol{0}_k & -\boldsymbol{B}^\top \\ \boldsymbol{B} & \boldsymbol{0}_{D-k} \end{bmatrix} \in \mathrm{skew}(D)$, where $\boldsymbol{B}$ satisfies $\overline{\boldsymbol{\xi}}_{\boldsymbol{Y}} = \boldsymbol{Y}_\perp \boldsymbol{B}$. We can use a following exponential map that is mathematically equivalent to (115) (Edelman et al. (1998)):

$$\exp_{\boldsymbol{Y}}^{\mathrm{Gr}}(t\overline{\boldsymbol{\xi}}_{\boldsymbol{Y}}) := \{\boldsymbol{Y}\boldsymbol{V}\cos(\boldsymbol{\Sigma} t) + \boldsymbol{U}\sin(\boldsymbol{\Sigma} t)\}\boldsymbol{V}^\top, \tag{116}$$

where $\boldsymbol{U}, \boldsymbol{\Sigma}, \boldsymbol{V}^\top = \mathrm{SVD}(\overline{\boldsymbol{\xi}}_{\boldsymbol{Y}})$.

Further, we can use the Padé approximation to approximate geodesics on Grassmann manifolds as follows:

$$\boldsymbol{Y}(t) = [\ \boldsymbol{Y} \quad \boldsymbol{Y}_\perp\ ] r_m(t\mathfrak{B}) \boldsymbol{I}_{D \times k} \approx \exp_{\boldsymbol{Y}}^{\mathrm{Gr}}(t\overline{\boldsymbol{\xi}}_{\boldsymbol{Y}}), \tag{117}$$

where $r_m(\boldsymbol{X})$ is the $m$th-order diagonal Padé approximation to the matrix exponential $\exp(\boldsymbol{X})$. See the expression of $r_m(\boldsymbol{X})$ in Moler & Loan (2003). The simplest member of this class is surely the first-order Padé approximation

$$\overline{R}_{\boldsymbol{Y}}(t\overline{\boldsymbol{\xi}}_{\boldsymbol{Y}}) := [\ \boldsymbol{Y} \quad \boldsymbol{Y}_\perp\ ] r_1(t\mathfrak{B}) \boldsymbol{I}_{D \times k} \tag{118}$$

$$= [\ \boldsymbol{Y} \quad \boldsymbol{Y}_\perp\ ] \left(\boldsymbol{I}_n - \frac{t}{2}\mathfrak{B}\right)^{-1} \left(\boldsymbol{I}_D + \frac{t}{2}\mathfrak{B}\right) \boldsymbol{I}_{D \times k}, \tag{119}$$

which is also known as the Cayley transform. From the error expression $\exp(\boldsymbol{Y}) = r_m(\boldsymbol{Y}) + O\left(\|\boldsymbol{Y}\|^{2m+1}\right)$ of the Padé approximation, we have

$$\overline{R}_{\boldsymbol{Y}}(t\overline{\boldsymbol{\xi}}_{\boldsymbol{Y}}) = \exp_{\boldsymbol{Y}}^{\mathrm{Gr}}(t\overline{\boldsymbol{\xi}}_{\boldsymbol{Y}}) + O\left(t^{2m+1} \|\overline{\boldsymbol{\xi}}_{\boldsymbol{Y}}\|^{2m+1}\right), \tag{120}$$

which is also given by Theorem 3 in Gawlik & Leok (2018b).

### D.6.2 Horizontal Retraction

From Definition 3 in Zhu & Sato (2021), (119) is a horizontal retraction, and

$$R_{[\boldsymbol{Y}]}\left(t\boldsymbol{\xi}_{[\boldsymbol{Y}]}\right) := \left[\overline{R}_{\boldsymbol{Y}}\left(t\overline{\boldsymbol{\xi}}_{\boldsymbol{Y}}^{\mathrm{h}}\right)\right] \tag{121}$$

is a retraction on $\mathrm{Gr}(k, D)$ as a quotient manifold defined by (2). This is because $\overline{R}$ satisfies the invariance condition that $\overline{R}_{\boldsymbol{Y}}\left(t\overline{\boldsymbol{\xi}}_{\boldsymbol{Y}}^{\mathrm{h}}\right) \sim \overline{R}_{\boldsymbol{Y}'}\left(t\overline{\boldsymbol{\xi}}_{\boldsymbol{Y}'}^{\mathrm{h}}\right)$ for all $\boldsymbol{Y} \in \mathrm{St}(k, D)$, $\boldsymbol{Y}' \in \mathrm{St}(k, D)$, $\overline{\boldsymbol{\xi}}_{\boldsymbol{Y}} \in T_{\boldsymbol{Y}}^{\mathrm{h}}\mathrm{St}(k, D)$ and $\overline{\boldsymbol{\xi}}_{\boldsymbol{Y}'} \in T_{\boldsymbol{Y}'}^{\mathrm{h}}\mathrm{St}(k, D)$ such that $\boldsymbol{Y} \sim \boldsymbol{Y}'$ and $\overline{\boldsymbol{\xi}}_{\boldsymbol{Y}}$ and $\overline{\boldsymbol{\xi}}_{\boldsymbol{Y}'}$ are horizontal lifts of $\boldsymbol{\xi}_{[\boldsymbol{Y}]} \in T_{[\boldsymbol{Y}]}\mathrm{Gr}(k, D)$ at $\boldsymbol{Y}$ and $\boldsymbol{Y}'$, respectively.

In low-rank cases, we can obtain an economical version of (119) as follows (Zhu & Sato (2021)).

$$\overline{R}_{\boldsymbol{Y}}\left(t\overline{\boldsymbol{\xi}}_{\boldsymbol{Y}}^{\mathrm{h}}\right) = \boldsymbol{Y} + t\overline{\boldsymbol{\xi}}_{\boldsymbol{Y}}^{\mathrm{h}} - \left(\frac{t^2}{2}\boldsymbol{Y} + \frac{t^3}{4}\overline{\boldsymbol{\xi}}_{\boldsymbol{Y}}^{\mathrm{h}}\right)\left(\boldsymbol{I}_k + \frac{t^2}{4}\overline{\boldsymbol{\xi}}_{\boldsymbol{Y}}^{\mathrm{h}\top}\overline{\boldsymbol{\xi}}_{\boldsymbol{Y}}^{\mathrm{h}}\right)^{-1}\overline{\boldsymbol{\xi}}_{\boldsymbol{Y}}^{\mathrm{h}\top}\overline{\boldsymbol{\xi}}_{\boldsymbol{Y}}^{\mathrm{h}}. \tag{122}$$

The inverse retraction $\left(\overline{R_{[\boldsymbol{Y}]}^{-1}}\right)_{\boldsymbol{Y}}^{\mathrm{h}} : \mathrm{St}(k, D) \to T_{\boldsymbol{Y}}^{\mathrm{h}}\mathrm{St}(k, D)$ of $\overline{R}_{\boldsymbol{Y}}\left(\overline{\boldsymbol{\xi}}_{\boldsymbol{Y}}^{\mathrm{h}}\right)$ is the following:

$$\left(\overline{R_{[\boldsymbol{Y}]}^{-1}\left([\boldsymbol{X}]\right)}\right)_{\boldsymbol{Y}}^{\mathrm{h}} = \overline{R}_{\boldsymbol{Y}}^{-1}\left(\boldsymbol{X}\right) \tag{123}$$

$$= 2\boldsymbol{Y}_\perp \boldsymbol{Y}_\perp^\top \boldsymbol{X}\left(\boldsymbol{I}_k + \boldsymbol{Y}^\top \boldsymbol{X}\right)^{-1} \tag{124}$$

$$= 2\left(\boldsymbol{X} - \boldsymbol{Y}\boldsymbol{Y}^\top \boldsymbol{X}\right)\left(\boldsymbol{I}_k + \boldsymbol{Y}^\top \boldsymbol{X}\right)^{-1}. \tag{125}$$

