# A   Appendix Overview

First, we present the details of the various proofs of Section 4 in Appendix B. Next, in Appendix C, we describe the network layers for the building GrCNF, and the detailed model architectures, hyperparameters, and implementation on the experiments. Finally, in Appendix D, we provide a summary of the fundamentals of a Grassmann manifold, which is the core concept of this study.

# B   Proofs

## B.1   Proposition 1

First, we invoked the following two corollaries.

**Corollary 1** (Diffeomorphism Invariance of Flows). *Let $F : \mathcal{M} \to \mathcal{N}$ be a diffeomorphism. If $X$ is a smooth vector field over $\mathcal{M}$ and $\theta$ is the flow of $X$, then the flow of $F_* X$[4] is $\eta_t = F \circ \theta_t \circ F^{-1}$, with domain $N_t = F(M_t)$ for each $t \in \mathbb{R}$.*

*Proof.* See Lee (2003, Corollary 9.14). □

**Corollary 2** (Homogeneity Property). *The horizontal lift $\overline{\xi}_{\boldsymbol{Y}}^{\mathrm{h}}$ at representative $\boldsymbol{Y} \in \mathrm{St}(k, D)$ relative to $\boldsymbol{\xi}_{[\boldsymbol{Y}]} \in T_{[\boldsymbol{Y}]} \mathrm{Gr}(k, D)$ satisfies the following homogeneity (equivariance) property (4) with regard to $^\forall \boldsymbol{Q} \in \mathcal{O}(k)$.*

$$\overline{\xi}_{\boldsymbol{Y}\boldsymbol{Q}}^{\mathrm{h}} = \overline{\xi}_{\boldsymbol{Y}}^{\mathrm{h}} \boldsymbol{Q}.$$

*Proof.* $\pi(\boldsymbol{Y}) = \pi(\boldsymbol{Y}\boldsymbol{Q})$ is true for $^\forall \boldsymbol{Y} \in \mathrm{St}(k, D), \boldsymbol{Q} \in \mathcal{O}(k)$. Therefore, $\pi(\boldsymbol{Y}) = (\pi \circ q)(\boldsymbol{Y})$ is true when defined as $q(\boldsymbol{Y}) = \boldsymbol{Y}\boldsymbol{Q}$. When the derivative $d\pi(\cdot)[\cdot]$ of both sides is applied to the horizontal lift $\overline{\xi}_{\boldsymbol{Y}}^{\mathrm{h}}$ of $\boldsymbol{\xi}_{[\boldsymbol{Y}]}$, the following is obtained:

$$d\pi(\boldsymbol{Y}) \left[\overline{\xi}_{\boldsymbol{Y}}^{\mathrm{h}}\right] = d(\pi \circ q)(\boldsymbol{Y}) \left[\overline{\xi}_{\boldsymbol{Y}}^{\mathrm{h}}\right] = d\pi(q(\boldsymbol{Y})) \left[dq(\boldsymbol{Y}) \left[\overline{\xi}_{\boldsymbol{Y}}^{\mathrm{h}}\right]\right] = d\pi(\boldsymbol{Y}\boldsymbol{Q}) \left[\overline{\xi}_{\boldsymbol{Y}}^{\mathrm{h}} \boldsymbol{Q}\right]. \quad (10)$$

Moreover, from (98) which is definition of horizontal lift, the following equation is true.

$$\boldsymbol{\xi}_{[\boldsymbol{Y}]} = d\pi(\boldsymbol{Y}) \left[\overline{\xi}_{\boldsymbol{Y}}^{\mathrm{h}}\right] = d\pi(\boldsymbol{Y}\boldsymbol{Q}) \left[\overline{\xi}_{\boldsymbol{Y}\boldsymbol{Q}}^{\mathrm{h}}\right]. \quad (11)$$

Subsequently, we obtain the following equation.

$$\boldsymbol{\xi}_{[\boldsymbol{Y}]} = d\pi(\boldsymbol{Y}\boldsymbol{Q}) \left[\overline{\xi}_{\boldsymbol{Y}\boldsymbol{Q}}^{\mathrm{h}}\right] = d\pi(\boldsymbol{Y}\boldsymbol{Q}) \left[\overline{\xi}_{\boldsymbol{Y}}^{\mathrm{h}} \boldsymbol{Q}\right]. \quad (12)$$

Finally, the uniqueness of the horizontal lift yields $\overline{\xi}_{\boldsymbol{Y}\boldsymbol{Q}}^{\mathrm{h}} = \overline{\xi}_{\boldsymbol{Y}}^{\mathrm{h}} \boldsymbol{Q}$. □

**Proposition 1.** *Let $\mathrm{Gr}(k, D)$ be a Grassmann manifold, $\mathsf{X}$ be any time-dependent vector field on $\mathrm{Gr}(k, D)$, and $F_{\mathsf{X},T}$ be a flow on a $\mathsf{X}$. Let $\overline{\mathsf{X}}$ be any time-dependent horizontal lift and $\overline{F}_{\overline{\mathsf{X}},T}$ be a flow of $\overline{\mathsf{X}}$. $\overline{\mathsf{X}}$ is a vector field on $\mathrm{St}(k, D)$ if and only if $\overline{F}_{\overline{\mathsf{X}},T}$ is a flow on $\mathrm{St}(k, D)$ and satisfies invariance condition $\overline{\mathsf{X}} \sim \overline{\mathsf{X}}'$ for all $\overline{F}_{\overline{\mathsf{X}},T} \sim \overline{F}_{\overline{\mathsf{X}}',T}$. Therefore, $\mathsf{X}$ is a vector field on $\mathrm{Gr}(k, D)$ if and only if $F_{\mathsf{X},T} := \left[\overline{F}_{\overline{\mathsf{X}},T}\right]$ is a flow on $\mathrm{Gr}(k, D)$, and vice versa.*

*Proof.* **Flow $F_{\mathsf{X},T}$ on $\mathbf{Gr}(k, D) \Rightarrow$ Vector Field $\mathsf{X}$ on $\mathbf{Gr}(k, D)$.** Let $\theta : \mathrm{St}(k, D) \times \mathcal{O}(k) \to \mathrm{St}(k, D), (\boldsymbol{Y}, \boldsymbol{Q}) \mapsto \boldsymbol{Y}\boldsymbol{Q}$ be a map representing the right action of the orthogonal group. In addition, let $F_{\mathsf{X},T}$ be a flow on $\mathrm{Gr}(k, D)$ and $\overline{F}_{\overline{\mathsf{X}},T}$ be a flow on $\mathrm{St}(k, D)$. These satisfy $\overline{F}_{\overline{\mathsf{X}}\boldsymbol{Q},T} \sim$

---

[4] $F_*$ denotes the pushforward, that is, another notation for the differential of $F$.

$\overline{F}_{\overline{X},T}, \overline{F}_{\overline{X}Q,T} \in F_{X,T}, \overline{F}_{\overline{X},T} \in F_{X,T}.$

$$\overline{X}\left(t, \overline{F}_{\overline{X}Q,t}\left(\boldsymbol{Y}\boldsymbol{Q}\right)\right) = \overline{X}\left(t, \overline{F}_{\overline{X},t}\left(\boldsymbol{Y}\right)\boldsymbol{Q}\right) \tag{13}$$

$$= \frac{d}{dt}\left\{\overline{F}_{\overline{X},t}\left(\boldsymbol{Y}\right)\boldsymbol{Q}\right\} \tag{14}$$

$$= \frac{d}{dt}\left(\theta \circ \overline{F}_{\overline{X},t}\right)\left(\boldsymbol{Y}\right) \tag{15}$$

$$= d(\theta)_{\boldsymbol{Y}}\left\{\frac{d}{dt}\overline{F}_{\overline{X},t}\left(\boldsymbol{Y}\right)\right\} \tag{16}$$

$$= d(\theta)_{\boldsymbol{Y}}\left\{\overline{X}\left(t, \overline{F}_{\overline{X},t}\left(\boldsymbol{Y}\right)\right)\right\} \tag{17}$$

$$= \overline{X}\left(t, \overline{F}_{\overline{X},t}\left(\boldsymbol{Y}\right)\right)\boldsymbol{Q}. \tag{18}$$

Thus, $\overline{X} \sim \overline{X}\boldsymbol{Q}$ is true. Therefore, $\overline{X}$ is the horizontal lift of the vector field $X$ on $\mathrm{Gr}(k, D)$ and is unique for $X$.

**Flow $F_{X,T}$ on $\mathrm{Gr}(k, D)$ $\Leftarrow$ Vector Field $X$ on $\mathrm{Gr}(k, D)$.** Let $\theta : \mathrm{St}(k, D) \times \mathcal{O}(k) \to \mathrm{St}(k, D), (\boldsymbol{Y}, \boldsymbol{Q}) \mapsto \boldsymbol{Y}\boldsymbol{Q}$ be a map representing the right action of the orthogonal group. In addition, let $\overline{X}$ be a vector field over a horizontal bundle $T^{\mathrm{h}}\,\mathrm{St}(k, D)$ on $\mathrm{St}(k, D)$ and $\overline{F}_{\overline{X},T}$ be its flow. From the Corollary 1,

$$\overline{F}_{\theta_* \circ \overline{X},T} = \theta \circ \overline{F}_{\overline{X},T} \circ \theta^{-1} \tag{19}$$

$$\overline{F}_{\theta_* \circ \overline{X},T} \circ \theta = \theta \circ \overline{F}_{\overline{X},T} \tag{20}$$

$$\overline{F}_{d(\theta)_{\boldsymbol{Y}}\overline{X},T} \circ \theta = \theta \circ \overline{F}_{\overline{X},T} \tag{21}$$

$$\overline{F}_{\overline{X}Q,T}\left(\boldsymbol{Y}\boldsymbol{Q}\right) = \overline{F}_{\overline{X},T}\left(\boldsymbol{Y}\right)\boldsymbol{Q}. \tag{22}$$

Note that $d(\theta)_{\boldsymbol{Y}}\overline{X} = \overline{X}\boldsymbol{Q}$ is derived from the Corollary 2 and (4) in Zhu & Sato (2021). This indicates that $\overline{F}_{\overline{X},T} \sim \overline{F}_{\overline{X}',T}$ is true for any $\overline{X}, \overline{X}' \in T^{\mathrm{h}}\,\mathrm{St}(k, D)$ that satisfies $\overline{X} \sim \overline{X}'$. Thus, a new flow can be defined as $F_{X,T} := \left[\overline{F}_{\overline{X},T}\right]$. This is a flow on a $\mathrm{Gr}(k, D)$. Because $\overline{X}$ is a vector field in a horizontal bundle $T^{\mathrm{h}}\,\mathrm{St}(k, D)$ on $\mathrm{St}(k, D)$, it is a horizontal lift of the vector field $X$ on $\mathrm{Gr}(k, D)$ and is therefore unique for $X$.

Thus, the proof is complete. $\qquad\square$

## B.2 Proposition 2

**Proposition 2.** *Let $\mathrm{Gr}(k, D)$ be a Grassmann manifold. Let $p$ be the probability density on $\mathrm{Gr}(k, D)$ and $F$ be the flow on $\mathrm{Gr}(k, D)$. Suppose $\overline{p}$ is a density on $\mathrm{St}(k, D)$ and $\overline{F}$ is a flow on $\mathrm{St}(k, D)$. Then, the distribution $\overline{p}_{\overline{F}}$ after transformations by $\overline{F}$ is also a density on $\mathrm{St}(k, D)$. Further, the invariance condition $\overline{p}_{\overline{F}} \sim \overline{p}_{\overline{F}'}$ is satisfied for all $\overline{F} \sim \overline{F}'$. Therefore, $p_F := [\overline{p}_{\overline{F}}]$ is a distribution on $\mathrm{Gr}(k, D)$.*

*Proof.* Let $\theta : \mathrm{St}(k, D) \times \mathcal{O}(k) \to \mathrm{St}(k, D) : (\boldsymbol{Y}, \boldsymbol{Q}) \mapsto \boldsymbol{Y}\boldsymbol{Q}$ be a map representing the right action of the orthogonal group.

$$\bar{p}_F (\theta \circ \boldsymbol{Y}) = \bar{p}_F (\theta \circ \boldsymbol{Y}) \frac{|\det \{J_\theta (\boldsymbol{Y})\}|}{|\det \{J_\theta (\boldsymbol{Y})\}|} = \frac{\bar{p}_{\theta^{-1} \circ F} (\boldsymbol{Y})}{|\det \{J_\theta (\boldsymbol{Y})\}|} \tag{23}$$

$$= \bar{p} \left( \left( F^{-1} \circ \theta \right) (\boldsymbol{Y}) \right) \frac{|\det \{J_{F^{-1} \circ \theta} (\boldsymbol{Y})\}|}{|\det \{J_\theta (\boldsymbol{Y})\}|} \tag{24}$$

$$= \left( \bar{p} \circ F^{-1} \right) \circ \theta (\boldsymbol{Y}) \frac{|\det \{J_{\theta \circ F^{-1}} (\boldsymbol{Y})\}|}{|\det \{J_\theta (\boldsymbol{Y})\}|} \tag{25}$$

$$= \theta \circ \left( \bar{p} \circ F^{-1} \right) (\boldsymbol{Y}) \frac{\left| \det \left\{ J_\theta \left( F^{-1} (\boldsymbol{Y}) \right) J_{F^{-1}} (\boldsymbol{Y}) \right\} \right|}{|\det \{J_\theta (\boldsymbol{Y})\}|} \tag{26}$$

$$= \theta \circ \left( \bar{p} \circ F^{-1} \right) (\boldsymbol{Y}) \frac{\left| \det \left\{ J_\theta \left( F^{-1} (\boldsymbol{Y}) \right) \right\} \right| \left| \det \{J_{F^{-1}} (\boldsymbol{Y})\} \right|}{|\det \{J_\theta (\boldsymbol{Y})\}|} \tag{27}$$

$$= \theta \circ \bar{p} \left( F^{-1} \right) (\boldsymbol{Y}) \left| \det \{J_{F^{-1}} (\boldsymbol{Y})\} \right| \frac{\left| \det \left\{ J_\theta \left( F^{-1} (\boldsymbol{Y}) \right) \right\} \right|}{|\det \{J_\theta (\boldsymbol{Y})\}|} \tag{28}$$

$$= \theta \circ \bar{p}_F (\boldsymbol{Y}) \frac{\left| \det \left\{ J_\theta \left( F^{-1} (\boldsymbol{Y}) \right) \right\} \right|}{|\det \{J_\theta (\boldsymbol{Y})\}|} \tag{29}$$

$$= \theta \circ \bar{p}_F (\boldsymbol{Y}), \tag{30}$$

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

(Y\right)\left[\bar{\xi}_Y^h\right] = \xi_{[Y]}$, of the tangent vector $\xi_{[Y]}$ at the point $[Y]$ on the Grassmann manifold. The proposed ODE solver maps and updates the tangent vector $\xi_{[Y]}$ at the point $[Y]$ onto the Grassmann manifold at each step using horizontal retraction (121). On the other hand, (b) the solver of Celledoni & Owren (2002), working on Stiefel manifolds, updates in each step by mapping the tangent vector $\bar{\xi}_Y$ at the point $Y$ onto the Stiefel manifold. In other words, the difference between the proposed ODE solver and the ODE solver on Stiefel manifolds is that the ODE solver on Stiefel manifolds works only on Stiefel manifolds, while the proposed ODE solver always updates in each step with the Stiefel manifold and Grassmann manifolds linked together.

ELBO $(P)$ can be decomposed as follows.

$$\text{ELBO}\left(P\right) = \log p_\psi(P) - D_{KL}\left(q_\phi(Y\,|\,P)||p_\psi(Y\,|\,P)\right) \tag{57}$$
$$= \mathbb{E}_{q_\phi(Y\,|\,P)}\left[\log p_\psi(P\,|\,Y)\right] - D_{KL}\left(q_\phi(Y\,|\,P)||p_\theta(Y)\right), \tag{58}$$

where $q_\phi(Y\,|\,P)$ is the inference model with parameter $\phi$, $p_\psi(P\,|\,Y)$ is the decoder model with parameter $\psi$, $p_\psi(Y\,|\,P)$ is the posterior distribution with parameter $\psi$, and $p_\theta(Y)$ is the prior distribution with parameter $\theta$. Further, $D_{KL}\left(q_\phi(Y\,|\,P)||p_\theta(Y)\right)$ can be formulated using differential entropy as follows.

$$D_{KL}\left(q_\phi(Y\,|\,P)||p_\theta(Y)\right) = -\mathbb{E}_{q_\phi(Y\,|\,P)}\left[p_\theta(Y)\right] - H\left[q_\phi(Y\,|\,P)\right]. \tag{59}$$

Thus, the final loss function is as follows.

$$\text{Loss} = -\text{ELBO}\left(P\right) \tag{60}$$
$$= -\mathbb{E}_{q_\phi(Y\,|\,P)}\left[\log p_\psi(P\,|\,Y)\right] - \mathbb{E}_{q_\phi(Y\,|\,P)}\left[p_\theta(Y)\right] - H\left[q_\phi(Y\,|\,

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

$$\exp_{\boldsymbol{Y}}^{\mathrm{Gr}}\left(t\overline{\boldsymbol{\xi}}_{\boldsymbol{Y}}\right) = [\ \boldsymbol{Y} \quad \boldsymbol{Y}_{\perp}\ ]\exp(t\mathfrak{B})\boldsymbol{I}_{D \times k}. \tag{115}$$

Here, $\exp$ on the right-hand side is the matrix exponential, and $\mathfrak{B} = \begin{bmatrix} \boldsymbol{0}_k & -\boldsymbol{B}^{\top} \\ \boldsymbol{B} & \boldsymbol{0}_{D-k} \end{bmatrix} \in \mathrm{skew}(D)$, where $\boldsymbol{B}$ satisfies $\overline{\boldsymbol{\xi}}_{\boldsymbol{Y}} = \boldsymbol{Y}_{\perp}\boldsymbol{B}$. We can use a following exponential map that is mathematically equivalent to (115) (Edelman et al. (1998)):

$$\exp_{\boldsymbol{Y}}^{\mathrm{Gr}}\left(t\overline{\boldsymbol{\xi}}_{\boldsymbol{Y}}\right) := \{\boldsymbol{Y}\boldsymbol{V}\cos(\boldsymbol{\Sigma}t) + \boldsymbol{U}\sin(\boldsymbol{\Sigma}t)\}\boldsymbol{V}^{\top}, \tag{116}$$

where $\boldsymbol{U}, \boldsymbol{\Sigma}, \boldsymbol{V}^{\top} = \mathrm{SVD}\left(\overline{\boldsymbol{\xi}}_{\boldsymbol{Y}}\right)$.

Further, we can use the Padé approximation to approximate geodesics on Grassmann manifolds as follows:

$$\boldsymbol{Y}(t) = [\ \boldsymbol{Y} \quad \boldsymbol{Y}_{\perp}\ ]r_m(t\mathfrak{B})\boldsymbol{I}_{D \times k} \approx \exp_{\boldsymbol{Y}}^{\mathrm{Gr}}\left(t\overline{\boldsymbol{\xi}}_{\boldsymbol{Y}}\right), \tag{117}$$

where $r_m(\boldsymbol{X})$ is the $m$th-order diagonal Padé approximation to the matrix exponential $\exp(\boldsymbol{X})$. See the expression of $r_m(\boldsymbol{X})$ in Moler & Loan (2003). The simplest member of this class is surely the first-order Padé approximation

$$\overline{R}_{\boldsymbol{Y}}\left(t\overline{\boldsymbol{\xi}}_{\boldsymbol{Y}}\right) := [\ \boldsymbol{Y} \quad \boldsymbol{Y}_{\perp}\ ]r_1(t\mathfrak{B})\boldsymbol{I}_{D \times k} \tag{118}$$

$$= [\ \boldsymbol{Y} \quad \boldsymbol{Y}_{\perp}\ ]\left(\boldsymbol{I}_n - \frac{t}{2}\mathfrak{B}\right)^{-1}\left(\boldsymbol{I}_D + \frac{t}{2}\mathfrak{B}\right)\boldsymbol{I}_{D \times k}, \tag{119}$$

which is also known as the Cayley transform. From the error expression $\exp(\boldsymbol{Y}) = r_m(\boldsymbol{Y}) + O\left(\|\boldsymbol{Y}\|^{2m+1}\right)$ of the Padé approximation, we have

$$\overline{R}_{\boldsymbol{Y}}\left(t\overline{\boldsymbol{\xi}}_{\boldsymbol{Y}}\right) = \exp_{\boldsymbol{Y}}^{\mathrm{Gr}}\left(t\overline{\boldsymbol{\xi}}_{\boldsymbol{Y}}\right) + O\left(t^{2m+1}\|\overline{\boldsymbol{\xi}}_{\boldsymbol{Y}}\|^{2m+1}\right), \tag{120}$$

which is also given by Theorem 3 in Gawlik & Leok (2018b).

### D.6.2 Horizontal Retraction

From Definition 3 in Zhu & Sato (2021), (119) is a horizontal retraction, and

$$R_{[\boldsymbol{Y}]}\left(t\boldsymbol{\xi}_{[\boldsymbol{Y}]}\right) := \left[\overline{R}_{\boldsymbol{Y}}\left(t\overline{\boldsymbol{\xi}}_{\boldsymbol{Y}}^{\mathrm{h}}\right)\right] \tag{121}$$

is a retraction on $\mathrm{Gr}(k,D)$ as a quotient manifold defined by (2). This is because $\overline{R}$ satisfies the invariance condition that $\overline{R}_{\boldsymbol{Y}}\left(t\overline{\boldsymbol{\xi}}_{\boldsymbol{Y}}^{\mathrm{h}}\right) \sim \overline{R}_{\boldsymbol{Y}'}\left(t\overline{\boldsymbol{\xi}}_{\boldsymbol{Y}'}^{\mathrm{h}}\right)$ for all $\boldsymbol{Y} \in \mathrm{St}(k,D)\,, \boldsymbol{Y}' \in \mathrm{St}(k,D)\,, \overline{\boldsymbol{\xi}}_{\boldsymbol{Y}} \in T_{\boldsymbol{Y}}^{\mathrm{h}}\,\mathrm{St}(k,D)$ and $\overline{\boldsymbol{\xi}}_{\boldsymbol{Y}'} \in T_{\boldsymbol{Y}'}^{\mathrm{h}}\,\mathrm{St}(k,D)$ such that $\boldsymbol{Y} \sim \boldsymbol{Y}'$ and $\overline{\boldsymbol{\xi}}_{\boldsymbol{Y}}$ and $\overline{\boldsymbol{\xi}}_{\boldsymbol{Y}'}$ are horizontal lifts of $\boldsymbol{\xi}_{[\boldsymbol{Y}]} \in T_{[\boldsymbol{Y}]}\,\mathrm{Gr}(k,D)$ at $\boldsymbol{Y}$ and $\boldsymbol{Y}'$, respectively.

In low-rank cases, we can obtain an economical version of (119) as follows (Zhu & Sato (2021)).

$$\overline{R}_{\boldsymbol{Y}}\left(t\overline{\boldsymbol{\xi}}_{\boldsymbol{Y}}^{\mathrm{h}}\right) = \boldsymbol{Y} + t\overline{\boldsymbol{\xi}}_{\boldsymbol{Y}}^{\mathrm{h}} - \left(\frac{t^2}{2}\boldsymbol{Y} + \frac{t^3}{4}\overline{\boldsymbol{\xi}}_{\boldsymbol{Y}}^{\mathrm{h}}\right)\left(\boldsymbol{I}_k + \frac{t^2}{4}\overline{\boldsymbol{\xi}}_{\boldsymbol{Y}}^{\mathrm{h}\top}\overline{\boldsymbol{\xi}}_{\boldsymbol{Y}}^{\mathrm{h}}\right)^{-1}\overline{\boldsymbol{\xi}}_{\boldsymbol{Y}}^{\mathrm{h}\top}\overline{\boldsymbol{\xi}}_{\boldsymbol{Y}}^{\mathrm{h}}. \tag{122}$$

The inverse retraction $\left(\overline{R_{[\boldsymbol{Y}]}^{-1}}\right)_{\boldsymbol{Y}}^{\mathrm{h}} : \mathrm{St}(k,D) \to T_{\boldsymbol{Y}}^{\mathrm{h}}\,\mathrm{St}(k,D)$ of $\overline{R}_{\boldsymbol{Y}}\left(\overline{\boldsymbol{\xi}}_{\boldsymbol{Y}}^{\mathrm{h}}\right)$ is the following:

$$\left(\overline{R_{[\boldsymbol{Y}]}^{-1}\left([\boldsymbol{X}]\right)}\right)_{\boldsymbol{Y}}^{\mathrm{h}} = \overline{R}_{\boldsymbol{Y}}^{-1}\left(\boldsymbol{X}\right) \tag{123}$$

$$= 2\boldsymbol{Y}_{\perp}\boldsymbol{Y}_{\perp}^{\top}\boldsymbol{X}\left(\boldsymbol{I}_k + \boldsymbol{Y}^{\top}\boldsymbol{X}\right)^{-1} \tag{124}$$

$$= 2\left(\boldsymbol{X} - \boldsymbol{Y}\boldsymbol{Y}^{\top}\boldsymbol{X}\right)\left(\boldsymbol{I}_k + \boldsymbol{Y}^{\top}\boldsymbol{X}\right)^{-1}. \tag{125}$$