# OpenReview forum: "Grassmann Manifold Flows for Stable Shape Generation"
_NeurIPS.cc/2023/Conference — NeurIPS 2023 poster_

### Official Review · Reviewer_D7Hq · 2023-07-02

**Soundness:** 4 excellent
**Presentation:** 3 good
**Contribution:** 3 good
**Rating:** 6
**Confidence:** 3

**Summary:**

This paper proposes a normalizing flow approach applicable to generate data on Grassmann manifold. The proposed method utilize the fact that Grassmann manifold is a quotient space of Stiefel manifold on which numerical calculation is feasible. Specifically, the method first constructs the flow from vector field and then transform a probabilistic density using the flow. The vector field is learned through a neural network. A prior probabilistic density is defined as a matrix-variate Gaussian distribution on Grassmann manifold. The experiments are conducted on three different types of datasets among which the first one is synthetic and the last one is molecular point clouds. The experimental results show that the proposed GrCNF can generate the distribution on Grassmann manifold.

**Strengths:**

This paper aims to resolve the problem of shape generation when the high-dimensional shapes intrinsically embed in a low-dimensional manifold, e.g., Grassmann manifold, using normalizing flow (NF). Extending NF from common Euclidean space to non-linear manifold is novel and meaningful, as the manifold hypothesis assumes that most high-dimensional data usually embeds in a low-dimensional manifold. The theoretical part looks sound. Experiments under three different settings verify the effectiveness of the proposed GrCNF to learn distributions on Grassmann manifold.

**Weaknesses:**

The main weakness (also my concern) is, as also mentioned by the authors in Line 325 - 328, whether the proposed GrCNF is applicable to  high-dimensional data? E.g., point clouds containing thousands of points? If yes, it will make the paper very strong in both theoretical and application aspects. (I also want to know what real-world high-dimensional data embeds in Grassmann manifold?) If not applicable to high-dimensional data, it will weaken the paper.

**Questions:**

In general:

1. What's the meaning of "stable shape" in the title? Why does the generation process is stable?

Method parts:

2. In Figure 1, what's the relation between T^v_YSt(k, D) and T^h_YSt(k, D) and why do we use T^h_Y, not use T^v_Y?

3. In Line 106, it mentions that "we use the representative Y ...", so how to choose the representative Y?

4. In Proposition 1, what's the relation between a vector field and a flow? I.e., how to compute a flow F_{X, T} given a vector field X?

5. Why do we need retraction in Eq.(4) in Proposition 3?

6. In Section 5.3 and Figure 2. does the output (through a neural network) is ensured in the tangent space of T^h_YSt(k, D) theoretically?

7. In Section 5.4, what's the goal of the ODE solver? Could you use one or two sentences to summarize it.

Experiment parts:

8. In the experiment in Section 6.1, I'm not sure why the spiral (and swissroll and circle ...) are 1-dimensional data in R^3, as they contain x and y two coordinates.

9. In the experiment in Section 6.2 and 6.3, is it possible to visualize the generated data, similar to Figure 3.

Implementation parts:

10. As the method is complicated in theory, will the code be released?





**Limitations:**

The authors mentions the two limitations of the proposed method in Line 324 - 328.

---

> ### Author Rebuttal · Authors · 2023-08-07
>
> Thank you for the detailed comments and positive feedback indicating that the presented framework is non-trivial and novel!
> We have prepared detailed responses to your questions below.
> We sincerely hope that these responses will address and alleviate any concerns.
>
> ****
> > **Question:1**
>
> When we refer to "Stable," we mean that our approach ensures the complete elimination of the possibility of unlearned orthogonal-transformed data in the trained model.
> As a result, it guarantees the generation of data after any orthogonal transformations during the data generation process.
> In this context, stability implies the robustness of the model to handle various orthogonal transformations and produce accurate data generation results.
>
> ****
> > **Question:2**
>
> As stated in (94) of the Appendix, the vertical and horizontal spaces are orthogonal to each other.
> Additionally, the tangent space on the Stiefel manifold, as shown in (96), is a direct sum of the vertical and horizontal spaces.
> The vertical space represents the components of $\mathcal{O}\left(k\right)$, while the horizontal space, orthogonal to the vertical space, realizes the invariance with respect to $\mathcal{O}\left(k\right)$ (as shown in (9) of the Appendix).
> Utilizing only the horizontal space is equivalent to manipulating points on the Grassmann manifold.
> For detailed mathematical discussions on this matter, please refer to Appendix D.4.3.
> **We have uploaded a PDF containing this figure in the Author Rebuttal (global response).
> Please refer to Figure 1&2 in the Author Rebuttal's PDF.**
>
> ****
> > **Question:3**
>
> We have the option to set the data within the dataset as representatives.
> In the ODE, each step is updated from the initial value $\boldsymbol{Y}$, and by applying the horizontal retraction at each step of the algorithm, we always obtain specific representative $\boldsymbol{Y}$ on the computer alongside their corresponding theoretical equivalence class $\left[\boldsymbol{Y}\right]$.
> Theoretically, any element of $\left[\boldsymbol{Y}\right]$ can be chosen as a representative.
>
> ****
> > **Question:4**
>
> A flow refers to the continuous movement of points over time.
> The direction of motion at each point is determined by the vector field X assigned to that point.
> Therefore, a flow is essentially a mapping determined by the vector field.
> In our proposed method, the flow is generated using ODE solver with numerous time steps.
> In other words, the flow can be viewed as the ODE solver itself, which utilizes the vector field to solve the differential equation.
>
> ****
> > **Question:5**
>
> The reason for requiring retraction in (4) is that when sampling from this distribution on a computer, it involves traversing the horizontal space.
> The desired sample point is the representative $\boldsymbol{Y}$ on the Grassmann manifold represented by the point $\left[\boldsymbol{Y}\right]$.
> Therefore, a mapping (retraction) from the horizontal space to the Stiefel manifold, where the representative exists, is necessary.
> The primary reason for passing through the horizontal space lies in the theoretical concept of horizontal lift, which allows us to represent the Grassmann manifold.
> However, intuitively, using the horizontal space simplifies the process of converting the matrix-variate gaussian distribution on the easier-to-sample Euclidean space into the representative $\boldsymbol{Y}$. (Please refer to Algorithm 1 for more details).
> For a more theoretical derivation, please see the proof of Proposition 3 in Appendix B.3.
> Additionally, We have uploaded a PDF containing this figure in the Author Rebuttal (global response).
> **Please refer to Figure 2 in the Author Rebuttal's PDF.**
>
> ****
> > **Question:6**
>
> The proposed ODE solver uses retraction to map the variables to the horizontal space after each ODE step.
> This ensures that the output at each ODE step exists in the horizontal space.
> We have uploaded a PDF containing figures related to your question in the global response.
> **Please refer to Figure 2 in the Author Rebuttal's PDF.**
>
> ****
> > **Question:7**
>
> The objective of our ODE solver is to obtain the value of $\gamma$ as given in (7).
> To achieve this, the ODE in (7) is solved using our ODE solver, and the resulting $\epsilon$ is mapped to obtain $\gamma$ through retraction.
>
> ****
> > **Question:8**
>
> The spiral shown in Figure 3 itself is not one-dimensional data. Instead, each individual white point within the spiral represents one-dimensional data.
> In a three-dimensional space, the one-dimensional data is visualized as a thin cyan line connecting the blue point and the origin, as depicted by the leftmost sphere in Figure 4.
> Since one-dimensional data lies on a line, it includes its antipodal points (i.e., points obtained through orthogonal transformations).
> The advantage of learning on the Grassmann manifold is that such antipodal points are learned without the need for data augmentation.
> For visualization purposes, drawing numerous lines passing through the origin might clutter the image, so in Figure 3, we represent the one-dimensional data as white points.
>
> ****
> > **Question:9**
>
> We have uploaded a PDF containing this figure in the Author Rebuttal (global response).
> **Please refer to Figure 3 in the Author Rebuttal's PDF.**
>
> ****
> > **Question:10**
>
> We apologize for any inconvenience, but according to our organization's policies, we do not have plans to release the code to the public.
> Instead, we have provided specific components that will allow you to fully reproduce our results from our paper.
> These components include a concrete sampling algorithm from the prior distribution (Algorithm 1), detailed network architectures (Table 3), ODE solver (7), specific loss functions ((56), (57), and original CNF implementations), comprehensive experimental conditions (Appendix C.3), and details of the data generation method (Listing 1).
> With access to these components, we believe you will be able to completely replicate our paper's findings.

---

> > ### Comment · Reviewer_D7Hq · 2023-08-19
> >
> > Thanks for the detailed response from the authors. After carefully reading the authors' rebuttal (to my review and other reviews), I think it addressed my concerns.
> >
> > (+) As the authors promised to include the experiment results for ShapeNet (response to Reviewer ioxc) in a camera-ready version, it will strengthen the paper.
> >
> > (-) As the authors cannot release the code (response to Question 10), it will weaken the paper.
> >
> > As mentioned by Reviewer 5FfM, it is also highly suggested to highlight the difference between the proposed method with [51].
> >
> > So I keep my original rating.

---

> > > ### Author Response · Authors · 2023-08-20
> > > **Thank you.**
> > >
> > > We sincerely appreciate the time you have taken to provide us with valuable feedback.
> > > Yes, we will make sure to highlight the distinctions from [51] within our Camera-ready version.
> > > We are truly grateful for your excellent advice.

---

### Official Review · Reviewer_5FfM · 2023-07-06

**Soundness:** 2 fair
**Presentation:** 2 fair
**Contribution:** 2 fair
**Rating:** 4
**Confidence:** 3

**Summary:**

This paper presents the application of continuous normalizing flow (CNF) to the Grassmann manifold for shape generation. The authors establish the theoretical foundations for learning distributions on the Grassmann manifold using CNF, enabling the generation of stable shapes. The proposed approach offers a promising generative model capable of generating subspace data.

**Strengths:**

The study represents the first exploration of applying a continuous normalizing flow (CNF) to the Grassmann manifold for shape generation.

The experiments conducted demonstrate the effectiveness of the proposed approach in generating promising shapes, particularly subspace data with low dimensions.

**Weaknesses:**

I have concerns regarding the significant contribution of the paper to the domain, given the existence of [51]. To address these concerns, it would be beneficial for the authors to include a comparative analysis with [51] in the experiments to demonstrate any potential superiority of the proposed method.

Furthermore, it is not clear how the proposed approach technically differs from the Stiefel-based ODE [7]. To provide a clearer understanding, it would be helpful if the authors could provide an intuitive illustration of the transfer from the Stiefel-based ODE to the Grassmann-based ODE.

Additionally, the paper's validation is limited to low-dimensional data, and there is a lack of empirical study on real-world, high-dimensional data such as point clouds. As a result, the current validation may not be sufficiently convincing to establish the effectiveness of the proposed approach in practical scenarios. Conducting experiments on high-dimensional data, such as point clouds, would enhance the credibility and generalizability of the proposed method.

**Questions:**

There should indeed be a comma rather than a period before finishing the equation, as seen in Eq. 8 on lines 232-233.

Regarding the performances for different D and k values on DWR, LJ13, and QM9, it would be valuable for the paper to provide an analysis and evaluation of the proposed method's performance in these scenarios. This would enhance the understanding of the method's effectiveness across different datasets and settings.

The reason why the proposed GrCNF outperforms the counterpart methods in Table 1 should be clarified in the paper. It would be beneficial for the authors to provide an explanation or analysis that highlights the specific advantages or characteristics of the proposed method that contribute to its superior performance.

Regarding the stability of shape generation in the proposed method compared to other methods, the paper should provide a clearer explanation to address this question. It would be helpful for the authors to discuss the specific mechanisms or techniques employed in the proposed method that lead to more stable shape generation results.

The feasibility and necessity of bringing invertibility to normalizing flows could be an interesting aspect to explore. It would be valuable for the authors to discuss the potential benefits and drawbacks of incorporating invertibility into the proposed method, considering the existing literature and the specific requirements and objectives of the task at hand.

**Limitations:**

The paper does not provide a thorough comparison with existing methods, such as [51], to demonstrate the superiority of the proposed approach. A comparative analysis would have helped assess the advancements of the proposed method over previous work.

The paper lacks empirical studies on real-world, high-dimensional data, such as point clouds. This limitation restricts the generalizability and applicability of the proposed approach to practical scenarios, as the effectiveness of the method has only been demonstrated on low-dimensional data.

It is not adequately explained how the proposed approach technically differs from the Stiefel-based ODE [7]. A clearer illustration of the transfer from Stiefel-based ODE to Grassmann-based ODE would have provided a better understanding of the method's technical contributions.

While the paper claims that the proposed GrCNF outperforms counterpart methods, it does not provide a comprehensive explanation for this superiority. It would be beneficial to discuss and analyze the specific factors or characteristics of the proposed method that contribute to its improved performance.

---

> ### Author Rebuttal · Authors · 2023-08-07
>
> Thank you for your detailed comments and valuable feedback! We have prepared detailed responses to your questions below. We sincerely hope that these responses will address and alleviate any concerns.
>
> ****
> > **Weakness:1. I have concerns regarding the significant contribution of the paper to the domain, given the existence of [51]. ...**
>
> Typically, density estimation is performed in Euclidean spaces (e.g., GAN, VAE).
> Geometric deep learning has recently aimed to extend these traditional Euclidean methods to the realm of Riemannian manifolds, with a representative approach found in [51].
> However, [51] only presents a highly general framework that includes Grassmann manifolds but does not address any practical methodologies for computations on the Grassmann manifold.
> In particular, it does not maintain symmetry. In fact, constructing models on the Grassmann manifold requires extensive discussions on how to represent subspace data on computers and ensure model symmetry.
> To tackle these issues, we have provided rigorous proofs in Proposition 1 to 3.
>
> ****
> > **Weakness:2. Furthermore, it is not clear how the proposed approach technically differs from the Stiefel-based ODE [7]. To provide a clearer understanding, it would be helpful if the authors could provide an intuitive illustration of the transfer from the Stiefel-based ODE to the Grassmann-based ODE.**
>
> **We have uploaded a PDF containing this figure in the Author Rebuttal (global response).
> Please refer to Figure 2 in the Author Rebuttal's PDF.**
> The most significant difference lies in the constraint of the vector field to the horizontal space, causing the ODE solver to operate while transitioning between the horizontal space and the Stiefel manifold at each time step.
> Proposition 1 to 3 ensures that the updates at each time step are simultaneously updated on the Grassmann manifold.
>
> ****
> > **Weakness:3. Additionally, the paper's validation is limited to low-dimensional data, and there is a lack of empirical study on real-world, high-dimensional data such as point clouds. As a result, the current validation may not be sufficiently convincing to establish the effectiveness of the proposed approach in practical scenarios...**
>
> Thank you for your valuable feedback. We wholeheartedly agree with your assertion that evaluation is indeed necessary.
> As a response to your feedback, we are currently planning to evaluate our method using the ShapeNet[82].
> However, due to the short two-week timeframe, conducting a comprehensive evaluation using such datasets is challenging because of time constraints.
> Thus, the results of these experiments will be included in the Camera-ready version of the paper.
> Finally, we would like to emphasize that this paper primarily focuses on establishing fundamental theories.
> Within our algorithms, we have introduced theoretical guarantees, laying the foundation for the next step, which involves conducting large-scale experiments.
>
> [82] Angel X. et al., ShapeNet: An Information-Rich 3D Model Repository. arXiv, 2015.
>
> ****
> > **Question:1. There should indeed be a comma rather than a period before finishing the equation, as seen in Eq. 8 on lines 232-233.**
>
> Thank you for your feedback. We have made the necessary updates. You will be able to see it in the Camera-ready version.
>
> ****
> > **Question:2. Regarding the performances for different D and k values on DWR, LJ13, and QM9, it would be valuable for the paper to provide an analysis and evaluation of the proposed method's performance in these scenarios. ...**
>
> Thank you for your valuable input.
> We believe that the experiments conducted with the DWR, LJ13, and QM9 datasets, each having different $k$ and $D$ values, have confirmed the fundamental characteristics of our method.
> This is supported by the fact that our approach consistently outperforms the conventional methods across all these datasets with varying $k$ and $D$ values.
> Particularly noteworthy is the significantly improved performance compared to GNF, which requires data augmentation.
> We acknowledge your concerns regarding the behavior of GrCNF with $k > 3$, and we fully agree that a detailed investigation is necessary.
> The datasets used in our current experiments are limited to $k \leq 3$, which means addressing the challenges you mentioned requires the selection of new datasets.
> As one of our future directions, we plan to include such evaluations in our analysis.
> We appreciate your feedback, and we are committed to enhancing the comprehensiveness of our research.
>
> ****
> > **Question:3. The reason why the proposed GrCNF outperforms the counterpart methods in Table 1 should be clarified in the paper.**
>
> In shape theory, a shape refers to the equivalence class of all configurations obtained by specific transformations (e.g., affine transformations) applied to a single base shape.
> GrCNF treats these equivalence classes as data for learning and generation.
> Affine transformations involve rotations, scalings, reflections, and shears, and learning in the space affected by these actions is very challenging from a combinatorial perspective.
> On the other hand, GrCNF achieves learning that entirely ignores the actions of affine transformations, thanks to the invariance provided by orthogonal transformations to orthonormal basis matrices.
> This capability allows GrCNF to overcome such difficulties, leading us to believe that in evaluations, GrCNF can exhibit higher performance compared to other traditional methods.
>
> ****
> > **Question:4. Regarding the stability of shape generation in the proposed method compared to other methods, the paper should provide a clearer explanation to address this question. ...**
>
> Similar to the previous question, the proposed method ensures stability by completely eliminating the possibility of unlearned orthogonal-transformed data in the trained model.

---

### Official Review · Reviewer_SWNf · 2023-07-07

**Soundness:** 3 good
**Presentation:** 4 excellent
**Contribution:** 4 excellent
**Rating:** 6
**Confidence:** 3

**Summary:**

This paper extends the previous Riemannian CNF to subspaces, i.e., Grassmann Manifold Gr(k,D), and tackle stable shape generation with nice transformation-invariant property. They derive the first and full mathematical formulation for learning invariant densities and flow on a Grassmann manifold. They successfully bring the complete model to life by using Stiefel Manifold to help construct the vector field effectively, with adapted ODE solver on Gr(k, D). They have also shown the superior performance of the proposed method on both toy and well-adopted public benchmarks compared to several existing seminar works.

**Strengths:**

[**originality**] the proposed work is the first one to explore continuous normalizing flow on subspace on the Grassmann Manifold. They not only creatively combine the success of quotient manifold for shape analysis, and the recent Riemannian CNF, but also contributed an elegant way of constructing the vector field through the connection to Stiefel manifold with horizontal lift.

[**quality & clarity**] The writing quality is over the bar, with clean explanation on the motivation, intuition, and design. As a person who is not that familiar with manifold theory, I found that the mathematical derivation is straight-forward to understand. They have presented complete and detailed propositions and proofs throughout the main paper and supp.

[**significance**]  The proposed framework is a pioneer work on exploring continuous normalizing flow on subspaces with nice equivariance-preserving property. Compared to existing baselines, this method is achieving SOTA performance on simple shape generation. It has great potential to benefit 3D molecules design, other high-dimensional geometry data generation including common objects.

**Weaknesses:**

1. From Figure 2, all that has been showing is on the Stiefel Manifold, without showing how it is connected to the target Grassmann manifold, which is confusing and not informative. Without this connection, I am not clear how the nice O(k)-invariant output v_out is achieved. The author might want to update the figure, or provide a better clarification;

2. As the author also mentions, it will make the proposed work more significant if showing experimental results on real-world object shapes.   More importantly, I am interested in knowing the major challenges to use the developed framework to handle such object shape data;

3. On the public benchmarks, the quantitative numbers are better than the baselines, it will make the comparison more complete if we can further show the visualizations of the generated samples;

**Questions:**

1. I am also wondering about the computation overhead, besides the expensive ODE solver, how many steps does it take to generate the final sample from a Gaussian density?

2. When comparing with the baselines, the proposed method will instead take the orthonormalized data, I am wondering whether this is a fair comparison setting.

---

> ### Author Rebuttal · Authors · 2023-08-07
>
> Thank you for the detailed comments and positive feedback indicating that the presented framework is non-trivial and novel!
> We have prepared detailed responses to your questions below.
> We sincerely hope that these responses will address and alleviate any concerns.
>
> ****
> > **Weakness:1. From Figure 2, all that has been showing is on the Stiefel Manifold, without showing how it is connected to the target Grassmann manifold, which is confusing and not informative. Without this connection, I am not clear how the nice O(k)-invariant output v_out is achieved. The author might want to update the figure, or provide a better clarification.**
>
> **We have uploaded a PDF containing this figure in the Author Rebuttal (global response).
> Please refer to Figure 1 in the Author Rebuttal's PDF.**
>
> ****
> > **Weakness:2. As the author also mentions, it will make the proposed work more significant if showing experimental results on real-world object shapes. More importantly, I am interested in knowing the major challenges to use the developed framework to handle such object shape data.**
>
> Thank you for your valuable feedback. We wholeheartedly agree with your assertion that evaluation is indeed necessary.
> As a response to your feedback, we are currently planning to evaluate our method using the ShapeNet[82].
> However, due to the short two-week timeframe, conducting a comprehensive evaluation using such datasets is challenging because of time constraints.
> Thus, the results of these experiments will be included in the Camera-ready version of the paper.
> Finally, we would like to emphasize that this paper primarily focuses on establishing fundamental theories.
> Within our algorithms, we have introduced theoretical guarantees, laying the foundation for the next step, which involves conducting large-scale experiments.
>
> [82] Angel X. et al., ShapeNet: An Information-Rich 3D Model Repository. arXiv, 2015.
>
> ****
> > **Weakness:3. On the public benchmarks, the quantitative numbers are better than the baselines, it will make the comparison more complete if we can further show the visualizations of the generated samples.**
>
> We have uploaded a PDF containing this figure in the Author Rebuttal (global response).
> **Please refer to Figure 3 in the Author Rebuttal's PDF.**
>
> ****
> > **Question:1. I am also wondering about the computation overhead, besides the expensive ODE solver, how many steps does it take to generate the final sample from a Gaussian density?**
>
> In our experiments, the number of steps for the trained models is as follows: 150 steps for Experiment 6.1, 160 steps for Experiment 6.2, and 220 steps for Experiment 6.3.
>
> ****
> > **Question:2. When comparing with the baselines, the proposed method will instead take the orthonormalized data, I am wondering whether this is a fair comparison setting.**
>
> We believe that we have constructed the experiments to be fair and equitable.
> In Experiment 6.2, all the comparative methods are evaluated using data transformed into orthonormal basis matrices.
> Moreover, in Experiment 6.3, by incorporating scale information into the ELBO computation in GrCNF, we demonstrated that even the ELBO surpasses traditional methods.
> These measures ensure that the comparison conditions are properly aligned for conducting the experiments.

---

### Official Review · Reviewer_ioxc · 2023-07-07

**Soundness:** 4 excellent
**Presentation:** 4 excellent
**Contribution:** 4 excellent
**Rating:** 8
**Confidence:** 3

**Summary:**

The paper presents a novel concept: Continuous Normalizing Flows on a Grassmann Manifold (GrCNF), a model that presents a more efficient and accurate way to capture and represent the geometric features of data. The unique proposition of GrCNF is its application of the Grassmann Manifold to grasp data equivariance, a key aspect that delineates the transformation response of data. GrCNF's efficacy is demonstrated via several experiments. The results affirm its capability to accurately generate data that depict complex geometric shapes, and outperform current Normalizing Flow models in metrics like log-likelihood and ELBO for diverse test datasets.

**Strengths:**

+ GrCNF is a novel model to bring continuous normalizing flows (CNF) into the field of Grassmann manifolds, expanding its applications to data manifolds with geometric structures.
+ GrCNF showed superior performance when compared to existing Normalizing Flow models in experiments with different datasets. It not only generated high-quality samples but also effectively learned the distribution on Gr(k, D).
+ GrCNF is able to handle subspace data, making it a versatile tool for handling various types of complex data.


**Weaknesses:**

1. The use of the Kronecker product at each ODE step is a demanding process, which suggests potential improvements in computational efficiency could be beneficial. This exploration could lead to exciting developments in scalability and applicability.
2. The study primarily focused on lower-dimensional data, and an examination of the GrCNF model's performance with higher-dimensional data could provide additional valuable insights.


**Questions:**

 In the artificial texture experiments, the antipodal points were generated using the trained GrCNF. While this approach demonstrates the learning capabilities of the model, it does not guarantee the accurate representation of all antipodal points in different contexts or datasets.

**Limitations:**

1. The proposed GrCNF has been tested on a limited variety of data types. Further exploration is required to understand its potential application and effectiveness on different data types, such as image, audio, and textual data.
2. In the experiments, GrCNF is mainly compared with existing normalizing flows methods. The inclusion of other machine learning models, such as generative adversarial networks (GANs) or autoencoders, could have provided a broader perspective on GrCNF's performance.

---

> ### Author Rebuttal · Authors · 2023-08-07
>
> Thank you for the detailed comments and positive feedback indicating that the presented framework is non-trivial and novel!
> We have prepared detailed responses to your questions below.
> We sincerely hope that these responses will address and alleviate any concerns.
>
> ****
> > **Question:1. In the artificial texture experiments, the antipodal points were generated using the trained GrCNF. While this approach demonstrates the learning capabilities of the model, it does not guarantee the accurate representation of all antipodal points in different contexts or datasets.**
>
> Our model can guarantee this property by (9) in the Appendix.
> The (9) represents the invariance of the model, ensuring that data transformed by the orthogonal group $\mathcal{O}\left(k\right)$, i.e., any reflection of the data, is learned without the need for data augmentation.
>
> ****
> > **Limitation:1. The proposed GrCNF has been tested on a limited variety of data types. Further exploration is required to understand its potential application and effectiveness on different data types, such as image, audio, and textual data.**
>
> and
>
> > **Limitation:2. In the experiments, GrCNF is mainly compared with existing normalizing flows methods. The inclusion of other machine learning models, such as generative adversarial networks (GANs) or autoencoders, could have provided a broader perspective on GrCNF's performance.**
>
> Thank you for your valuable feedback.
> We wholeheartedly agree with your assertion that evaluation is indeed necessary.
> As a response to your feedback, we are currently planning to evaluate our method using the ShapeNet[82].
> However, due to the short two-week timeframe, conducting a comprehensive evaluation using such datasets is challenging because of time constraints.
> Thus, the results of these experiments will be included in the Camera-ready version of the paper.
> Finally, we would like to emphasize that this paper primarily focuses on establishing fundamental theories.
> Within our algorithms, we have introduced theoretical guarantees, laying the foundation for the next step, which involves conducting large-scale experiments.
>
> [82] Angel X. et al., ShapeNet: An Information-Rich 3D Model Repository. arXiv, 2015.

---

### Official Review · Reviewer_gPKP · 2023-07-08

**Soundness:** 2 fair
**Presentation:** 2 fair
**Contribution:** 2 fair
**Rating:** 4
**Confidence:** 4

**Summary:**

This paper is on developing techniques for sampling from probability distributions on Grassmann manifolds and utilizing it to generating shapes. The shapes are associated with the probability distributions on Grassmann and sampling from the distribution generates shapes. The sampling itself it based on flows under vector fields associated with the gradient of potentials (negative logs of pdfs). A large amount of paper is dedicated towards defining vectors fields and flows on a Grassmann manifold assuming one knows how to do it on a Stiefel manifold. Some of these constructions are implemented using deep neural networks. The paper provides some examples of shape and texture generation and compares it to some similar methods.

**Strengths:**

-- The paper is trying to make a connection between a number of distinct and important mathematical quantities: shapes, distributions on Grassmann manifolds, random sampling, gradient flows, etc.

-- The geometry part of the paper is solid. The use of horizontal lift to remove the nuisance group is a standard technique in analysis on quotient spaces.



**Weaknesses:**


-- I find this paper somewhat confusing to read. The presentation relies on CNF and Riemannian CNF etc, while these quantities are standard flows under given gradient fields (associated with given pdfs).


-- Perhaps the authors can clarify what is different in CNFs from a integral curves or flows under vector fields such as gradient vector fields. See for example, the paper by Liu et al., Optimal Linear Representations of Images for Object Recognition, IEEE TPAMI, May 2004. These are very standard constructions for either Stiefel or Grassmann manifolds.

-- The connection with shapes is also unclear. What do the authors mean by a shape? They should explain precisely (preferably in mathematical terms) on how shapes are represented and how sampling from a distribution on a Grassmann becomes generation of a shape.

**Questions:**


See my question listed above.

The authors need to clarify in simple layman's terms the big picture of their approach. This can be done without using undefined terminology and stating ideas precisely.

**Limitations:**


Yes, they have listed some limitations of the computational steps of their framework.

---

> ### Author Rebuttal · Authors · 2023-08-07
>
> Thank you for your detailed comments and valuable feedback!
> We have prepared detailed responses to your questions below.
> We sincerely hope that these responses will address and alleviate any concerns.
>
> ****
> > **Weakness:1. I find this paper somewhat confusing to read. The presentation relies on CNF and Riemannian CNF etc, while these quantities are standard flows under given gradient fields (associated with given pdfs).**
>
> and
>
> > **Weakness:2. Perhaps the authors can clarify what is different in CNFs from a integral curves or flows under vector fields such as gradient vector fields. See for example, the paper by Liu et al., Optimal Linear Representations of Images for Object Recognition, IEEE TPAMI, May 2004. These are very standard constructions for either Stiefel or Grassmann manifolds.**
>
> We greatly appreciate your provision of the reference.
> The foundational theory of our proposed method lies in the principles of differential geometry, encompassing integral curves and gradient vector fields known as flows.
> However, finding a convenient approach to learning densities on a manifold that adhere to the symmetry imposed by the orthogonal group is not straightforward and presents non-trivial challenges.
> Moreover, the construction of manifold continuous normalizing flows that maintain the symmetry has remained unclear.
> To address these issues, we deduced implications akin to Propositions 1-3, which paved the way for a practical solution.
> This solution, predicated on obtaining the gradient of the potential function, allows for the derivation of a CNF.
> Specifically, leveraging these insights, we constructed the algorithm (GrCNF) by parametrically modeling continuous normalizing flows and realizing a modelization for computing gradients (referred to as $\mathsf{Grad}$) with respect to the potential function $v_{\operatorname{out}}=\mathsf{MILs}\left(\mathsf{HorP}\left(\boldsymbol{Y}\right)\right)$ within the proposed neural architecture.
> We assert that constructing potential functions invariant under orthogonal groups on the Grassmann manifold is notably more flexible and straightforward compared to directly parameterizing complex densities invariant under orthogonal groups or flows equivariant under orthogonal groups, as attempted in Liu et al.'s work.
>
> ****
> > **Weakness 3: The connection with shapes is also unclear. What do the authors mean by a shape? They should explain precisely (preferably in mathematical terms) on how shapes are represented and how sampling from a distribution on a Grassmann becomes generation of a shape.**
>
> In shape theory, "shape" refers to the equivalence class $\left[\boldsymbol{Y}\right]$ of all specific elements $\boldsymbol{YQ}$ obtained through transformations $\boldsymbol{Q}$ (such as affine transformations) of a single base shape $\boldsymbol{Y}$ belonging to a particular class. As shown in Eq.(1), the equivalence class is equivalent to the linear subspace $\operatorname{span}\left(\boldsymbol{Y}\right)$, making it an element of the Grassmann manifold. According to [57], shape is defined as the shape space $S\left[x_1,\cdots,x_D\right] = \\{ \xi=\left(\xi^1,\cdots,\xi^D\right), \sum_{j=1}^{j=D}\xi^jx_j=0, \sum_{j=1}^{j=D}\xi^j=0 \\},$ where $\left(x_1,\cdots,x_D\right)$ are systems of point $x_j \in \mathbb{R}^k$, which is precisely $\operatorname{span}\left(\boldsymbol{Y}\right)$.
> GrCNF treats this equivalence class $\left[\boldsymbol{Y}\right]$, i.e., "shape," as data for learning and generation. Theoretically, sampling is done for obtaining the equivalence class $\left[\boldsymbol{Y}\right]$, while on the computer, the base shape (representative $\boldsymbol{Y}$) is obtained.
> Therefore, a noteworthy characteristic of the proposed method is that the "shape" itself is learned during the learning and generation processes. During generation, one can obtain any base shape $\boldsymbol{Y}$ using a trained model on the computer.

---

### Author Rebuttal · Authors · 2023-08-07

We have included a PDF with the figures as part of our response. Kindly review the PDF to access the pertinent information and visual illustrations.
Please note that the figures in the uploaded PDF are presented in a smaller size due to limitations in available space.
Rest assured, in the Camera-ready version, we will furnish larger and more visually distinct figures for your convenience.
Should you have any additional inquiries, please don't hesitate to contact us. We value your engagement and eagerly anticipate addressing any inquiries you might have.

---

### Decision · Program_Chairs · 2023-09-21

**Decision:**

Accept (poster)

**Comment:**

This paper generalizes Continuous Normalizing Flows (CNFs) to work on Grassmann Manifolds, by representing flows on these manifolds using tangent vector fields and constructing a prior density with an efficient sampling. The Grassmann data is used to represent and learn distributions on shapes, where a “shape” $Y$ with equivalent representations $YQ$ can be represented as an element $span(Y)$ in the Grassmann. The paper presents several contributions such as modeling of shapes with Grassmann manifold, adapting CNFs to this manifold, and presenting applications to 3D problems with superior performance. Some weaknesses of this paper include experiments that are mostly low dimensional and not "real world", computational demanding divergence computation (inherited/similar to other CNFs training). Please include discussion/relation to Stiefel based CNFs in camera ready as well as incorporating other requested/promised changes during rebuttal period.